# PITX2 dosage-dependent changes in pacemaker cell state underlie sinus node dysfunction and atrial arrhythmias

Lieve E. van der Maarel [1], M. Ridwane Mungroo [1,6], Otto J. Mulleners [1,6], Mathilde R. Rivaud[1,2], Arie O. Verkerk [1,3], Karel van Duijvenboden[1], Freek H. T. Tiel Groenestege[1], Jeffrey D. Steimle [4], Lianne Fokkert[1], Corrie de Gier-de Vries[1], Mischa Klerk[1], Arie. R. Boender[1], James F. Martin [4], Bjarke Jensen[1], Gerard J. J. Boink [1,3,5], Harsha D. Devalla [1] & Vincent M. Christoffels [1] ✉

Physiologically relevant increases in transcription factor dosage and their role in development and disease remain largely unexplored. Genomic deletions upstream of the *Paired-like homeodomain transcription factor* gene (*PITX2*), identified in patients with sinus node dysfunction and atrial fibrillation and modeled in mice (*delB*), rewire the local epigenetic landscape, increasing *PITX2* expression. Here, we demonstrate that pacemaker cardiomyocytes in the embryonic *delB* sinus node ectopically express PITX2 at physiological dosages in a heterogeneous pattern. The prenatal *delB* sinus node forms discrete subdomains showing PITX2 dosage-dependent mild or severe loss of pacemaker cardiomyocyte identity. Respective subdomain sizes and severity of sinus node dysfunction and atrial arrhythmia susceptibility align with PITX2 dosage. Ectopic *PITX2c* expression in human induced pluripotent stem cell-derived pacemaker cardiomyocytes causes *PITX2* dosage-dependent transcriptional and electrophysiological changes paralleling those in *delB* mice. Our findings provide a mechanistic link between genetic variation–driven ectopic *PITX2* expression, sinus node dysfunction and atrial arrhythmogenesis, illustrating how spatiotemporally defined increases in transcription factor dosage can translate into developmental defects and disease predisposition.

Tissue-specific transcription factors (TFs) play a crucial role in regulating transcription by interacting with specific DNA sequences within non-coding regulatory DNA elements. Their function is often dosage-sensitive and spatiotemporally restricted, driving cellular states and developmental trajectories[1,2]. Consistently, a reduction in TF levels (haploinsufficiency) is strongly associated with disease and loss-of-

function variants are rarely found in the general population[3]. Genome-wide association studies have revealed thousands of trait-associated variants in regulatory elements that likely alter TF binding sites[4–6], subtly influencing target gene expression levels per allele[7,8]. Enrichment of these trait-associated variants around genes encoding TFs further indicates that subtle variations in TF dosage strongly

[1]Department of Medical Biology, Amsterdam Cardiovascular Sciences, Amsterdam Reproduction and Development, Amsterdam University Medical Centers, University of Amsterdam, Amsterdam, the Netherlands. [2]Laboratory of Experimental Cardiology, Leiden University Medical Center, Leiden, the Netherlands. [3]Department of Clinical Cardiology, Amsterdam Cardiovascular Sciences, Amsterdam University Medical Centers, University of Amsterdam, Amsterdam, the Netherlands. [4]Department of Integrative Physiology, Baylor College of Medicine, Houston, TX, USA. [5]PacingCure BV, Amsterdam, the Netherlands. [6]These authors contributed equally: M. Ridwane Mungroo, Otto J. Mulleners. ✉e-mail: v.m.christoffels@amsterdamumc.nl

contribute to trait variation. While minor variation in TF levels may lead to trait variation, larger dosage reductions can lead to severe developmental disorders[9]. The impact of reduced TF dosage on gene regulatory networks and development has been studied in hypomorphic, haploinsufficient or knockout cell or animal models[10–16]. However, trait- and disease-associated variants have been associated with both increased and decreased gene expression at similar frequencies and several genomic deletions or translocations lead to gain in expression and developmental defects[17–19]. The mechanistic understanding of how increased TF expression within trait-relevant dosages affects cellular processes remains limited and requires further investigation[20].

The paired-like homeodomain TF PITX2 is a downstream driver of the Nodal left-right asymmetry pathway and is required for development of several organ systems, including the heart[21–23]. During development, *Pitx2c* is selectively expressed in the left-sided components of the heart, including the left atrium, not in the right atrium or sinus node (SAN). *Pitx2c*-deficient mouse models have revealed that *Pitx2c* is required to impose left identity on the sinus venosus and atrium. This prevents left-sided SAN formation and SAN-associated gene expression by largely unexplored mechanisms[24–26].

Single nucleotide polymorphisms (SNPs) upstream of *PITX2* are strongly associated with atrial fibrillation (AF), the most common cardiac arrhythmia, yet the mechanisms that link *PITX2* to AF are not fully understood[27–30]. AF is characterized by rapid, irregular ectopic electrical activation in the atria or pulmonary vein, which disrupts the regular pacing of the atria by the SAN and increases risk for multiple adverse outcomes, including heart failure and stroke[31]. The major limitations of current therapeutic strategies aimed at restoring sinus rhythm or maintaining ventricular rate include their potential toxicity and proarrhythmic risk[32]. A deeper understanding of AF etiology is necessary for atria-specific interventions and those that target the underlying pathophysiology of AF. It is generally assumed that AF-associated SNPs reduce *PITX2* expression, possibly in addition to that of other (non) coding genes nearby, in left atrial (LA) or pulmonary vein tissue[33,34]. In accordance, *Pitx2* haploinsufficiency or deletion of an enhancer region required for *Pitx2* expression in the left atrium promotes an arrhythmogenic phenotype in mice[10,25,35–38]. Moreover, *PITX2* loss-of-function in human induced pluripotent stem cell (hiPSC)-derived atrial cardiomyocyte (CM)-like cells induces an AF-prone state[39,40].

Despite this, previous investigations associate AF with reduced *PITX2* expression in human left atrial samples, increased right atrial expression, or no association with AF[29,37,39–47], demonstrating that the relation between variants, direction and location of modified *PITX2* expression and AF is only partially understood. Recently, the AF-associated SNPs upstream of *PITX2* have been associated with SAN dysfunction (SND)[48], and also with AF driven by triggers originating from the sinus venosus (where the SAN is localized)[49]. In SND, the activation of the atria by the SAN is impaired due to impaired function of specialized pacemaker CMs (PCs) or propagation of the action potential (AP) to the atrial myocardium[50]. These observations indicate that a common genetic component underlies the co-occurrence of SND and AF[51,52].

We have previously identified 7 unrelated families with overlapping deletions in the gene desert 1 Mbp upstream of *PITX2*, in which affected members present with SND and AF[50]. Mice homozygous for a deletion of the paralogous region (*delB*) recapitulated this arrhythmogenic phenotype. *Pitx2* was induced in the adult *delB* SAN region and in hiPSC-derived CMs homozygous for the deletion found in one of the families, which may account for the observed arrhythmias. Here, we sought to understand the mechanism linking increases in *PITX2* dosage and the observed predisposition for SND and atrial arrhythmias (AA). We examined and quantified the spatiotemporal pattern of ectopic PITX2 expression and the transcriptional states,

morphogenesis and function of the SAN and atria of *delB* mice. Furthermore, we ectopically expressed *PITX2c* in hiPSC-derived PC-like cells, defined PITX2 dosage-dependent transcriptional state changes and assessed how these changes modify PC electrophysiology. Our results show that PITX2 mis-expression results in altered transcriptional and functional PC state, SND and atrial arrhythmogenesis in a dosage-dependent manner, revealing how spatiotemporally defined increases in TF dosage translate into developmental defects and disease predisposition.

## Results

### Early onset of ectopic PITX2 expression in *delB* PCs

Previously, we reported that adult homozygous *delB/delB* mice exhibit transcriptome changes and ectopic *Pitx2* expression in right atrial (RA) tissue that included the SAN region[50]. We investigated the location and developmental onset of *Pitx2* expression in the *delB* hearts and the cell-type(s) involved. Ectopic *Pitx2c* expression was detected by in situ hybridization in the *delB/delB* SAN as early as embryonic day (E)10.5, when SAN morphogenesis has just been initiated (Fig. 1a). In contrast, the RA and other *Pitx2*-negative tissues, such as the right side of the atrioventricular canal[53] did not show ectopic expression. Immunostaining further revealed co-expression of ISL1 and PITX2 in the differentiating PCs of the E10.5 *delB/delB* SAN (Fig. 1b). Heterozygous *delB/+* embryos display ectopic *Pitx2* expression in the SAN only, in a pattern very similar to that of *delB/delB* embryos (Fig. 1c and Supplementary Fig. 1a). In *wild-type* fetal hearts, PITX2 expression is restricted to the myocardium of the superior caval vein, left atrium and pulmonary vein[24] (Fig. 1d). However, we noted sporadic PITX2+ nuclei in the core of the SAN, revealing that *Pitx2* is sometimes activated in the SAN (Supplementary Fig. 1c). In addition, we observed PITX2+ nuclei in the SAN head domain in continuity with and extending from the PITX2+ superior caval vein myocardium. In both the *delB/+* and *delB/delB* SAN, a large fraction of PCs express PITX2, coinciding with a reduction in the proportion of PCs that express ISL1 (Fig. 1d). We conclude that while PITX2+ PCs infrequently occur in the *wild-type* SAN, the topological disruption associated with *delB* induces exaggerated and heterogeneous ectopic expression of PITX2 in PCs from the onset of SAN development.

### Progressive loss of pacemaker-associated gene expression in the *delB* SAN

In the E12.5 *Pitx2*+ *delB/delB* SAN, 3 days following the initiation of SAN formation, we observed reduced PC nuclear expression of TFs required for PC differentiation, ISL1[54], TBX3[55], and SHOX2[56,57] (Fig. 2a and Supplementary Fig. 1b). The expression of HCN4, a PC-specific ion channel responsible for phase 4 depolarization, essential for pacemaker function and embryogenesis[58], remained largely unchanged at this stage (Fig. 2a). The E14.5 SAN also showed reduced ISL1, TBX3 and SHOX2 expression, which was now accompanied by the induction of NKX2-5 expression in the SAN head periphery (Fig. 2b). This indicates that NKX2-5 expression is no longer suppressed by SHOX2[59]. Furthermore, we detected clusters of CMs in the SAN exhibiting strongly reduced HCN4 expression and ectopic CX40 expression (Fig. 2b and Supplementary Fig. 2a). By the end of gestation, at E17.5, we detected two distinct PC subdomains in both the *delB/+* and *delB/delB* SAN, HCN4low subdomains that that lack PC-associated gene expression (including ISL1, SHOX2, TBX3 and HCN4) alongside seemingly unaffected HCN4high subdomains (Fig. 2c, d and Supplementary Fig. 2b). In situ hybridization of adjacent sections reveals concomitant loss of *Hcn4* and gain of *Gja5* in the SAN at E17.5 (Supplementary Fig. 1d). 3D volumetric reconstruction showed that the volume of the *Hcn4*+ SAN domain was strongly reduced in *delB/delB* fetal mice compared to *wild-type* mice ($n = 5$ and 8, respectively; $p = 0.0086$) (Supplementary Fig. 1e, f). In summary, *delB* and the consequent ectopic activation of PITX2 expression in the SAN PCs correlate with two distinct PC states

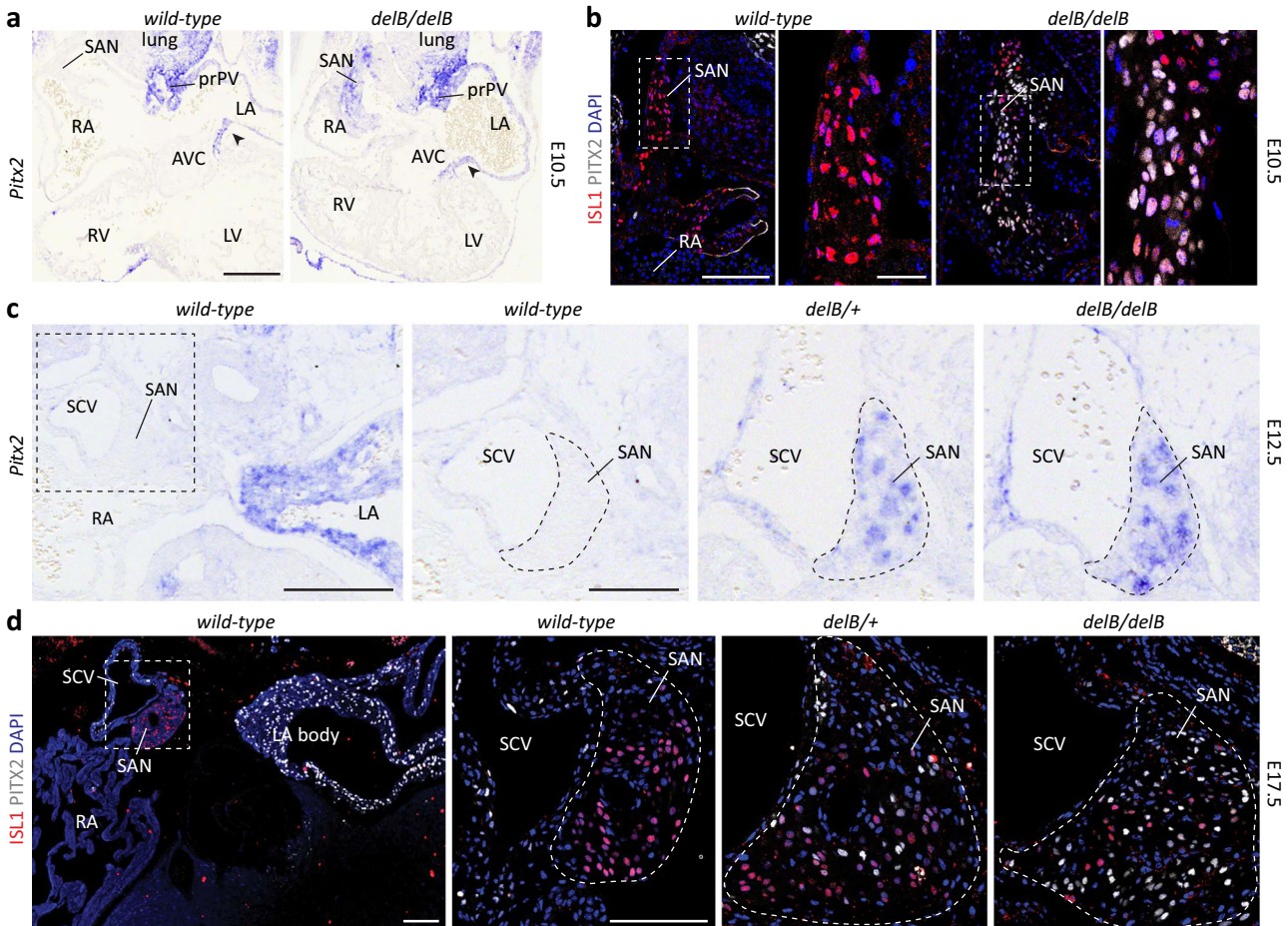

**Fig. 1 | Ectopic *Pitx2* expression in the developing *delB/delB* SAN. a** In situ hybridization reveals absence of *Pitx2* expression at E10.5 in the *wild-type* SAN (*n* = 3) and early ectopic *Pitx2* expression in the SAN alongside maintained expression in the LA, prPV, and AVC (arrow) *delB/delB* heart (*n* = 4). Scale is 200 μm. **b** Immunostaining shows absence of PITX2+ nuclei in the ISL1+ *wild-type* SAN (*n* = 3) and PITX2+ nuclei alongside ISL1+ nuclei in the *delB/delB* SAN (*n* = 3) at E10.5. Scale is 100 μm. Insets show co-localization of ISL1 and PITX2 expression in the *delB/delB* SAN. Scale is 10 μm. **c** In situ hybridization shows that *Pitx2* clearly demarcates the left from the RA and is normally absent from the SAN (*n* = 2). Scale is 200 μm. *Pitx2* is ectopically expressed in the *delB/+* (*n* = 3) and *delB/delB* (*n* = 2) SAN (outlined) at E12.5. Scale is 100 μm. **d** Immunostaining shows that PITX2 expression is normally restricted to the SVC and LA and absent from the ISL1+ SAN (*n* = 3). Ectopic PITX2 expression in the *delB/+* (*n* = 3) and *delB/delB* (*n* = 7) SAN (outlined) is heterogeneous and coincides with a loss of ISL1 expression by the end of gestation at E17.5. Scale is 200 μm and 100 μm. SAN sinus node, RA right atrium, LA left atrium, prPV primordial pulmonary vein, AVC atrioventricular canal, E embryonic day, RV right ventricle, LV left ventricle, SCV superior caval vein.

resulting from the progressive loss of essential pacemaker TF expression and their target genes and a gain in gene expression associated with the atrial myocardium (Fig. 2e).

The activation of *Pitx2* expression during SAN development may have caused the replacement of PCs by other cell types. However, the progressive loss of expression of PC-associated genes, like *Hcn4*, over the course of *delB* SAN development indicates that PITX2 expression causes differentiating PCs to change state. To determine whether PITX2 can cause a state change in fully differentiated post-mitotic PCs, we used AAV9-Myo4A expression vectors to overexpress *PITX2c* or *mCherry* in *wild-type* mice 2 days after birth (Supplementary Fig. 3a). Immunostaining for PITX2, GFP and CM-specific PCM-1 confirmed high transduction efficiency in the myocardium, including the SAN (Supplementary Fig. 3b). Both HCN4 and ISL1 expression was depleted in PITX2+ cells in the AAV9-PITX2c SAN (Supplementary Fig. 3d), demonstrating that ectopic *PITX2* expression in differentiated PCs suppresses PC gene expression.

### A unique transcriptional profile in PITX2+ PCs
To define the transcriptomes of the differentially affected HCN4^{high} and HCN4^{low} PC subdomains in the late fetal (E17.5) *delB/delB* SAN, also in

relation to the RA, we applied spatial transcriptomics on formalin-fixed, paraffin-embedded sections of *wild-type* (*n* = 3) and *delB/delB* (*n* = 6) hearts. Transcripts were detected using the GeoMx Whole Transcriptome Mouse Atlas Probe mix for NGS, containing probes for 19966 unique gene transcripts. We immunolabeled sections for TNNI3 to enrich for CMs and HCN4, which clearly demarcated the boundaries between HCN4^{high} and HCN4^{low} PCs, to segment regions of interest (ROIs) based on HCN4 and TNNI3 signal intensity. In doing so, we determined ROIs containing the *wild-type* HCN4+ TNNI3+ SAN PCs and TNNI3+ RA CMs, as well as the *delB/delB* HCN4^{high} and HCN4^{low} TNNI3+ SAN PCs and TNNI3+ RA CMs (Fig. 3a). We determined that all ROIs contained sufficient cells for reliable transcript quantification (Supplementary Data 14).

Principal component analysis (PCA) captured the distinct transcriptional profiles of the RA and SAN and those of the *wild-type* HCN4+ SAN and *delB/delB* HCN4^{high} and HCN4^{low} subdomains (Fig. 3b). Comparing *delB/delB* with *wild-type* fetal RA confirmed that the RA is transcriptionally unaffected and that *Pitx2* is not expressed in the RA (Fig. 3b–d). Both *delB/delB* HCN4^{high} and HCN4^{low} subdomains were transcriptionally distinct from the *wild-type* HCN4+ SAN (108 upregulated and 87 downregulated transcripts; $p_{adj} < 0.05$ and 286

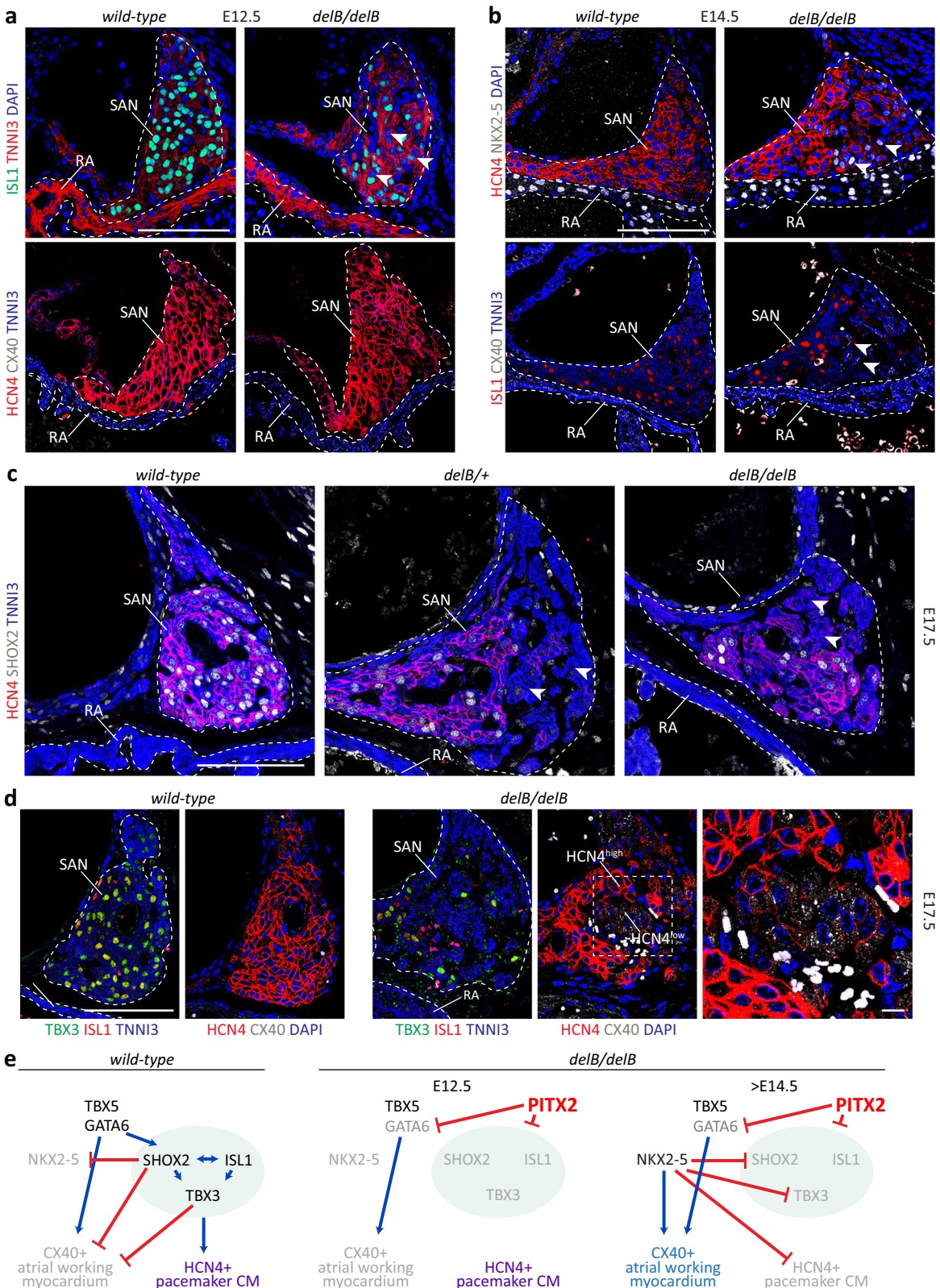

upregulated and 274 downregulated transcripts; $p_{adj} < 0.05$, respectively) and from each other (122 upregulated and 101 downregulated in *delB/delB* HCN4$^{low}$; $p$adj < 0.05) (Supplementary Fig. 4a and Supplementary Data 1–3). The expression of genes in a 3.5 Mbp region surrounding the *delB* deletion was unchanged in the *delB/delB* SAN and RA, except for *Pitx2* in the *delB/delB* SAN (Fig. 3c). Comparing the *wild-type* HCN4+ SAN to the *delB/delB* HCN4$^{high}$ and HCN4$^{low}$ subdomains

revealed a significant stepwise increase in *Pitx2* dosage from negligible in the *wild-type* SAN to increasingly higher levels in the *delB/delB* HCN4$^{high}$ and HCN4$^{low}$ subdomains, respectively (Fig. 3c and Supplementary Data 1–3). Pacemaker markers *Isl1, Tbx3, Shox2,* and *Hcn4* showed a reciprocal stepwise reduction, whereas atrial CM markers *Gja1* and *Scn5a* showed a stepwise increase in the *delB/delB* HCN4$^{high}$ and HCN4$^{low}$ subdomains (Fig. 3d and Supplementary Data 1–3).

**Fig. 2 | Progressive loss of pacemaker-associated gene expression in the developing *delB* SAN. a** ISL1 is normally expressed in PCs throughout the SAN (outlined) at E12.5 (*n* = 5). Loss of ISL1 expression was detected as early as E12.5 in the *delB/delB* (*n* = 5) SAN (arrows). The expression domain of downstream functional markers HCN4 (pacemaker marker) and CX40 (working atrial myocardium marker) remain unchanged (*wild-type*, *n* = 5; *delB/delB*, *n* = 4). Scale is 100 μm. **b** Immunostainings show progressive loss of ISL1 and HCN4 (*wild-type*, *n* = 2; *delB/delB*, *n* = 3 alongside activation of NKX2-5 and CX40 expression (arrows) in the E14.5 SAN domain (outlined, *wild-type*, *n* = 2; *delB/delB*, *n* = 2). Scale is 100 μm. **c** Clusters of TNNI3+ cardiomyocytes in the SAN domain that have lost upstream (SHOX2) and

downstream (HCN4) pacemaker gene expression by the end of gestation (E17.5, arrows) (*wild-type*, *n* = 3; *delB/delB*, *n* = 3). Scale is 100 μm. **d** Adjacent sections show co-localization of ISL1 and TBX3 expression with HCN4 in the *wild-type* SAN at E17.5 (*n* = 2). Clusters of TNNI3+ cells in the *delB/delB* SAN lack ISL1, TBX3 (*n* = 3) and HCN4 expression and express CX40 (panel, *n* = 4). Scale is 100 μm and panel scale is 10 μm. **e** Pacemaker-associated or working myocardium-associated gene expression is the product of a gene regulatory network. This gene regulatory network is progressively disrupted in the *delB/delB* SAN. Red arrows repress, and blue arrows drive. PC pacemaker cardiomyocyte, SAN sinus node, E embryonic day, RA right atrium, SCV superior caval vein, CM cardiomyocyte.

Interestingly, the expression of *Scn5a* (encoding the alpha subunit of the cardiac sodium channel essential for conduction and implicated in SND and AF[60–63]) in the *delB/delB* HCN4[low] subdomain increased to levels beyond those normally found in the atrial working myocardium (Fig. 3d and Supplementary Data 1–3). *Tbx5*, encoding a TF required for formation of the atria and SAN[64] remained unchanged (Fig. 3d). Gene ontology (GO) analysis revealed downregulation of terms associated with PC identity/function (e.g., sinoatrial node development (GO:0003163) and regulation of heart rate (GO:0002027)) and the upregulation of terms associated with the working atrial myocardium (e.g., atrial cardiac muscle cell action potential (GO:0086014) and cell communication by electrical coupling involved in cardiac conduction (GO:0086064)) (Fig. 3e and Supplementary Data 1–3).

Cluster analysis comparing gene expression in *wild-type* HCN4+ SANs, *delB/delB* HCN4[high] subdomains, HCN4[low] subdomains and *wild-type* and *delB/delB* RAs shows that *delB* PCs acquire features more similar to atrial CMs while also gaining unique characteristics (Fig. 3f). Clusters 1 and 2 show a stepwise enrichment for RA-expressed transcripts, whereas clusters 3 and 4 show a stepwise depletion of SAN-expressed transcripts (clusters 3 and 4) in the *delB* subdomains (Fig. 3f). GO analysis shows that differentially expressed genes (DEGs) in cluster 1 are linked with heart morphogenesis and development, while cluster 3 DEGs are associated with SAN development and heart rate regulation (Supplementary Fig. 4b and Supplementary Data 5). Clusters 2 and 4 show stepwise changes in gene expression beyond RA levels (Fig. 3f and Supplementary Data 5). GO analysis shows that cluster 2 DEGs are associated with metabolism, muscle contraction and sarcomere organization (Fig. 3f and Supplementary Data 5).

Genes encoding all TFs previously implicated in the development and function of the SAN (*Gata6*[65], *Shox2*[56], *Tbx18*[66], *Isl1*[54], *Tbx3*[55], *Hand2*[67]) were downregulated in *delB/delB* HCN4[low] subdomains, except for *Tbx20*[48], which was upregulated (Fig. 3g). Additionally, we find that genes that drive pacemaker activity or atrial conduction are also strongly deregulated in *delB/delB* HCN4[low] subdomains, including *Hcn4*, *Hcn1*, *Scn5a*, *Gja1*, *Gja5*, and *Kcnj3* (Fig. 3g). Ca[2+] handling plays a central role in the maintenance of PC automaticity and contractility[68–70]. Expression of key mediators of intracellular Ca[2+] transport and handling in CMs is disrupted in HCN4[low] PCs compared to the *wild-type* HCN4+ SAN, including *Pln*, *Atp2a2*, *Ryr2*, and *Calm1*. Also, *Mfn2*, involved in the maintenance of mitochondria-sarcoplasmic reticulum interactions and implicated in SND[71] is downregulated in HCN4[low] PCs compared to *wild-type* HCN4+ PCs.

Direct comparison between the *delB/delB* HCN4[high] and HCN4[low] subdomains revealed that HCN4[high] subdomains are similarly, but much less strongly transcriptionally affected compared to HCN4[low] subdomains (Fig. 3h). Thus, the direction of change of most significant DEGs is congruent in both subdomains, whereas only a few genes are uniquely differentially expressed in either the HCN4[low] or HCN4[high] subdomain. Additionally, comparing the degree by which gene expression changed in the HCN4[high] and HCN4[low] subdomains shows that the majority of DEGs (*p* < 0.001) have a stronger response in the HCN4[low] domain,

indicating that higher PITX2 expression levels coincide with stronger transcriptional deregulation (Supplementary Fig. 4c).

We next assessed how the levels of ectopic PITX2 expression in the *delB* SAN compared to those of myocardium of the LA that normally expresses PITX2. We quantified PITX2 signal intensity per nucleus in equivalent sections in the *wild-type* SAN (*n* = 2; 1073 nuclei), *delB/delB* SAN (*n* = 3; 1126 nuclei), *wild-type* LA body (LAB) (*n* = 2; 4999 nuclei) and *wild-type* LA auricle (LAA) (*n* = 2; 6445 nuclei). In the *wild-type* LA, the LAB exhibited higher PITX2 signal than the LAA, and PITX2 signal in *delB/delB* SAN nuclei was intermediate to that found in the LAB and LAB (Fig. 4a). These data show that PITX2 is ectopically expressed in SAN PCs at physiological levels.

To compare the state of PITX2+ PCs with that of PITX2+ LA CMs, we assessed the transcriptional profiles of HCN4+, HCN4[high] and HCN4[low] SAN subdomains, and of the LAB and LAA of *wild-type* (*n* = 3), *delB/+* (*n* = 6) and *delB/delB* (*n* = 3) E17.5 fetal mice (Fig. 4b and Supplementary Data 6–10). PCA grouped *wild-type*, *delB/+* and *delB/delB* LAB and *wild-type*, *delB/+* and *delB/delB* LAA ROIs together, and the transcriptomes of these regions did not overlap with those of any SAN sample (Fig. 4b), indicating that the *delB* LA is transcriptionally unaffected and showing that PITX2+ PCs do not acquire a LACM-like state. Furthermore, *Pitx2* expression levels in the *delB* SAN does not exceed physiological levels found in the LA (Fig. 4c), consistent with the PITX2 protein quantification (Fig. 4a). Comparing the transcriptomic changes in the *delB/delB* HCN4[low] subdomain with those found when comparing atrial CMs with PCs[72] confirms that PITX2+ PCs acquire some aspects of atrial CM identity (Supplementary Fig. 4d). However, comparing these transcriptomic changes to those found between *Pitx2* haploinsufficient LACMs and *wild-type* LACMs[34] suggests little overlap in the response to changes in PITX2 dosage in the SAN and LA (Supplementary Fig. 4e).

## PITX2 dosage, reduced ISL1 and a threshold response in *delB* PCs

The ectopic expression of PITX2 and reduction in ISL1 expression in both the E17.5 *delB/+* and *delB/delB* SAN are highly heterogeneous (Fig. 1d and Fig. 5a). Although we identified two distinct PC states in the *delB* SAN, nuclei expressing variable levels of PITX2 and ISL1 are distributed throughout the *delB* SAN head in a seemingly stochastic pattern (Fig. 5a and Supplementary Fig. 5a, b). To align PITX2 expression level with the loss of ISL1 expression in PCs, we quantified PITX2 and ISL1 signal intensity per nucleus in equivalent sections containing *wild-type* (*n* = 3, 224 nuclei), *delB/+* (*n* = 3, 155 nuclei) and *delB/delB* (*n* = 3, 320 nuclei) SAN head domains. Most nuclei in the *wild-type* SAN were ISL1+, except for the aforementioned sporadic PITX2+ nuclei (Fig. 5b and Supplementary Fig. 1c, Supplementary Fig. 5a). In the *delB/+* and *delB/delB* SANs, signal intensities per nucleus of both ISL1 and PITX2 were distributed across a large range, with the *delB/delB* SANs exhibiting higher PITX2 signal intensities over a larger range (Supplementary Fig. 5a, b). These data indicate a relationship between the number of *delB* alleles and the average level of PITX2 (Supplementary Fig. 5a, b). By comparing the signal intensity of ISL1 and PITX2 in individual nuclei per genotype, we found that in *wild-type*, *delB/+* and *delB/delB* SANs, the expression of PITX2 and ISL1 was

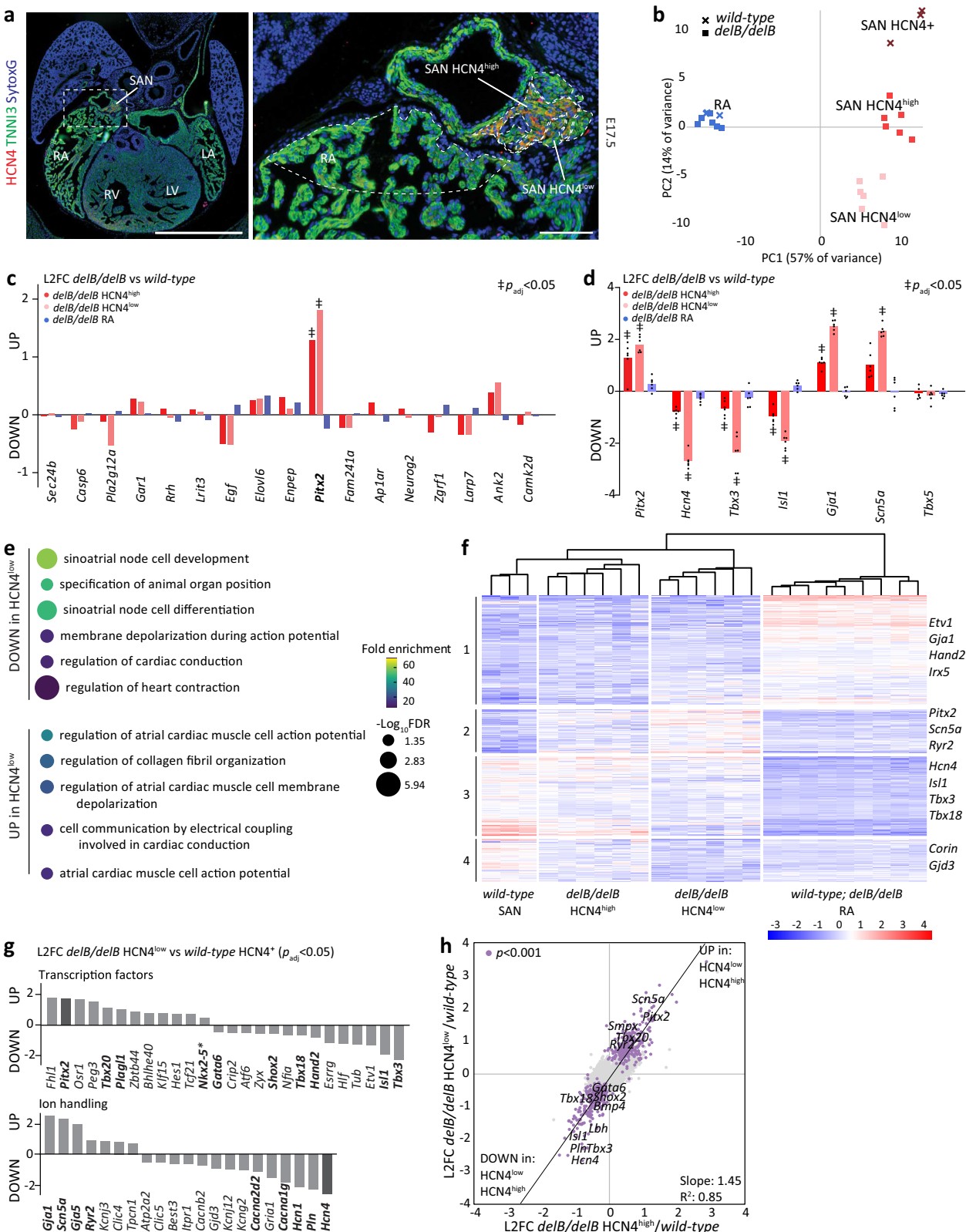

negatively correlated (Fig. 5b and Supplementary Fig. 5a). Taken together, these data indicate that PITX2 dosage-dependently suppresses ISL1 in PCs.

Transcriptomes of *delB/+* and *delB/delB* HCN4<sup>high</sup> subdomains clustered together, as well as those of *delB/+* and *delB/delB* HCN4<sup>low</sup> subdomains, both being distinct from the transcriptome of *wild-type* HCN4+ SAN (Fig. 4b). The expression profiles of key marker genes

confirmed the stepwise increase or decrease in the *wild-type* HCN4+ SAN, *delB* HCN4<sup>high</sup> and *delB* HCN4<sup>low</sup> subdomains (Fig. 5c). The direction and magnitude of the Log$_2$Fold-change of DEGs between the *wild-type* HCN4+ SAN and *delB/+* HCN4<sup>high</sup> subdomains and between *wild-type* HCN4+ SAN and *delB/delB* HCN4<sup>high</sup> subdomains, respectively, were equivalent, as were those between *wild-type* HCN4+ SAN and *delB/+* HCN4<sup>low</sup> subdomains and between *wild-type* HCN4+ SAN and

**Fig. 3 | Spatial transcriptomics reveal a heterogeneous response in the *delB/delB* SAN. a** Local changes in gene expression in the *delB/delB* SAN head and RA were defined for spatial transcriptomics by segmenting regions of interest based on HCN4 and TNNI3 expression in *wild-type* (*n* = 3) and *delB/delB* (*n* = 6) fetal (E17.5) hearts. Scale is 1 mm and 100 μm. **b** Principal component analysis shows that the *delB/delB* fetal RA is transcriptionally unaffected. HCN4low and HCN4high subdomains in the *delB/delB* SAN have distinct gene expression profiles and both are transcriptionally distinct from the *wild-type* HCN4+ SAN. **c** Comparing the expression of genes surrounding the gene desert on chromosome 4q25 to that of the *wild-type* SAN or RA reveals that *Pitx2* is significantly differentially expressed in the *delB/delB* HCN4high (*p*adj = 0.002) and HCN4low (*p*adj = 5.4E-7) subdomain compared to the *wild-type* SAN. More *Pitx2* was detected in the HCN4low subdomains than HCN4high subdomains (*p* = 0.047). Wald test corrected for multiple comparisons using the Benjamini-Hochberg method. **d** Comparing the expression of relevant PC and working atrial myocardium markers in the *delB/delB* SAN and RA with that of the *wild-type* SAN or RA, respectively, reveals step-wise changes in gene expression in the HCN4high and HCN4low *delB/delB* subdomains. The *delB/delB* RA is transcriptionally unaffected. Wald test corrected for multiple comparisons using

the Benjamini-Hochberg method. **e** Gene ontology analysis on the up- and down-regulated genes in the *delB/delB* HCN4low vs *wild-type* HCN4+ SAN. **f** Heatmap generated following a cluster analysis comparing the *wild-type* HCN4+ SAN with the *delB/delB* HCN4high and HCN4low subdomains and the *wild-type* and *delB/delB* RA. While the *delB/delB* SAN transcriptome does acquire certain aspects of the RA transcriptome (clusters 1 and 3), clusters 2 and 4 suggest that these cells acquire a unique identity that is dissimilar from that found in the RA. **g** Several key transcription factors and ion handling genes are significantly differentially expressed in *delB/delB* HCN4low cells (*p*adj < 0.05). *(*p* < 0.05). Wald test corrected for multiple comparisons using the Benjamini-Hochberg method. **h** Scatterplot showing that the *delB/delB* HCN4low domain is similarly, but more severely affected than the *delB/delB* HCN4high domain, compared to the *wild-type* HCN4+ SAN. Purple dots indicate genes that are significantly deregulated (*p* < 0.001) in both the *delB/delB* HCN4low domain and the HCN4high domain compared to the *wild-type* HCN4+ SAN. Exact *p* and *p*adj values, L2FC, fold enrichment and FDR values are listed in Supplementary Data 1–6. Source data are provided in the Source Data file. SAN sinus node, RA right atrium, E embryonic day, L2FC Log2Fold-change, FDR false discovery rate.

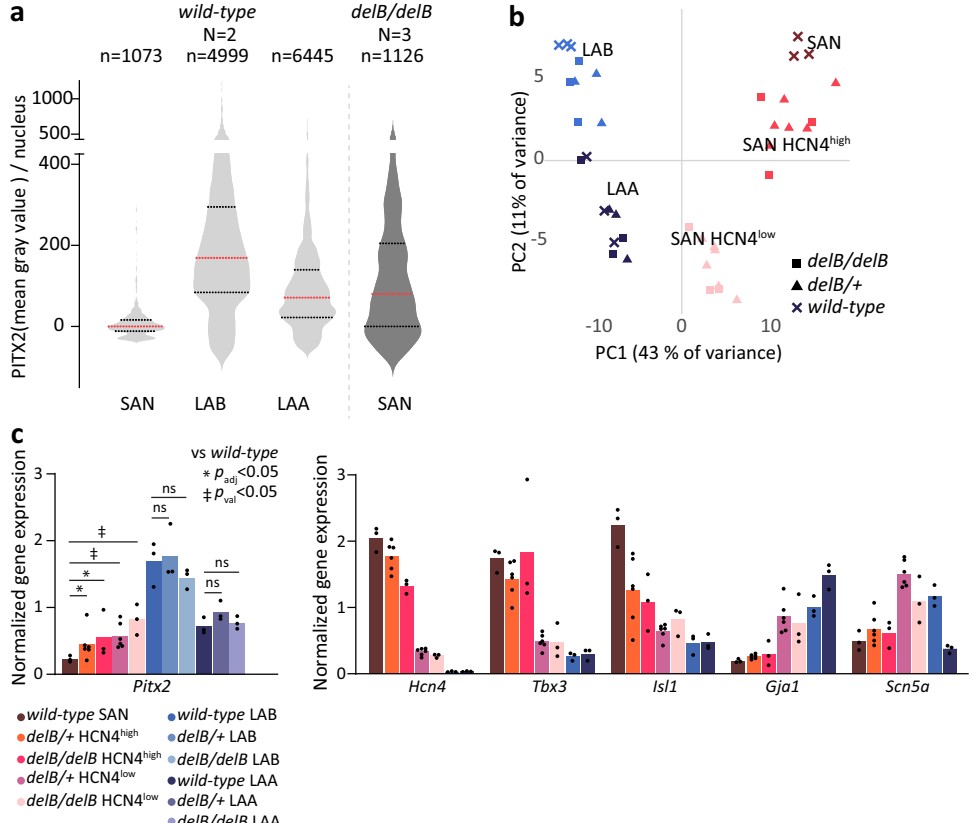

**Fig. 4 | The transcriptional profile of PITX2+ *delB* PCs is distinct from that of left atrial cardiomyocytes. a** Quantification of normalized mean gray value per nucleus in the *wild-type* (*N* = 2) SAN, LAB and LAA (1073, 4999, 6445 nuclei, respectively) and the *delB/delB* (*N* = 3; 1126 nuclei) E17.5 SAN reveals that ectopic PITX2 expression level in the *delB/delB* SAN is intermediate to that found in the *wild-type* LAB and LAA (*p* < 0.0001; two-sided Kolmogorov-Smirnov test). **b** Local changes in gene expression in the E17.5 *delB/delB* SAN head domain, LAB and LAA were defined for spatial transcriptomics by segmenting regions of interest based on HCN4 and TNNI3 expression in *wild-type* (*n* = 3), *delB/+* (*n* = 6) and *delB/delB* (*n* = 3) fetal (E17.5) hearts. Principal component analysis demonstrates the

distinct gene expression profile of the LAA and LAB and that the HCN4high and HCN4low subdomains in both the *delB/+* and *delB/delB* SAN are similarly transcriptionally affected. **c** Normalized expression of genes of interest shows that the expression levels of pacemaker- and working myocardium-associated transcription factors and ion channels in the *wild-type* (*n* = 3), *delB/+* (*n* = 6) and *delB/delB* (*n* = 3) SAN do not align with those in the LAA (*n* = 3) or LAB (*n* = 3). Wald test corrected for multiple comparisons using the Benjamini-Hochberg method. Exact *p* and *p*adj values, and L2FC are listed in Supplementary Data 6–14. Source data are provided in the Source Data file. E embryonic day, SAN sinus node, LA left atrium, LAB left atrial body, LAA left atrial auricle, L2FC Log2Fold-change.

*delB/delB* HCN4low subdomains (Fig. 5d). These comparisons indicate that the cell state changes in the *delB* HCN4high and HCN4low subdomains are independent of genotype and that PCs exhibit a threshold response to PITX2 dosage.

We next quantified the fraction of PC nuclei in HCN4high and HCN4low subdomains in *wild-type* (*n* = 5), *delB/+* (*n* = 6) and *delB/delB* (*n* = 6) fetal hearts and found that the *delB/delB* HCN4high subdomain contains a smaller fraction of total PC nuclei and the HCN4low

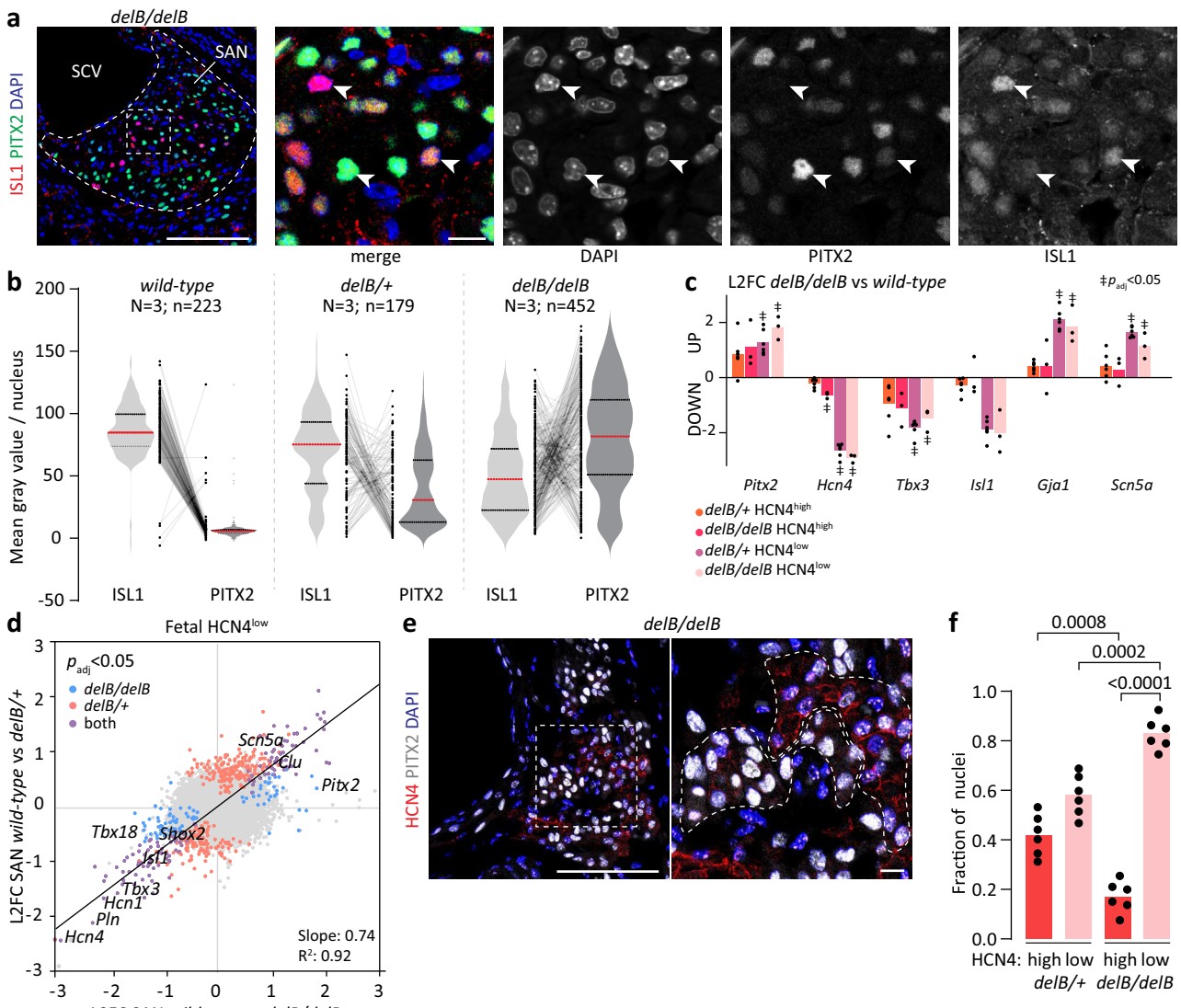

**Fig. 5 | Dosage-dependent changes in *delB* pacemaker cardiomyocytes and concomitant gain of PITX2 expression and loss of ISL1 expression. a** Fetal (E17.5) *delB/delB* SAN nuclei exhibit variable ISL1 and PITX2 expression levels. Scale is 100 μm and panel scale is 10 μm. **b** Violin plots showing the quantification of normalized mean gray value per nucleus in *wild-type* (*N* = 3; *n* = 223), *delB/+* (*N* = 3; *n* = 176) and *delB/delB* (*N* = 3; *n* = 452) fetal SANs (two-sided Kolmogorov-Smirnov test, *p* < 0.0001). There is an overall incremental gain in PITX2 and reduced ISL1 expression levels in both the *delB/+* and *delB/delB* SAN, and a progressive shift in PITX2:ISL1 expression balance in the *delB/+* and *delB/delB* fetal SAN towards less ISL1 and more PITX2. Red line indicates median. Two-sided Kolmogorov-Smirnov test *p* < 0.0001. **c** Relative expression levels of genes of interest show stepwise, *Pitx2*-correlated changes in the expression levels of pacemaker- and working myocardium-associated transcription factors and ion channels in the *delB/+* (*n* = 6)

and *delB/delB* (*n* = 3) SAN compared to the *wild-type* (*n* = 3) SAN. Wald test corrected for multiple comparisons using the Benjamini-Hochberg method. **d** Scatterplot showing the relative expression of genes in the *delB/+* and *delB/delB* HCN4[low] subdomains compared to the *wild-type* HCN4+ SAN demonstrates that while the *delB/+* HCN4[low] subdomains are less severely affected than their *delB/delB* counterparts, differentially expressed genes behave similarly. Wald test corrected for multiple comparisons using the Benjamini-Hochberg method. **e** Variable PITX2 expression levels in the *delB/delB* HCN4[high] and HCN4[low] subdomain at E17.5. Scale is 100 μm, and panel scale is 10 μm. **f** Fraction of total nuclei in HCN4[high] and HCN4[low] subdomain in equivalent sections of the *delB/+* (*n* = 6) and *delB/delB* (*n* = 6) fetal SAN head domain. Two-tailed paired T test. Source data are provided in the Source Data file. E embryonic day, SAN sinus node, L2FC Log₂Fold-change.

subdomain contains a larger fraction of total PC nuclei compared to their respective subdomains in *delB/+* SANs (Fig. 5e, f). This indicates that the higher average PITX2 levels in *delB/delB* PCs compared to *delB/+* PCs translate to substantially larger fraction of strongly affected PCs (HCN4[low]) in the *delB* SAN.

### *PITX2c* expression in hiPSC-PCs mimics transcriptional changes in the *delB* SAN

Our findings in *delB* mice indicate a causal relationship between ectopic PITX2 expression, PC state changes in deletion carriers and

SND, but do not directly address the conservation of this mechanism. To investigate these aspects, cultures of hiPSC-PCs[73] were transduced with AAV6 expression vectors driving either *PITX2c-p2a-H2B-EGFP* or *mCherry-p2a-H2B-EGFP* (control) expression by the CM-specific *TNNT2* promoter, followed by single-cell RNA sequencing (scRNA-seq) 7 days after transduction (Supplementary Fig. 6a). Flow cytometry revealed >85% transduction efficiency and expression over a wide range of levels (Supplementary Fig. 6b). Based on the expression of *TNNT2*, we determined that clusters 0, 1, 2, 4, 7 and 8 represented CMs (Fig. 6a, b). Cluster 0 (C0) is the largest cluster and represents PC-like cells (*ISL1,*

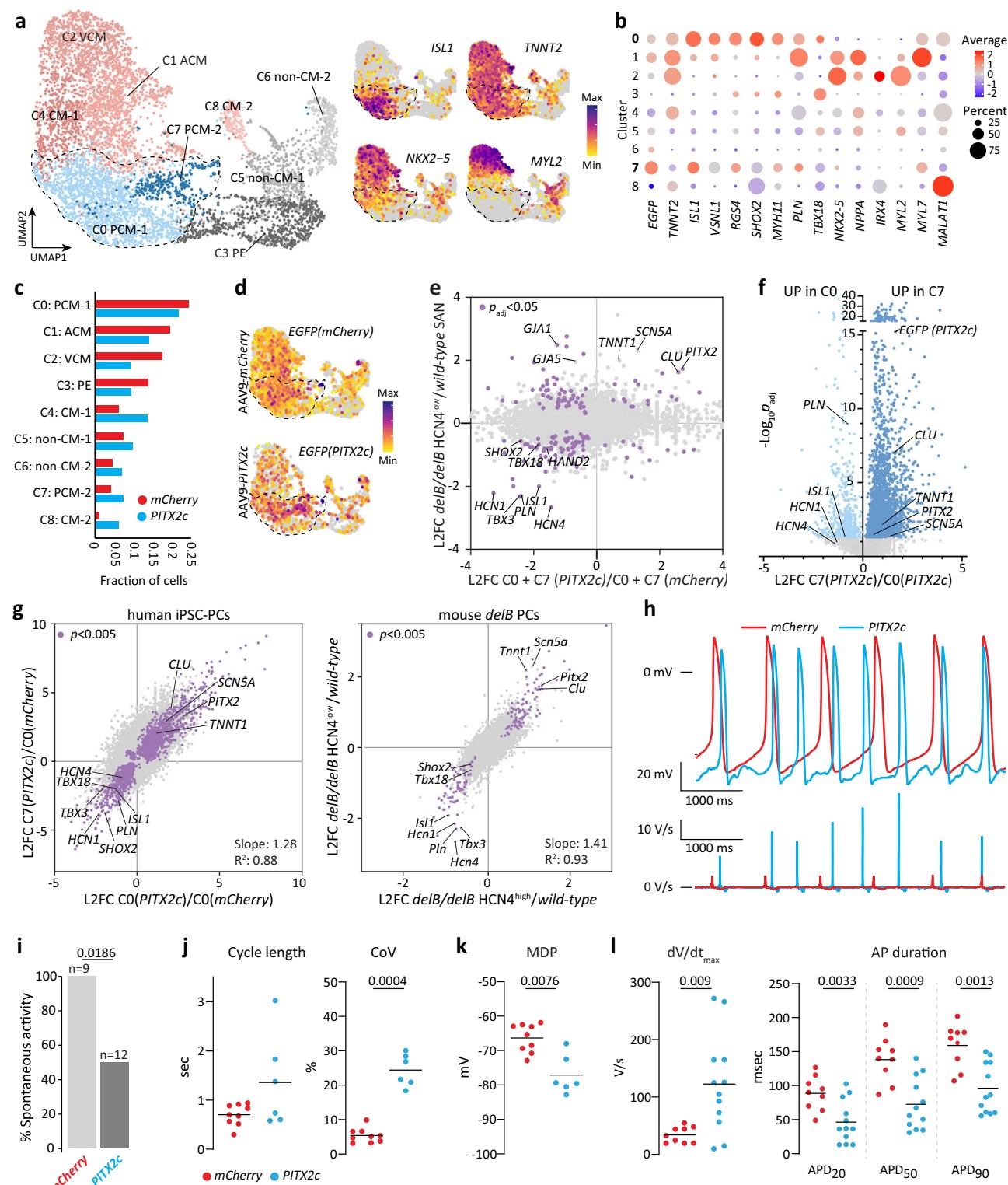

*SHOX2, TBX3, HCN4, RGS4, NKX2-5*$^{low}$) while C1 represents atrial-like CMs (*NPPA, NPPB*), C2 ventricular-like CMs (*MYL2, IRX4*) and C4 CMs of a mixed subtype (Fig. 6b and Supplementary Fig. 6d). C7 expressed PC-associated markers at lower levels compared to C0 (Fig. 6b and Supplementary Fig. 6d). C8 was separated from the other clusters in the Uniform Manifold Approximation and Projection (UMAP) and was highly *MALAT1* positive, indicating these cells were stressed or damaged[74,75] (Supplementary Fig. 6d). C8 is disproportionally composed of AAV9-*PITX2* cells, indicating that these cells may be stressed by high (supra-physiological) *PITX2* levels (Supplementary Fig. 6c).

Non-CM C3 expresses *TBX18* and *TCF21*, suggesting a proepicardial-like cell state, which was also identified in previous equivalent differentiations[73]. C5 and C6 expressed fibroblast, endothelial and mesenchymal marker genes. As C0 and C7 represent PC-like cells and are closely associated in the UMAP, we decided to focus on these two clusters.

We found that all established PC-enriched genes downregulated in fetal mouse *delB* SAN were also downregulated in AAV6-*PITX2c* PCs in C0 and C7 (*ISL1, SHOX, TBX18, TBX3, HCN4, BMP4, PLN, RGS4, HAND2,* etc.) (Fig. 6e, Supplementary Data 11–12). Moreover, several

**Fig. 6 | Ectopic *PITX2c* expression in human iPSC-derived PCs disrupts transcriptional state and function. a** UMAP representation of the single-cell transcriptomics of hiPSC-derived PCs transduced with AAV6-*mCherry* or AAV6-*PITX2c* alongside the distribution of PC and working myocardium-associated marker expression. PC C0 and C7 are outlined. **b** Heat-map showing the proportion of cells per cluster expressing PC and working myocardium-associated genes and the average expression of those genes per cluster. **c** The contribution of AAV6-*mCherry* (red) or AAV6-*PITX2c* (blue) PCs to each cluster. **d** Heat-map showing the distribution of *EGFP* expression in AAV6-*mCherry* and AAV6-*PITX2c* cells. PC C0 and C7 are outlined. **e** Scatterplot showing the relative expression of genes in the mouse *wild-type* HCN4+ SAN and *delB/delB* HCN4^low SAN compared to that of AAV6-*PITX2c* PC C0 and C7 and AAV6-*mCherry* C0 and C7 demonstrating that the most significantly deregulated PC genes are deregulated similarly in *delB* mice and AAV6-*PITX2c* hiPSC-PCs. Purple dots indicate genes that are significantly differentially expressed ($p_{adj} < 0.05$) in both comparisons. Wald test corrected for multiple comparisons using the Benjamini-Hochberg method. **f** Expression analysis of AAV9-*PITX2c* PC C0 (light blue) and C7 (dark blue) indicates a dichotomous response in AAV9-*PITX2c* hiPSC-PCs that correlates with *PITX2c* expression level. Wald test corrected for multiple comparisons using the Benjamini-Hochberg method. **g** Scatterplots showing the relative expression of genes in hiPSC-PCs and the mouse SAN, indicating that C0 (*PITX2*) is comparable to the *delB/delB* HCN4^high subdomain

while C7 is comparable to the *delB/delB* HCN4^low subdomain. Key affected genes behave similarly in hiPSC-PCs and the mouse SAN. Purple dots indicate genes that are significantly differentially expressed ($p_{adj} < 0.05$) in both comparisons. Wald test corrected for multiple comparisons using the Benjamini-Hochberg method. **h** Typical examples of spontaneous activity (top panel) and the AP upstroke velocity ($V_{max}$; bottom panel) in AAV6-*mCherry* (red) and AAV6-*PITX2c* (blue) hiPSC-PCs. **i** While all measured AAV6-*mCherry* PCs ($n = 9$) showed spontaneous activity, only 50% of AAV6-*PITX2c* PCs ($n = 12$) are spontaneously active (two-tailed Fisher's exact test, $p = 0.0186$). **j** Of the AAV6-*PITX2c* PCs that retained spontaneous activity ($n = 6$), *PITX2* increased the coefficient of variation (CoV; SD/average) (two-tailed Mann-Whitney test, $p = 0.0004$) while differences in cycle length did not reach the threshold of significance. **k** Dot plot showing that the MDP is lower in AAV6-*PITX2c* PCs ($n = 6$) compared to AAV6-*mCherry* controls ($n = 9$) (two-tailed Mann-Whitney test, $p = 0.0076$). (**l**) $V_{max}$ ($p = 0.009$) and AP duration at 20% ($p = 0.0033$), 50% ($p = 0.0009$) and 90% ($p = 0.013$) repolarization following Kir2.1 injection under dynamic clamp conditions (two-tailed Mann-Whitney test). AP parameters were established for 9 AAV6-*mCherry* control and 12 AAV6-*PITX2c* PCs. Exact *p* and $p_{adj}$ values and L2FC values are listed in Supplementary Data 1, 2, 11 and 12. Source data are provided in the Source Data file. PC pacemaker cardiomyocyte, SAN sinus node, C cluster, CM cardiomyocyte, L2FC Log₂Fold-change, AP action potential, MDP maximal diastolic potential.

functionally relevant atrial CM genes were upregulated, including *SCN5A* and *CLU*, which were also strongly upregulated in *delB* SANs (Fig. 6e). The level of *PITX2* across C0 and C7 cells (combined) negatively correlated with the levels of several PC genes, and positively correlated with the levels of several of the upregulated atrial genes (Supplementary Fig. 6g). While C0 is slightly enriched for AAV6-*mCherry* cells, C7 is enriched for AAV6-*PITX2c* cells (Supplementary Fig. 6c). Moreover, *PITX2* is expressed at higher levels in C7 than in C0 (Fig. 6f). These data suggest that C0 and C7 represent PC-like cells that responded to PITX2 in a dosage-dependent manner. Comparing transcriptional changes of C0-*PITX2c* with those of C7-*PITX2c* revealed a transcriptome response that correlates with *PITX2c* expression level (Fig. 6g). The majority of PC-associated genes were more downregulated in C7-*PITX2c* than in C0-*PITX2c*, whereas atrial genes like *CLU* and *SCN5A* were more upregulated in C7-*PITX2c* than in C0-*PITX2c*. When comparing differential expression between C0-*PITX2c* vs C0-*mCherry* and C7-*PITX2c* vs C0-*mCherry*, we found that the vast majority of significantly DEGs showed the same direction of change in C0 and C7, but that the magnitude of change was larger in C7 compared to C0. Differential expression between *delB/delB* HCN4^high subdomains vs *wild-type* HCN4+ SAN and HCN4^low subdomains vs *wild-type* HCN4+ SAN, respectively, showed very similar behavior (Fig. 6g). Moreover, of many genes, including the PC-associated gene set (*ISL1*, *SHOX2*, *TBX3*, *HCN4*, *PLN*, etc.) and atrial genes (*SCN5A*, *CLU*), both direction of change and magnitude difference were conserved between the hiPSC-PC clusters and the *delB/delB* HCN4^high and HCN4^low subdomains. Together, our data suggest that human PCs exhibit a PITX2 dosage-dependent threshold response reminiscent of that observed in *delB* PCs.

To assess the electrophysiological consequences of the *PITX2* dosage-dependent transcriptional state changes described above, we next performed a series of AP measurements on hiPSC-PCs transduced with AAV6-*mCherry* (control) or AAV6-*PITX2c*. First, we recorded the APs of 9 AAV6-*mCherry* and 12 AAV6-*PITX2c* PCs in unstimulated conditions. While all AAV6-*mCherry* hiPSC-PCs showed spontaneous activity, only 50% of AAV6-*PITX2c* PCs showed spontaneous activity (Fig. 6h, i). Spontaneous activity was also more variable in AAV6-*PITX2c* PCs, as demonstrated by the increase coefficient of variation (CoV; SD/average) (Fig. 6j). There was no significant difference in cycle length (Fig. 6j). Finally, the maximal diastolic potential (MDP) was more hyperpolarized (Fig. 6k), the maximum AP upstroke velocity ($V_{max}$) was increased and AP duration at 90% repolarization was significantly

shorter AAV6-*PITX2c* PCs compared to controls (Supplementary Fig. 7a).

To test whether the higher $V_{max}$ AAV6-*PITX2c* PCs are due to the more hyperpolarized MDP or increased *SCN5A* expression level, we additionally measured APs elicited at 1 Hz under conditions of a Kir2.1 injection using dynamic clamp, as illustrated in Supplementary Fig. 7b. Using dynamic clamp, APs could be evoked in all hiPSC-PCs measured, and Fig. 7c shows representative APs (top panel) and the $V_{max}$ in AAV6-*mCherry* and AAV6-*PITX2c* PCs (bottom panel). Under dynamic clamp conditions, MDP in AAV6-*PITX2c* PCs was similar to that of controls (Supplementary Fig. 7d), thus the Na⁺ channel availability will not be affected by differences in membrane potential. Again, we observed an increased dV/dt (Fig. 6l), which is thus due to an increased expression of *SCN5A*. In addition, shorter APs were observed (Fig. 5l). Taken together, our patch clamp data confirm that the transcriptional state changes that occur upon ectopic *PITX2c* expression in hiPSC-PCs substantially alter PC electrophysiology, including PC spontaneous activity, $V_{max}$ and AP duration.

## Transcriptional changes, SND and atrial arrhythmias in adult *delB* mice

To translate the PITX2 dosage-associated changes in SAN composition in *delB/+* and *delB/delB* mice to cardiac function and arrhythmogenesis, we examined ECG parameters of adult *wild-type*, *delB/+* and *delB/delB* mice. Several abnormal ECG phenotypes were observed at rest in both *delB/+* and *delB/delB* mice, including alternative P wave deflections and morphologies, both with and without sinus pauses, junctional rhythms and ectopic atrial beats (Fig. 7a and Supplementary Fig. 8; Supplementary Table 1). When we compared the direction of the first P wave deflection in *wild-type*, *delB/+* and *delB/delB* mice, we found that both *delB/+* ($n = 27$) and *delB/delB* ($n = 45$) mice had negative P wave deflections more often than *wild-type* ($n = 33$) animals ($p = 0.0003$ and $p < 0.0001$, respectively). *DelB/delB* mice had significantly more frequent negative first P wave deflections than *delB/+* mice ($p = 0.0015$) (Fig. 7b and Supplementary Table 1). Unlike *delB/delB* mice, *delB/+* mice displayed neither slower heart rate nor higher heart rate variability compared to *wild-type* mice (Fig. 7c, d and Supplementary Table 2). Wenckebach cycl length (WBCL) was prolonged in *delB/delB* mice ($p = 0.0146$) but not in *delB/+* mice (Supplementary Fig. 9a). Other baseline ECG parameters, including PR interval, P and QRS duration, and QT interval of *delB/+* and *delB/delB* mice, were unaffected compared to *wild-type* mice (Supplementary Table 2).

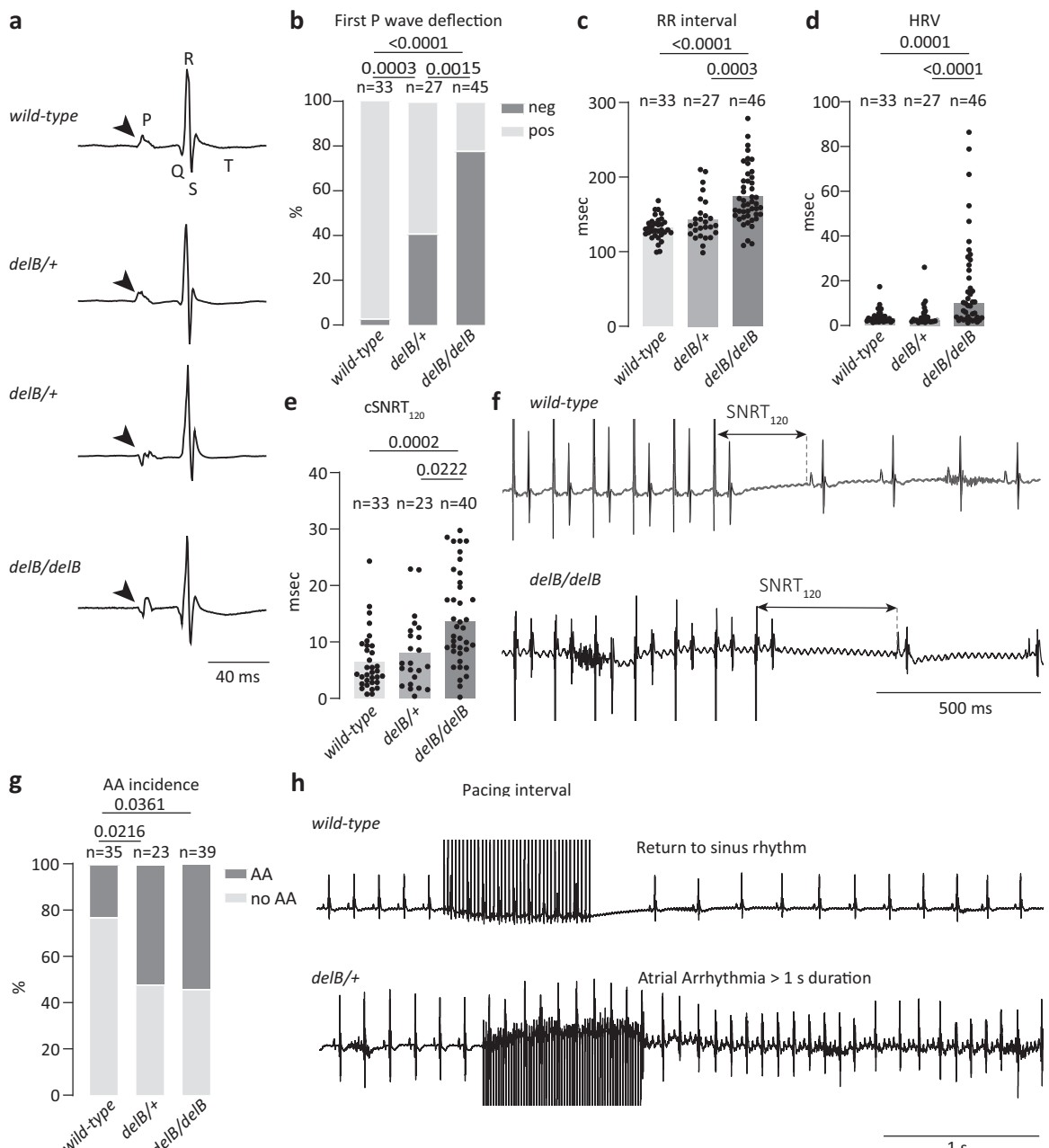

**Fig. 7 | *DelB/delB* mice present with sinus node dysfunction and are susceptible to atrial arrhythmias. a** Typical examples of ECGs of adult *wild-type*, *delB/+* and *delB/delB* mice. Arrows point to the first deflection of the P-wave, indicating negative deflections and thus, ectopic activation of the atria in *delB/+* and *delB/delB* mice. **b**–**d** Bar graphs showing the direction of the first P wave deflection (Chi-square test), RR interval and HRV (standard deviation of RR interval) in *wild-type*, *delB/+* and *delB/delB* mice. (Kruskal-Wallis test with Dunn's multiple comparisons test). **e** cSNRT$_{120}$ in *wild-type*, *delB/+* and *delB/delB* mice; typical examples show

cSNRT$_{120}$ calculated from the last stimulus artifact to the first spontaneous P-wave (Ordinary one-way ANOVA and Tukey's multiple comparisons test). **f** Representative examples of ECG traces showing SNRT$_{120}$. **g** Incidence (Chi-square, $p < 0.05$) of induced AA in *wild-type*, *delB/+* and *delB/delB* mice. **h** Typical examples of induced AAs. Source data are provided in the Source Data file. ECGs electrocardiograms, HRV heart rate variation, AA atrial arrhythmia, cSNRT$_{120}$ corrected sinus node recovery time after 120 ms pacing.

We supplied overdrive suppression to the RA through in vivo atrial stimulation using a transesophageal catheter to examine SAN recovery time (SNRT) at 120 msec stimulation cycles (SNRT$_{120}$). In *delB/delB* mice, but not in *delB/+* mice, SNRT$_{120}$ was significantly prolonged compared to *wild-type* mice (Fig. 7e, f). Both *delB/+* and *delB/delB* mice were more susceptible to AA upon atrial stimulation than *wild-type* mice ($p = 0.0216$ and $p = 0.0361$, respectively) but there was

no significant difference in AA susceptibility between *delB/+* and *delB/delB* mice (Fig. 7g, h). Total AA duration upon stimulation was comparable between genotypes (Supplementary Fig. 9b).

Atropine and propranolol were administered to block the autonomic nervous system (ANS) and test intrinsic SAN function. ANS block prolonged the RR interval, heart rate variation, WBCL and AA incidence across all genotypes but did not change the SNRT$_{120}$

(Supplementary Fig. 9c–g). ANS block did not change genotype-associated ECG phenotypes, except for a greater number of *delB/delB* mice with a regular rhythm and single P wave shape ($p < 0.05$, Supplementary Table 3). ANS block did not change the genotype-dependent differences in RR interval, heart rate variation, $SNRT_{120}$, AA incidence or WBCL (Supplementary Fig. 9c–g). Together, these results confirm that SND and atrial arrhythmogenesis in *delB* mice are both SAN-intrinsic and independent of ANS function. Notably, the milder arrhythmogenic phenotype seen in *delB/+* mice compared to *delB/delB* mice suggests a tolerance threshold for PITX2 dosage in the SAN, beyond which SAN function is impaired.

To connect *PITX2* dosage-dependent transcriptional deregulation with the arrhythmogenic phenotype in adult *delB* mice, we performed spatial transcriptomic analysis of the adult *wild-type* ($n = 3$), *delB/+* ($n = 3$) and *delB/delB* ($n = 3$) TNNI3+ SAN and adjacent RA (Supplementary Fig. 10a). We were unable to discriminate between HCN4^low or HCN4^high subdomains due to the very weak HCN4 signal in adult heart sections. PCA clearly separated the SAN from RA samples, confirming their distinct transcriptional profiles (Supplementary Fig. 10b). Comparing the *delB/delB* SAN with the *wild-type* SAN identified 108 up- and 97 downregulated genes ($p < 0.01$) whereas *delB/+* SAN only showed 23 DEGs ($p < 0.01$) relative to the *wild-type* SAN (Supplementary Fig. 10c and Supplementary Data 15, 16). These findings are consistent with the genotype-dependent electrophysiological changes of adult *delB* mice. We found that 89 genes were deregulated in both the fetal *delB/delB* HCN4^low subdomain and the adult *delB/delB* SAN ($p < 0.01$), and the direction of change was largely consistent across datasets (Supplementary Fig. 10d). Therefore, the transcriptional changes established in *delB* PCs during development seem to persist into adulthood. Furthermore, while the fetal *delB/delB* RA (Fig. 3b–d) and adult *delB/+* RA were transcriptionally unaffected (Supplementary Fig. 10e), adult *delB/delB* RA were affected (169 ($p < 0.01$) or 52 ($p_{adj} < 0.05$) DEGs compared to adult *wild-type* RA) (Supplementary Fig. 10e and Supplementary Data 17). GO analysis of the DEGs ($p_{adj} < 0.05$) showed enrichment for terms associated with cardiac stress (including GO:0042026 and GO:0009408; Supplementary Fig. 10f and Supplementary Data 18). Together, these data suggest that while the deregulation of PC state is initiated during development and persists into adulthood, a cardiac stress response is initiated in the adult *delB/delB* RA that is absent from the adult *delB/+* RA and the fetal *delB* RA.

## Discussion

Here, we investigated a mouse model, *delB*, that recapitulates a human cardiac arrhythmia syndrome associated with *PITX2*[50] to examine the transcriptional and functional consequences of physiologically relevant increases in TF dosage. We found that in *delB* mice, PITX2 is ectopically expressed in SAN PCs in a heterogeneous pattern from the onset of SAN morphogenesis onwards. Our analyses indicate that mis-expressed PITX2 alters PC state and function in a dosage-dependent manner in both *delB* mice and in a human PC model. PCs that express PITX2 at lower average levels acquire a mildly altered HCN4^high state, and those expressing PITX2 at higher average levels acquire a strongly altered HCN4^low state. These observations indicate that PCs exhibit a PITX2 dosage-dependent threshold response. We propose a model to explain distinct PC phenotypes and functional consequences across a range of PITX2 dosages (Fig. 8a). Compared to heterozygous *delB* PCs, homozygous *delB* PCs express on average more PITX2, causing more PCs to pass threshold and form larger HCN4^low subdomains of dysfunctional PCs within the SAN at the expense of functional PCs, and will therefore exhibit more severe SND. We hypothesize that also in a large fraction of human deletion carriers, elevated PITX2 dosage results in a greater proportion of PCs acquiring a dysfunctional HCN4^low state and that this is sufficient to reach the threshold for SND, and, subsequently, AF.

PC specification and differentiation is under the control of a TF network that includes relatively broadly expressed TFs TBX5 and GATA6 and PC-specific TFs ISL1, SHOX2 and TBX3[76–79] (Fig. 2e). This network drives the expression of genes essential for PC function, including those that encode key ion channels like *Hcn4*, which is necessary for pacemaker function and heart contractions during embryogenesis[58], while suppressing genes encoding TFs and functional proteins that may interfere with SAN function, including atrial CM-enriched *Nkx2-5, Scn5a, Gja1* and *Gja5*[59,76]. PC differentiation or function is impaired upon reduction or loss of any of the aforementioned PC TFs[11,54–57,59,80]. Our immunofluorescence analysis revealed that this essential PC TF network collapsed in HCN4^low PCs, causing loss of PC identity and function. *Gata6* expression is approximately 2-fold reduced in the *delB/delB* HCN4^low subdomain (Fig. 3g). In mice, *Gata6* haploinsufficiency predisposes to AA and SND[65]. Closer inspection of the histological data in this study suggests *Gata6* haploinsufficient fetal SANs form subdomains of more or less affected PCs, somewhat reminiscent of the subdomains we observed in *delB* fetal SANs, indicating a GATA6 dosage-dependent threshold response in PCs. Because TF networks that control cell state are themselves stabilized by positive feedback and autoregulation, repression of ectopically expressed PITX2 in altered PCs is unlikely to be sufficient to re-establish the PC TF network.

Ectopic *PITX2c* expression in *delB* SAN PCs and in hiPSC-PCs induced transcriptional state changes that disrupted core PC electrophysiological properties, including their ability to spontaneously depolarize. PITX2+ PCs show less spontaneous activity, impairing AP initiation, while altered SAN morphology, and thus the SAN-RA interface, may hinder AP propagation from the SAN to the surrounding working myocardium, resulting in chronic, irregular atrial activation. Several genes upregulated in *delB* PCs are known to interfere with SAN function. The TFs SHOX2 and NKX2-5 regulate PC fate through a mutually antagonistic mechanism that defines the SAN-RA boundary during development[53,59,81]. In the *delB/delB* HCN4^low domain, loss of SHOX2 expression coincides with NKX2-5 mis-expression (Fig. 2b, Supplementary Fig. 1b), likely contributing to the activation of the atrial CM gene program. Both HCN4^low PCs and AAV6-*PITX2c* PCs mis-express the cardiac sodium channel (*SCN5A*/Nav1.5) and high-conductance gap junction subunits *Gja5*/CX40 and *Gja1*/CX43 required for myocardial conductivity[82]. $V_{max}$ was increased in AAV6-*PITX2c* PCs, consistent with increased Nav1.5-driven sodium current. Activation of these atrial-enriched genes in a subset of PCs is likely to exacerbate SND, as their expression in PCs may disrupt AP synchronization or the generation of sufficient current to overcome current to load mismatch[69,70,83–85].

PC function requires strictly regulated $Ca^{2+}$ handling, and several genes involved in this process, including *Pln, Atp2a2, Ryr2, Calm1*, and *Mfn2*[71,86,87], were deregulated in *delB* SANs. Transitional PCs, a phenotypic intermediate between core PCs and atrial CMs, are essential to overcome the current to load mismatch and activate the atrial myocardium[83,84,88]. Transitional PC dysfunction causes SAN exit block, as is seen in transition zone-specific *Shox2*-deficient mice or heterozygous *SCN5A* loss-of-function allele carriers[59,61,88]. In addition to SND, we observed a possible exit block in *delB/delB* mice, suggesting that transitional PCs are also affected (Supplementary Table 1 and Supplementary Fig. 8).

Compared to working CMs, PCs are characterized by under-developed contractile machinery, few mitochondria, and low metabolic demand relying on glycolysis rather than mitochondrial respiration[71,89,90]. In CMs, PITX2 maintains mitochondrial structure and function and $Ca^{2+}$ homeostasis[10,43,45,91–95]. *PITX2* deficiency in left atrial CMs or iPSC-derived atrial CMs causes mitochondrial dysfunction and induces a metabolic shift from mitochondrial respiration towards glycolysis, while gain-of-function activates genes encoding electron transport chain components and the expression of reactive oxygen

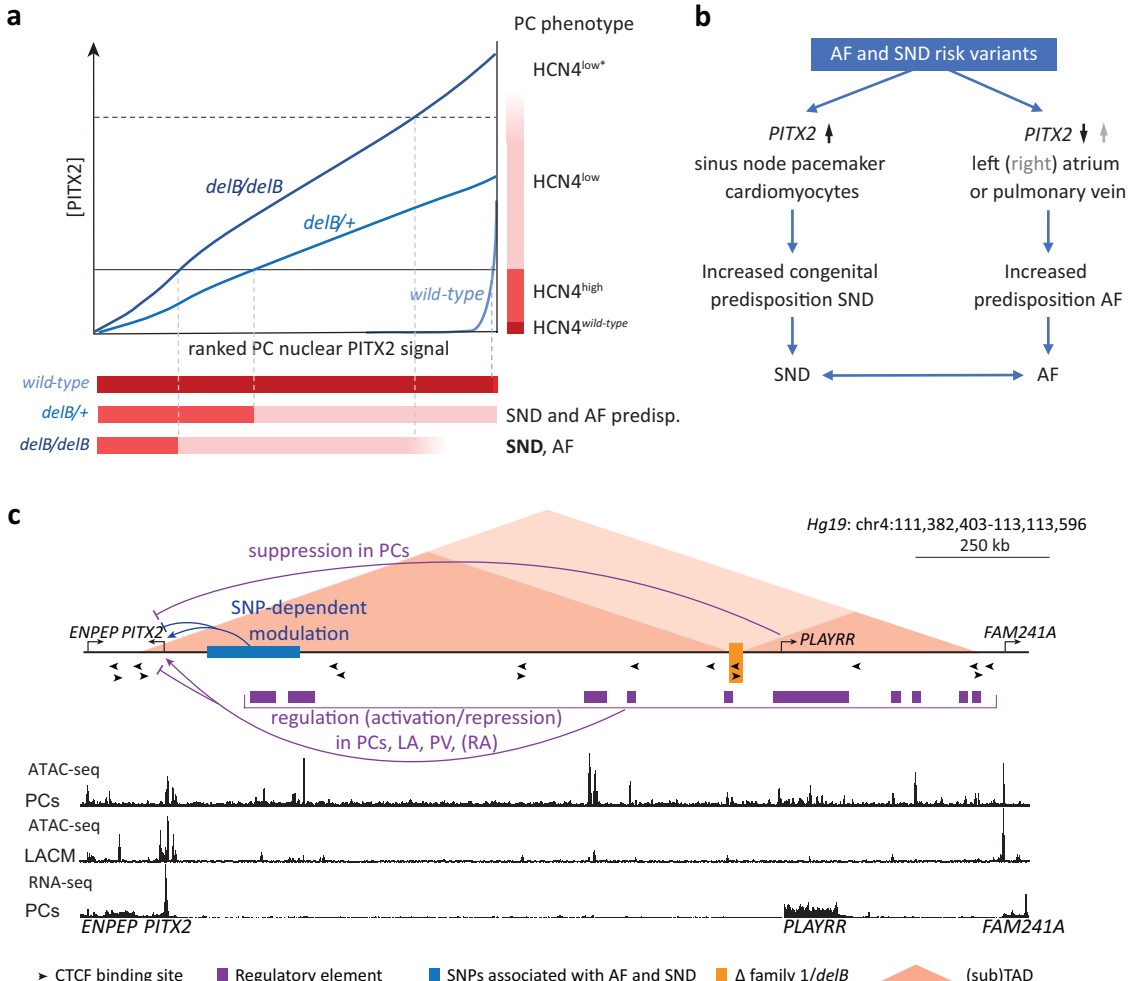

**Fig. 8 | Ectopic *Pitx2* expression dosage-dependently disrupts the PC gene program, providing a mechanistic link between PITX2, SND and AF. a** PITX2 dosage-dependent PC phenotype in the *wild-type*, *delB/+*, *delB/delB* SAN and how the proportion of affected PCs translates into predisposition for atrial arrhythmias and SND. **b** We propose that nucleotide variants modulate PITX2 expression levels in the SAN, atria and PV myocardium, driving SND and AF. **c** Overview of the human (*Hg19*) PITX2 locus, including the CTCF binding sites that maintain local chromatin conformation, the variant region near PITX2 associated with AF and SND, and the familial deletion recapitulated in the *delB* mouse. ATAC-seq tracks generated using human iPSC-derived PCs[113] and LACMs[114] and RNA-seq of human iPSC-derived PCs[113] highlight differentially accessible DNA regions and thus, putative PC-specific regulatory elements. PC pacemaker cardiomyocyte, SND sinus node dysfunction, SAN sinus node, AF atrial fibrillation, SNP single nucleotide polymorphism, PV pulmonary vein, LA left atrium, RA right atrium, TAD topologically associated domain, CM cardiomyocyte.

species scavengers[40,43,96]. Conversely, oxidative stress in *Pitx2*-haploinsufficient mice promotes pro-arrhythmic remodeling and AF[97]. In *delB* PCs and AAV6-*PITX2c* PCs, we observed altered expression of metabolic and mitochondrial genes that align with those previously found in atrial CMs, suggesting that, also in PCs, PITX2 regulates these gene programs while they obtain a more atrial working CM-like state.

PITX2 has been found to modulate a *Tbx5*-dependent gene regulatory network that maintains atrial rhythm[38]. In this mechanism, TBX5 activates *Pitx2*, and PITX2 suppresses expression of TBX5-induced ion handling genes, including *Scn5a*, *Gja1*, and *Ryr2*. In contrast, in *delB* PCs and AAV6-*PITX2c* PCs, these genes are upregulated, indicating that the regulatory function of PITX2 is context-dependent. Indeed, in different CM cell states identified in adult left atrial and pulmonary vein myocardium, PITX2 was predicted to cooperate with different CM cell state-enriched TFs and co-factors to regulate CM cell state-enriched gene expression[34]. This is consistent with our findings that PITX2 expression in PCs causes unique transcriptome changes (Figs. 3f, 4b and Supplementary Fig. 4d, e).

The downstream effects of changes in TF dosage are often non-linear[98] and exhibit tissue-specific threshold behavior. For example,

while the lungs require high *Pitx2c* levels for normal morphogenesis, lower *Pitx2c* levels are sufficient to establish important aspects of left-atrial identity[22,23]. We found that already at low levels of PITX2 expression, *delB* PCs reduce ISL1 expression and change their state, indicating that PCs have a low tolerance for PITX2 expression with large consequences for gene regulation. Yet, in *wild-type* mice, sporadic PCs express PITX2, and we noted a PITX2+ domain adjacent to the SAN head expressing PITX2 as well, suggesting that some PITX2 is tolerated, or even required. Heterozygous loss of function of *Pitx2* has been associated with affected SAN function[10,38], an observation that has remained poorly understood in the context of the presumption that PITX2 is normally absent from the SAN and RA, but that may relate to expression of PITX2 in sporadic PCs and parts of the developing SAN we observed.

Previously, we have shown that the deletion of CTCF sites approximately 1 Mbp distant from *PITX2* alters the topology of the entire locus (TAD fusion), causing altered interaction frequencies and durations between the *PITX2* promoter and regulatory sequences, including those driving the lncRNA *PLAYRR*, a suppressor of *Pitx2* in the developing gut and possibly in the SAN[50,99,100] (Fig. 8c). Together, these

data suggest the presence of a latently active regulatory network that drives *PITX2* expression in PCs, which is actively repressed to prevent *PITX2* expression in PCs. This stochastic activation-repression system may be unstable and "leak" and would be vulnerable to genetic variation that affects the function or interactions of regulatory DNA elements involved in this activation-repression system (Fig. 8c). We hypothesize that this vulnerability to deregulation and the low tolerance of PCs for PITX2 explain the variability in phenotype of heterozygous deletion carriers, as the presence of variants in deletion carriers will additionally modulate the level of ectopic *PITX2* expression. The higher the average *PITX2* expression level, the larger the proportion of HCN4[low] PCs over functional PCs, the more likely the threshold for SND is reached and the more severe the SND will be. Likewise, we hypothesize that AF/SND SNPs that decrease or elevate *PITX2* expression in LA/pulmonary vein CMs, may also derepress *PITX2* expression in PCs, predisposing to SND (Fig. 8b). SND and AF have been well established to frequently co-exist, and AF can contribute to SND[51,52]. Although PITX2 is not expressed in the *delB* RA, the adult *delB/delB* RA showed transcriptional changes consistent with a cardiac stress response. Prolonged, severe SND in *delB/delB* mice resulting in chronic, irregular atrial activation may promote a pro-arrhythmogenic state of RA tissue and increase susceptibility to AF (Fig. 8c).

We note a few limitations of our study design. The spatial transcriptomics analysis was limited to the transcripts provided by the probe atlas, which excludes non-coding RNAs like *Playrr* and splice variants, etc., and was of low resolution, requiring ROIs of >50 nuclei. The hiPSC PC model yielded <60% PC-like cells and involved *PITX2* overexpression using AAV-mediated transduction, which was therefore not well-controlled in terms of level. However, the scRNA-seq data suggested that a large fraction of the PC cells expressed PITX2 at levels within a physiologically relevant range. The model could be refined using inducible expression methods, thereby controlling level and enabling assessment of reversibility of PITX2-induced state changes, or introduce the *delB* deletion in iPSC lines followed by PC differentiation.

In summary, we have shown that the ectopic expression of *Pitx2* in SAN PCs severely disrupts the PC gene program in a PITX2 dosage-dependent manner, resulting in SND and atrial arrhythmogenesis, providing a mechanistic link between *PITX2* and SND and AF.

## Methods

### Transgenic mice
*DelB* mice (FVB/NJ background (Janvier Labs)) were previously generated in the Amsterdam UMC transgenic mouse core facility using CRISPR/Cas9[50]. The region deleted in *delB* mice is as follows: *mm10*: chr3:128,329,001-128,346,349. Animals were housed under a light regime from 07:00 to 19:00 (60 lux) at 20–24 °C and relative humidity of 45-66%.

### In vivo electrophysiology
Following anesthetization using 5% isoflurane (Pharmachemie B.V. 061756), both male and female adult mice ranging between 12 and 33 weeks of age were placed on a thermostated mat (37 °C). All mice received a steady flow of 1.5% isoflurane during all experiments. Electrodes connected to an electrocardiogram (ECG) amplifier (Powerlab 26T, AD Instruments) were inserted subcutaneously in the limbs and the ECG was measured for 5 min. The last 4 min of the 5-minute ECG were screened for any atypical cardiac events, and the last 60 s were used to determine ECG parameters (Lead I, L-R; RR-, PR- and QT-intervals, P-wave and QRS-durations). In adult mice, an octopolar CIB'er (NuMED) was inserted through the esophagus to capture and pace the atria. All pacing protocols were performed at 1.5x the atrial capture threshold, which was determined separately for each mouse. SNRT was evaluated using a 4 s pacing train (cycle length at 120 msec) and calculated as the interval between the last pacing stimulus and initiation of the first spontaneous P-wave. SNRT was normalized to resting heart rate (cSNRT=SNRT-RR interval) to account for inter-individual variations in sinus rate. The WBCL was evaluated using a 4 s pacing train starting at 100 msec, decreasing the pacing interval by 2 msec each train until the first atrioventricular block was detected. The susceptibility to AA was evaluated by 1 or 2 s pacing bursts starting with a cycle length of 60 msec, decreasing the pacing interval by 2 msec down to a cycle length of 10 msec. The duration of AA episodes is the sum of time each animal experienced an AA episode >1 s (2 passes). In a subset of adult mice, a combined beta-adrenergic and muscarinic cholinergic blockade, blocking the autonomic nervous system (ANS block), was achieved by intraperitoneal injection of 0.5 mg/kg atropine and 1 mg/kg propranolol as previously described[101]. ECGs and pacing protocols described above were repeated 10 min after the ANS block was administered.

### Embryo isolation
Following a timed pregnancy protocol, pregnant mice were anesthetized for 5 min using 4% isoflurane (Pharmachemie B.V. 061756) and sacrificed by cervical dislocation. Embryos younger than embryonic day (E) 14.5 were isolated in cold PBS, and the yolk sac was collected for genotyping. Fetuses older than E14.5 were isolated in cold PBS, decapitated and the thorax was isolated for sufficient fixation. Tail samples were collected for genotyping by PCR. The embryos and fetuses were fixed overnight in 4% paraformaldehyde and embedded in paraplast.

### Immunohistochemistry
Paraplast-embedded fixed tissues were sectioned at 5 or 7 μm in series. Following removal of the embedding media, epitopes were unmasked by boiling for 5 min in a high-pressure cooking pan in Antigen Unmasking Solution (Vector H3300-250). Tissue sections were incubated with the primary antibody overnight in 4% BSA in PBS. The antibodies used are as follows: rabbit-anti-HCN4 (Merck Millipore AB5805), sheep-anti-PITX2 (Bio-Techne AF7388), rabbit-anti-ISL1 (GeneTex GTX102807), goat-anti-TBX3 (Santa Cruz Biotechnology SC 31656), mouse-anti-SHOX2 (Abcam AB55740), goat-anti-CX40 (Santa Cruz Biotechnology SC 20466), goat-anti-NKX2-5(LabNed LN2027081), goat-anti-TNNI3 (Hytest 4T21/2), mouse-anti-TNNI3 (Merck Millipore MAB1691), chicken-anti-GFP (AvesLabs GFP1020), rabbit-anti-PCM-1 (Atlas Antibodies/Bio-connect (HPA023370)), Alexa Fluor 488 donkey anti-mouse IgG (Invitrogen), Alexa Fluor 488 donkey anti-goat IgG (Invitrogen), Fluor 555 donkey anti-mouse IgG (Invitrogen), Fluor 555 donkey anti-goat IgG (Invitrogen), Fluor 555 donkey anti-rabbit IgG (Invitrogen), Fluor 555 donkey anti-sheep IgG (Invitrogen), Fluor 647 donkey anti-rabbit IgG (Invitrogen), Fluor 647 donkey anti-sheep IgG (Invitrogen), DAPI (Sigma, D9542). Sections were mounted in PBS/Glycerol 1:1 and stored at 4 °C. Images were acquired using a Leica DM6000, TCA SP8 X confocal and Stellaris 5 confocal microscope.

### In situ hybridization
Tissues were fixed overnight in 4% paraformaldehyde, embedded in paraplast and sectioned at 7 μm. cDNA probes for *Hcn4*, *Gja5*, and *Pitx2c* were used and in situ hybridization was performed as previously described[53,102]. Images were acquired with a Leica DM5000 microscope. 3D volume reconstructions of the *Hcn4*+ expression domain in the RA at E17.5 *wild-type* (*n* = 8) and *delB/delB* (*n*= 5) fetuses were generated using Amira 3D 2021.2.

### Image analysis
The relative expression level of PITX2 and ISL1 in individual nuclei was quantified by immunostaining equivalent sections containing *wild-type* (*n* = 2, 239 nuclei total), *delB/+* (*n* = 3, 201 nuclei total) and *delB/delB* (*n* = 3, 280 nuclei total) for DNA (nuclei, DAPI), PITX2 and ISL1. Nuclei in the SAN head domain were segmented using ImageJ 1.54 f

using StarDist 2D 0.3.0, and the mean gray value was determined per nucleus per channel to quantify the relative expression of PITX2 and ISL1[103–105]. The number of PITX2+ nuclei were quantified by immunostaining equivalent sections containing the *wild*-type (*n* = 5), *delB/+* (*n* = 6), and *delB/delB* (*n* = 6) fetal SAN head domain for DNA (nuclei, DAPI), HCN4 and PITX2 expression. Images were then acquired using the Stellaris 5 confocal microscope. Nuclei in the SAN head domain were again segmented using ImageJ 1.54 f using StarDist 2D 0.3.0 and assigned to either HCN4[high] or HCN4[low] subdomains based on HCN4 expression. The mean gray value per nucleus was then determined per nucleus per region per channel.

## Spatial transcriptomics

Formalin-fixed paraffin-embedded *wild*-type, *delB/+*, and *delB/delB* E17.5 fetuses and adult mice were sectioned at 5 μm in series and screened by immunohistochemistry using a rabbit-anti-HCN4 (Merck Millipore AB5805) and goat-anti-TNNI3 (Hytest 4T21/2) antibody to identify the SAN head domain. The samples were prepared for the Nanostring GeoMx Digital Spatial Profiler according to the GeoMx Manual Slide Preparation User Manual (MAN-10150-3) using anti-HCN4, anti-cTNNI3 and Sytox Green Nucleic Acid Stain (Thermo Fisher Scientific S7020) and the GeoMx Whole Transcriptome Atlas Mouse RNA Probes for NGS (Bruker Spatial Biology). ROIs were segmented using HCN4 signal intensity thresholds that clearly demarcated the E17.5 HCN4+ SAN domain from the RA and the E17.5 HCN4[high] *delB* subdomain from the HCN4[low] *delB* SAN subdomains. ROIs were segmented based on TNNI3 signal intensity in the adult heart. The size and the number of nuclei per ROI are listed in Supplementary Data 14.

## Spatial transcriptomics data analysis

FASTQ files were trimmed, stitched, aligned and deduplicated using the Nanostring GeoMx NGS pipline (version 2.3.3.10). The generated DCC files were processed with the R GeoMxTools package (version 3.8.0) using the configuration file, the sample sheet and the probe metadata file (v1). The percentage of reads trimmed, stitched, aligned and deduplicated was calculated for each segment, all of which passed the thresholds of 80, 80, 75 and 50 percent. Approximately 6000 genes were detected in the smallest segment, which contained 17 nuclei. No segments were excluded after the segment quality control. Low-performing probes were removed based on the limit of quantification, defined as two standard deviations above the geometric mean of the negative control probe counts (210 negative control probes are included in the NanoString GeoMx Mouse Whole Transcriptome Atlas). Normalization was performed using quantile-normalization. Differential gene expression analysis was performed using limma (version 3.60.1). Multiple testing correction was performed using the Benjamini-Hochberg method. Differentially expressed genes were defined as an adjusted $p < 0.05$ and a $\log_2$fold-change (L2FC) > 0.25.

## Adeno-associated virus type 6 (AAV6) preparation

AAV6 were produced in HEK293T cells. Briefly, low passage HEK293T cells were seeded in 145 mm dishes at a density of $1.5 \times 10^7$ cells per dish in DMEM-GlutaMax (ThermoFisher Scientific 31966-047) supplemented with 10% fetal bovine serum (FBS) (Sigma F7524) and 1% penicillin-streptomycin (P/S) (ThermoFisher Scientific 15140-122). Cells were transfected with 15 μg of the AAV transfer plasmid (pAAV-*cTnT-PITX2c-p2a-H2B-EGFP* or pAAV-*cTnT-mCherry-p2a-H2B-EGFP*) and 33 μg AAV6 helper plasmids (pDP6) using linear polyethylenimine(PEI)-Max (Polysciences Inc 24765-1) at a ratio of PEI:DNA ratio of 2:1. After 24 h, medium was refreshed with Opti-MEM medium (ThermoFisher Scientific 11058-021) and the cells were incubated for 48 h at 37 °C. AAV6 in the medium was obtained by collecting and filtering the medium through a 0.22 μm pore sized PVDF syringe filter, and then concentrated using Amicon™ Ultra-15 Centrifugal Filter Units (Millipore/Merck, #UFC910024), according to the manufacturer's instructions. Genomic titre was determined by qPCR.

## Adeno-associated virus type 9-Myo4A (AAV9-Myo4A) preparation

AAV9 vectors were produced by transfecting low passage HEK293T (ATCC CRL-3216) cells, which were seeded a day prior at a density of $1.5 \times 10^7$ cells per dish in DMEM-GlutaMax supplemented with 10% FBS and 1% P/S. The cells were transfected with 13 μg Rep2Cap9-Myo4A plasmid[106], 21 μg AdenoHelper plasmid and 13 μ AAV transfer plasmid (AAV9Myo4A-*hTNNT2p-i-ACCuORF-hPITX2c-IRES-H2B-GFP-WPRE* or ssAAV9Myo4A-*hTNNT2p-mCherry-P2A-H2B-GFP-WPRE*) using PEI-Max at a PEI:DNA ratio of 2:1, during which the medium was replaced with Opti-MEM medium. The cells were collected 72 h after transfection, and the medium was concentrated by tangential flow filtration using the ÄKTA flux s system (GE Healthcare) to a final volume of 20 mL. Cells and medium were then combined, lysed by two freeze-thaw cycles, and followed by Benzonase (Merck 1016950001) treatment. AAV9 vectors were purified by iodixanol density-gradient ultracentrifugation, after which the AAV-containing fraction was collected and buffer buffer-exchanged and concentrated to PBS containing 0.001% Pluronic F68 (ThermoFisher Scientific 24040032) using Amicon Ultra-15 100 kDa centrifugal filter units (Millipore/Merck, #UFC910024). Concentrated AAV vectors were aliquoted and stored at −80 °C until use. Genomic titre was determined by qPCR.

## Flow cytometry to evaluate transduction efficiency

Cultures of hiPSC-derived PCs were transduced with AAV6 particles on day 18. Virus-containing medium was removed the next day, and 7 days post-transduction, the cells were prepared for flow cytometry. Cells were dissociated using 1x TrypLE Select (Thermo Fisher Scientific #12563011) and resuspended in a buffer containing 0.5 M EDTA (Thermo Fisher Scientific #15575020) and 10% BSA (Sigma-Aldrich, #A8022). Acquisition was performed on FACSSymphony A1 Cell Analyzer (Beckton Dickinson). Data were analyzed using FlowJo version 10.

## Single-cell RNA sequencing

Previously described hiPSC-derived PCs[73] (female, LUM-C0099iCTRL04; registered in the Human Pluripotent Stem Cell Registry (https://hpscreg.eu/cell-line/LUMCi004-A)) were thawed and seeded in matrigel-coated wells at a density of $10^6$ cells per well, and were transduced with AAV6 (*TNNT2* promoter-driven *PITX2c-p2a-H2B-EGFP* or *mCherry-p2a-H2B-EGFP*). Virus was removed from the cells the next day. 7 days post-transduction, they were dissociated using 1x TrypLE Select (Thermo Fisher Scientific #12563011). Single-cell RNA sequencing (scRNA-seq) was performed on the 10x Genomics platform using the Chromium Single Cell 5′ v2 Reagent Kits according to the manufacturer's instructions. Gene expression libraries were sequenced on the NovaSeqXPlus system (Illumina) with 150 bp paired-end reads. ScRNA-seq reads were mapped to the human genome (hg19 plus a custom added overexpression cassette) using STARsolo (version 2.7.11a) (https://github.com/alexdobin/STAR). The alignment was performed with default parameters unless otherwise specified. The non-default parameters used were: Configure Chemistry Options- Cv3; Matching the Cell Barcodes to the WhiteList- Multimatching to WL is allowed for CBs with N-bases (CellRanger 3, 1MM_multi_Nbase_pseu-docounts); Maximum to minimum ratio for UMI count- 10 for *mCherry* samples and 12.5 for *PITX2c* samples (based on Barcode Rank plots). The resulting BAM files were processed to extract raw counts for subsequent downstream analysis with Seurat[107]. Briefly, high-quality single cells were selected according to the following parameters: gene count >350, mRNA molecule count >500 and, as cardiomyocytes naturally exhibit increased mitochondrial gene expression, a more permissive threshold of <20% for mitochondrial gene count was

used[108]. Next, the SCTransform command was used for normalization, scaling and identification of variable features (nfeatures = 3000). Raw counts were first normalized using Normalize Data, and the data were split by sample. Each sample was then independently normalized with SCTransform. To ensure consistent gene representation, the dataset was restricted to the common gene set shared by all samples. 3000 highly variable features were then selected using SelectIntegrationFeatures, followed by PrepSCTIntegration and FindIntegrationAnchors to identify shared biological states across samples. These anchors were used to perform integration via IntegrateData, yielding a batch-corrected expression matrix. Dimensionality reduction was performed using PCA and UMAP with the top 10 principal components. The top 10 principal components were used to compute a nearest-neighbor graph (FindNeighbors). Clustering was then performed at multiple reductions (FindClusters with resolution = 0.4, 0.6, 0.8, 1.0, 1.4), and the resolution 0.4 clustering was used for downstream analyses. FeaturePlot, VlnPlot and DoHeatMap commands were used to visualize gene expression.

For each given cluster in the UMAP, the number of cells from the AAV9-*mCherry* and AAV9-*PITX2* samples and determined the relative contribution of each sample to a cluster, the ratio between the cell numbers normalized to the total cell number of each sample was calculated (Supplementary Fig. 6c). A Z-test[109] was used to determine whether the difference in cell contributions per sample were significant (Supplementary Fig. 6c).

## Cellular electrophysiology

Coverslips with hiPSC-PCs were put in a recording chamber on the stage of an inverted microscope (Nikon Diaphot, Melville, New York, NY, USA) and the cells were superfused with modified Tyrode's solution (36 ± 0.2 °C) containing (mM): NaCl 140, KCl.4, CaCl$_2$ 1.8, MgCl$_2$ 1.0, glucose 5.5, HEPES 5.0; pH 7.4 (adjusted with NaOH). A custom-made slider for two filter blocks was used to detect *EGFP*- and *mCherry*-positive cells. Action potentials (APs) were measured from single CTRL and PITX2 hiPSC-PCs using the amphotericin-perforated patch clamp methodology and an Axopatch 200B amplifier (Molecular Devices, Sunnyvale, CA, USA). Data acquisition and data analysis were performed with custom software. Patch pipettes were pulled from borosilicate glass (TW100F-3, World Precision Instruments Germany Gmb) using a custom vertical microelectrode puller and had tip resistances of 2.0–3.0 MΩ after filling with the pipette solution containing (in mM): K−gluconate 125, KCl 20, NaCl 10.0, amphotericin-B 0.44, HEPES 10, and pH 7.2 (KOH). All potentials were corrected for the estimated liquid junction potential[110]. Signals were low-pass filtered (cut-off frequency: 5 kHz) and digitized at 5 (spontaneous APs) and 40 (paced APs) kHz. The cell membrane capacitance was determined by dividing the decay time constant of the capacitive transient in response to hyperpolarizing voltage-clamp steps of 5 mV (from a holding potential of −40 mV) by the series resistance.

APs were measured without pacing as well as at 1 Hz pacing frequency with 3-ms, 1.2× threshold current pulses through the patch pipette. The latter was performed under the condition of an in silico injected 2 pA/pF Kir2.1 current using dynamic clamp[111] to anticipate the potential presence of quiescent cells, and potential differences in MDP and cycle length, which all will indirectly affect the availability of ion channels underlying cardiac AP[112]. In the presence of Kir2.1 injected currents, hiPSC-PCs all become quiescent, the MDP of hiPSC-PCs is between -80 and -85 mV, and APs can be evoked at a fixed 1 Hz pacing frequency. We analyzed the cycle length, MDP, maximum AP upstroke velocity ($V_{max}$), AP amplitude (APA), and APD at 20%, 50%, and 90% repolarization (APD$_{20}$, APD$_{50}$, and APD$_{90}$, respectively). Parameters from 10 consecutive APs were averaged.

## GO enrichment analysis

GO enrichment analysis was carried out using the Protein ANalysis THrough Evolutionary Relationships (PANTHER) Classification System 19.0.

## Statistical and reproducibility

Statistical analysis was performed using GraphPad Prism version 10 for Windows, GraphPad Software, San Diego, CA. The method used to test statistical significance and the number of biological replicates (*n*) are stated in the figure legends. Data are presented as means unless otherwise stated. No data were excluded, and the Investigators were blinded during experiments and analyses.

## Ethics statement

The guidelines from the Directive 2010/63/EU of the European Parliament and Dutch government were followed for all animal care, housing, husbandry and experiments. All animal experimental protocols were approved by the Animal Experimental Committee of the Amsterdam University Medical Centers and animal group sizes were based on previous experience.

## Reporting summary

Further information on research design is available in the Nature Portfolio Reporting Summary linked to this article.

## Data availability

The scRNA-seq data generated in this study have been deposited in the GEO database under the accession code GSE293813. Nanostring GeoMx-DSP datasets are available under accession codes GSE291746 (E17.5 *wild-type* and *delB/del*B SAN and RA), GSE291749 (E17.5 *wild-type*, *delB/+*, *delB/delB* SAN, RA, and LA), and GSE291752 (adult *wild-type*, *delB/+*, and *delB/delB* SAN and RA). Source data are provided with this paper.

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

## Acknowledgements

This work was supported by Dutch Research Council OCENW.G-ROOT.2019.029 (to V.M.C.); European Innovation Council Pathfinder Challenges Project 101115295 - Nav1.5-CARED and TRANSITION - Project 101099608 – TRACTION (to V.M.C. and G.J.J.B.); Prof. Dr. AF Moorman Fund of the Amsterdam University Fund (to B.J. and V.M.C.); Dutch Heart Foundation and Hartekind Foundation Grant CVON2019-002 OUT-REACH (to V.M.C.); Health Holland PPP-Grant 2024 CURE-VT/VF (to G.J.J.B. and V.M.C.); Out of the Box grant 2022 from the Amsterdam Cardiovascular Sciences Institute, Amsterdam, The Netherlands (to H.D.D. and V.M.C.). J.F.M. was supported by the National Institutes of Health (HL 169511, HL 171574, HL 118761, HL 177644, and HL 173242) and the Vivian L. Smith Foundation to J.F.M. J.D.S. was supported by NHLBI R00HL169742. M.R.R. was supported by the Dutch Heart Foundation and Dutch CardioVascular Alliance 01-002-2022-0118 EmbRACE. H.D.D. is supported by European Research Council Starting grant 101116688. We thank Fernanda M. Bosada (Amsterdam UMC), Jean-Jacques Schott (Nantes Université) and Julien Barc (Nantes Université) for their contributions during the initiation of these studies.

## Author contributions

Conceived/designed elements of the study: L.V.D.M., O.J.M., M.R.M., H.D. and V.M.C. All authors acquired, analyzed or interpreted data. The manuscript was drafted by L.V.D.M., M.R.M. and V.M.C. All authors critically revised the manuscript and approved the final version.

## Competing interests

G.J.J.B. reports ownership interest in PacingCure BV, and J.F.M. is a cofounder and owns shares in Medley Therapeutics. Other authors declare no competing interests.
