## [Transparent Peer Review file · Nature Communications]

PITX2 dosage-dependent changes in pacemaker cell state underlie sinus node dysfunction and atrial arrhythmias

Corresponding Author: Professor Vincent Christoffels

Version 0:

Reviewer comments:

Reviewer #1

(Remarks to the Author)

This manuscript builds on a previous study including some of the same authors published in Nature Communication in 2024 describing the effects of a deletion in the gene desert 1 Mbp upstream of PITX2 identified in patients and the development of a delB mouse model that recapitulates the patient phenotype of sinus node bradycardia and atrial fibrillation. Using the delB mouse model, the previous study revealed an ectopic upregulation of PITX2 in the right atria due to the deletion. Here the authors further investigated the effect of ectopic expression of PITX2 in the right atria and identify that the upregulation is confined to the pacemaker cells of the SAN. Using both the delB mouse model and induced human pluripotent stem cell derived SAN pacemaker cells they elegantly show how increased expression of PITX2 results in changes to the pacemaker phenotype of the cells in the SAN. Using delB/+ and delB/delB mice the authors were able to make conclusions on the correlation of PITX2 gene dosage with phenotype severity both on the expression level and functional level. This study contributes valuable new insights into the underlying mechanisms of sinus node dysfunction in patients with mutations in non-coding regions of the genome. However, the following questions should be addressed to further strengthen the manuscript before publication.

1) In the introduction the authors mention that reduced expression of PITX2 in the left atria has been shown to cause atrial fibrillation (AF) in several studies. However, they do not mention the levels of PITX2 expression in their delB mouse model. What is the expression of PITX2 in the RA and LA and ventricles of the delB mouse compared to wildtype. Could these changes explain the AF phenotype shown by the authors? Related to this question, and assuming that there is no changes in PITX2 expression in the atria of the delB mouse, how do the authors explain that the bradycardia due to ectopic PITX2 expression in the SAN is resulting in AF. For the general (not cardiovascular expert) audience of Nature Communications it would be helpful to add a more clear explanation of this in the discussion.

2) The model presented in Fig. 7c suggests upregulation of PITX2 in pacemaker cells and downregulation of PITX2 in the left atrium. Have both of these expression changes ever been shown in the same patient /animal model? It would be important to more clearly explain this in the discussion. Otherwise, a model focused on the delB mutation studied here might be more appropriate.

3) The manuscript text in lines 292-297 and Fig. 4g, h does not align. The text states that PC nuclei positive for ISL1, PITX2 or both were analyzed but the figures only show analysis of PITX2 positive nuclei. In addition, the authors make the statement that "the delB/delB HCN4^{high} subdomain contains a smaller fraction of total PC nuclei and the HCN4^{low} subdomain contains a larger fraction of total PC nuclei compared to their respective subdomains in delB/+ SANs". To support this claim statistical analysis between the HCN4^{high} and HCN4^{low} groups in the delB/delB vs delB/+ samples needs to be included.

4) The use of delB/+ and delB/delB vs wildtype mice and dissection of HCN4^{high} vs HCN4^{low} regions allowed the authors to make conclusions on gene expression changes based on PITX2 dosage. However, they never show a direct correlation of PITX2 expression levels to pacemaker gene expression changes. A graph detailing the actual expression levels of PITX2 and the corresponding gene expression changes would be a valuable addition to the manuscript. Such analysis might further help to define the expression threshold of PITX2 that needs to be reached to delineate pacemaker cell differentiation.

5) The MALAT1-based exclusion of clusters 4+8 is an unconventional practise not widely used in the field (manuscript text

lines 311-312 and Supplementary Figure 4c). In addition, the authors exclude cells based on high levels of MALAT1 expression while the cited publication recommends to exclude cells with 0/low MALAT1 expression. Could the authors please explain this discrepancy and show further gene expression of these clusters to confirm poor cell quality. For example, what is the mitochondrial gene count for these clusters? This is particularly important because based on Supplementary Fig. 4i the AAV6-PITX2 sample is skewed towards contribution to the MALAT1+ clusters. By excluding these clusters, the authors ignore a large fraction of AAV6-PITX2-derived cells. Why do AAV6-PITX2 cells contribute stronger to the MALAT1+ cell clusters? This should be discussed in the manuscript.

6) It is not clear how the authors go from the original UMAP with 9 cell clusters to the reduced UMAP with 5 cell clusters (Fig. 5b, Supplementary Fig. 4c, d). Removal of clusters 4 and 8 due to high MALAT1 expression would have resulted in 7 clusters. Where did the other 2 cell clusters go? Along the same lines it is not clear whether cluster 7 in Supplementary Fig. 4c and Supplementary Fig. 4d are the same cells. Was the data re-clustered after removal of clusters 4 and 8? Please provide a more clear explanation of this analysis step.

7) The claims that the authors make with regards to PITX2 expression and proportion of AAV6-PITX2 cells in the different clusters is not well supported with the current display figures. It would be helpful to add an expression UMAP of PITX2 similar to what is shown for ISL1 in Figure 5c. It would further be helpful to show a UMAP that displays the origin of each cell from either the PITX2 or mCherry experimental group in the main Figure 5 following figure 5b. In addition, Figure 5f is elegant but could be better explained / labelled in the graph to make it clear that this is the change in cell distribution between AAV6-mCherry and AAV6-PITX2 samples and does not display gene expression.

8) In the abstract the authors state: "Ectopic expression of PITX2c in human induced pluripotent stem cell-derived PCs resulted in dysfunction and PITX2 dosage dependent transcriptional changes reminiscent of those observed in the sinus node subdomains of *delB* mice." However, the presented data does not support the claim of PITX2 dose dependent transcriptional changes well. A more detailed analysis of how PITX2 expression levels relate to expression changes e.g. via a graph displaying PITX2 expression levels and corresponding gene expression changes, similar to the comment in point #4, would be better supporting this claim.

Along the same lines it would be interesting to directly compare changes in gene expression between clusters 0 and 7 in the AAV6-PITX2 samples to identify which expression changes define whether cells lose their PC phenotype and "fall" into cluster 7 or maintain the PC phenotype and "remain" in cluster 0. How does PITX2 expression differ between cells in these clusters – is it significantly different?

9) The presented data suggests that ectopic PITX2 expression in pacemaker cells results in upregulation of the atrial gene program based on a few genes (SCN5A, GJA1, GJA5, NKX2-5). How do the PITX2+HCN4^{low} cells compare to atrial and ventricular cardiomyocytes on the transcriptome level? Do they convert to atrial cardiomyocytes? Or do they take on a non-physiological intermediate phenotype? This question could be addressed using either the mouse or human induced pluripotent stem cell expression data and comparison to existing transcriptome data of atrial and ventricular cardiomyocytes.

10) Finally, the use of human induced pluripotent stem cells is an asset to this study allowing the validation of the observed mechanism in a human model system. To increase the impact and potential future clinical translation of the findings of this study it would be important to know whether inhibition of the ectopic PITX2 expression can rescue the phenotype and re-establish the pacemaker phenotype in the AAV6-PITX2 SAN pacemaker cells.

Minor:

1) In line 167-186 the authors state: "Immunostaining further revealed co-expression of ISL1 and PITX2 in the differentiating PCs of the E10.5 *delB/delB* SAN (Fig. 1b)". Fig 1b does not clearly show the co-expression. Please show separate fluorescent channels or a more clear overlay image with insets to better support this claim.

2) In Supplementary Fig. 4b the labels of the flow plots appear to be switched. It is also misleading to label the upper right quadrant in the 2nd plot with 93% while it has almost no cells. I recommend adding the enlarged label in the appropriate quadrant.

3) In Fig. 5e, it is not clear what the purple color code indicates – what is the legend? In addition, genes seem randomly highlighted in bold. It would be more clear if the authors highlight all the genes mentioned in the text in bold or explain the reasoning as to why some genes are bolded and others are not.

4) Similarly, in Fig. 5h it is not clear what the purple color code indicates.

5) The authors are referring to the PITX2 transfected cells as Pitx2+ PCs which gets confusing when talking about the single cell RNA-seq data because it is not clear whether the authors refer to all the cells expressing PITX2 or to the experimental group of cells that was transfected with PITX2 AAV6. To make this more clear to the reader I suggest referring to the PITX2 overexpressing cells as AAV6-PITX2 cells consistently.

6) In Supplementary Fig. 4g the GO terms seem to be sorted from lowest fold change to highest fold change based on the color code. This is counterintuitive and similar figures in the manuscript seems to be sorted from highest to lowest fold change as one would expect.

7) In line 325 the figure reference is probably supposed to state “Supplementary Fig. 4i, j” instead of “Fig 4i, j”.

8) In the legend of Fig. 7 the labels for panel c and b are switched.

Reviewer #2

(Remarks to the Author)

Overview

This data-rich manuscript builds on a prior report of a novel Mendelian arrhythmia syndrome causing sinus node dysfunction (SND) and atrial fibrillation (AF). In these families, deletion of a noncoding region containing a CTCF binding site disrupts a TAD boundary at the *Pitx2* locus, resulting in ectopic expression of *Pitx2* in the SAN, presumably causing premature sinus node dysfunction and atrial fibrillation. In the present work, the mechanistic basis for these phenotypes is investigated further through an examination of SAN development, spatial organization of SAN gene expression, and heart rhythm in a mouse model of the human deletion syndrome (‘delB’ mice). In addition, the consequences of *Pitx2* misexpression are explored in an SAN iPSC model system. Although the mouse model was generated and characterized in the prior study, the present work provides substantial additional mechanistic data on the phenotypes of these mice and introduces spatial transcriptomics to characterize distinct regions within the SAN.

The main findings are (1) a few *Pitx2*-expressing cells are present in or near the WT SAN, (2) misexpression of *Pitx2* in the SAN primordium of delB mice is present at the onset of SAN morphogenesis and results in downregulation of key SAN transcription factors; (3) delB/delB embryonic SAN exhibits patchy expression of *Hcn4* with *Hcn4*^{low} and *Hcn4*^{high} areas that can be distinguished using spatial transcriptomics; (4) *Pitx2* dose-dependently represses *Isl1* and SAN gene expression in mouse and hiPSC SAN cultures with corresponding changes in SAN electrophysiology in vivo and in vitro. Based on these findings, the authors conclude that some rare *Pitx2* expressing cells in the SAN and perinodal area may be necessary for normal heart rhythm, that pacemaker cells are sensitive to low levels of *Pitx2* expression, but that once *Pitx2* expression exceeds a threshold in a given PC, pacemaker gene expression essentially collapses with reprogramming to a different non-PC-like cell state. These findings may explain the clinical phenotype and variable expressivity observed in patients with the deletion syndrome and could have implications for the impact of common genetic variation at this locus on heart rhythm.

Overall, the quality of the raw data and data analysis is uniformly high. New and rich datasets are offered to the field. Mouse physiology studies are, if anything, overpowered and these data are very convincing. In general, the data support the conclusions although, as detailed below, there may be alternative explanations for some key findings that are not thoroughly considered but could have a major effect on the interpretation and explanatory model. In terms of conceptual novelty, it is already well established that *Pitx2* represses SAN gene expression so in that sense these findings fit well with existing literature and can be seen as incremental. However, if a threshold effect of *Pitx2* could be more convincingly demonstrated, this could offer novel insights on transcription factor dose-dependence in the SAN.

Specific Comments:

1. Line 173. The authors define a “dorsal SAN head domain” that is *Pitx2*⁺, *Hcn4*^{-low}, *Isl1*^{-negative}. This profile resembles the adjacent SVC/sinus venosus myocardium. From the images in Supplementary Figure 1C that are intended to demonstrate this anatomical structure, it’s not clear what would justify ascribing a new transcriptional domain to the SAN head versus just a region where there is some interdigitation of SAN PCs with SVC myocardium.

2. Figure 2e proposes that *Pitx2* gradually “reprograms” PCs by repressing *Isl1/Tbx3/Shox2*, resulting in de-repression of *Nkx2-5* and eventually upregulation of *Gja1*, downregulation of *Hcn4*, and further reduction in *Isl1/Shox2/Tbx3*. This model is certainly plausible and could explain the observations. However, can the authors exclude another scenario wherein *Pitx2* misexpression either blocks specification of SAN head cells, or leads to SAN hypoplasia, allowing the expansion of other nearby cell types into the SAN (SVC myocardium, SAN transitional cells, *Osr1*⁺ DMP-derived tissue), accounting for some additional cell type heterogeneity in SAN sections? Because PCs are not irreversibly labeled prior to the onset of *Pitx2* misexpression, it’s hard to rule out this other scenario from the data presented. Because such an experiment is not currently feasible, do the authors have any data on proliferation of the SAN head and/or nearby tissue compartments in delB versus WT mice after the onset of *Pitx2* misexpression?

3. Related to the previous comment, could the alternative scenario also partly explain the apparently patchy nature of different expression domains (clustering of HCN4-LOW cells into domains) despite the stochastic nature of *Pitx2* misexpression level within the SAN which does not seem to show the same degree of spatial clustering (e.g. Figure 4a)? This would be an alternative to postulating a “threshold response” to explain the data in Figure 4D, where delB/+ and delB/delB cluster by cell state (HCN4-low vs HCN4-high) rather than genotype. Because other embryonic cell types are not represented on the PCA (e.g., DMP derivatives, non-SAN head PCs), it’s hard to know whether the HCN4-LOW regions might represent expansion of these cell types beyond their usual borders when the SAN head is smaller. In figure 4c, for example, the HCN4LOW region appears non-uniform with a predominantly *Hcn4*^{-negative} region interdigitating with a few *Hcn4*^{-high} cells. Put differently, could the focus on identification and characterization of gene expression domains be obscuring a broader morphogenetic defect in the dorsal atrium and can the authors comment on whether SAN, RA, SVC/sinus venosus, and venous valve/DMP are different in the delB genotype?

4. Related to the above comments, since the hiPSC-PC *Pitx2* overexpression system revealed a wide range of *Pitx2* overexpression levels, was a threshold effect observed in terms of downstream gene expression or cell state that paralleled the dichotomization observed in the mice into *Hcn4*-HIGH and *Hcn4*-LOW regions? If not, would these human data then

militate against the threshold model?

5. How does the level of pitx2 seen in the delB SAN compared to WT LA?

6. Since some C7 cells were observed in the AAV6-mCherry hiPSC cells, can the authors assign a cell type to this cluster? Could C7 represent SVC/sinus venosus cells?

Minor:

8. Figure 5m – dV/dt max not Vmax

Reviewer #3

(Remarks to the Author)

Summary

In this manuscript van der Mareel et al., investigate the phenotype of the 'delB' mouse, with a 15kb deletion of an intergenic region upstream of Pitx2. Previous work from the lab (Baudic et al.) had found that delB mice show an increased risk of atrial arrhythmias, mimicking the phenotype observed in 7 human families with deletions within the paralogous region. Furthermore, they found increased expression of Pitx2 in delB mice, particularly within the SAN, which was proposed to be downstream of changes in chromatin conformation.

In this work, the authors further characterize the delB phenotype, finding that heterozygous and homozygous delB embryonic hearts show stepwise increases in the number of Pitx2 expressing cells within the SAN. Downstream of this misexpression of Pitx2, SAN cells show transcriptional changes, downregulating the 'pacemaker program' (including TFs such as ISL1 and Shox2, and key channel HCN4), while upregulating an atrial cardiomyocyte-like program (increasing NKX2.5, and gap junction proteins). The authors also generated human iPSC-derived pacemaker cells with PITX2C overexpression, and found that they undergo transcriptional changes recapitulating the observations in the delB mouse, as well as showing electrophysiological changes indicative of pacemaker dysfunction. The manuscript combines a number of models to provide convincing evidence that increased Pitx2 expression (due to either altered transcriptional regulation or transduction), alters the cell state of pacemaker cells causing a shift towards a more CM-like state with accompanying functional changes.

Please find specific comments below:

It would be interesting to understand more about how Pitx2 is interconnected with Gata6 expression in regulating SAN development and function. The authors discuss how Gata6 +/- mice also show similar mixed domains of HCN4 positive and negative cells and how both delB and AAV6-PITX2c-hiPSC-PCs show decreased Gata6 expression. Gharibeh, L. et al. also found that Pitx2 is upregulated in Gata6 +/- mice. Would increased Gata6 expression rescue the phenotype seen in the delB or AAV6-PITX2c model? On a more minor note, in Figure 2E Gata6 (and TBX5) is shown without yet any discussion or staining, perhaps this could be incorporated into the final model, or an additional staining could be performed- this would also validate the transcriptional changes seen at the protein level to observe the reduction in Gata6.

For the immunofluorescence or in situ panels show in figures 1 and 2 how many hearts were examined?

In figure 2C the HCN4 low and high cells seem to form distinct domains with the HCN4 low domain somewhat consistently towards the outer aspect of the SAN - is this representative and could the authors comments on why this pattern might be occurring if so? In contrast to the sporadic distribution of Pitx2+ cells described for the WT SAN, is the spatial distribution of the HCN4low cells (which presumably are also Pitx2 high) in the delB (heterozygotes and homozygotes) SAN regulated?

The panels showing changes in CX40 expression are perhaps not the clearest- the arrows help, but alongside the punctae indicated by the arrows there appears to be cytoplasmic/nuclear staining (in figure 2D too). Why might this differential localization be occurring?

In supplementary figure 1B for the ISL1 staining at E12.5 it is slightly difficult to see a clear decrease (also in comparison to the later stages examined, at E17.5), perhaps this could be quantified (number of positive cells or intensity) to make this clearer? This would also clarify the timecourse of this progressive loss of the 'PC program' during SAN development.

In figure 3D are these step wise changes in gene expression also significant (beyond Pitx2)?

In figure 3H there are grey dots and purple dots, is the grey representing genes below a cut off (and what would this cut off be)?

In figure 3G what statistical test is being used and do the bars represent average values? Is only NKX2.5 significant? Other TFs and ion handling genes appear to have larger fold changes and also given from the normalized count Pitx2 at least did seem to show a significant difference? Is there some underlying variability? There is also a switch from saying WT SAN to WT HCN4+, but does this represent the same population or a subpopulation of the WT SAN cells, i.e. was just the HCN4+ population considered?

In supplementary figure 2C the panels on the left are not described (and seems to be possibly be a duplicate of figure 2D)?

In supplementary figure 3C the staining for HCN4 is quite difficult to see, could a different LUT be used, or the LUT inverted? Arrows or a box might also help in distinguishing the positive and negative signal.

In order to examine whether increased Pitx2 expression also impacts human pacemaker cells, AAV6-PITX2c-hiPSC-PCs is used, is the level of Pitx2 expression comparable to the hiPSC 15kb $-/-$ model? Is this a physiologically/pathologically relevant level of altered expression?

In supplementary figure 4B are the labels flipped over? It seems that the PITX2 cells are mCherry positive and the mCherry cells are negative?

In figure 5E a comparison is made between cluster 0 in the human data and the mouse delB HCN4 low population, with many genes showing no correlation, how comparable are these systems given the differences between them (15kb deletion vs pitx2 OE, in vitro vs in vivo)? Later in 5H when comparisons are made within each system and then presented side by side, the broader conservation of transcriptional changes seems more apparent. Also, in supplementary figure 4I there are fewer PITX2c cells and it is also mentioned that there are more many more PITX2 cells in cluster 8, do these cells show proliferation defects or increased cell death?

In the model shown in 7B are the RNA/ATAC seq tracks from mouse or human? And from which model or the WT?

Finally, in the manuscript there is discussion about a shift in the metabolism in both of these models of increased PITX2 expression, but would it be possible to examine this further? Perhaps by staining for mitochondria or looking at the oxygen consumption of the hiPSCs?

Reviewer #4

(Remarks to the Author)

Here the authors present a study evaluating the effect of dosage-dependent changes in PITX2 expression in the sinus node and atrial arrhythmias. The overall topic is important and the use of complementary models is a strength. Please see below specific comments.

1. The abstract should be revised to provide a more concise and clearer message with an easier structure to follow, including the broad relevance of the topic.
2. The introduction would also benefit from including a definition of AF for the non-expert and explain why it is important to study it – what are the gaps in that area.
3. Line 208: cell numbers per ROI should be shown in the supplementary data section
4. How was annotation of spatial transcriptomic slides undertaken? The choice for markers used to define the ROI (HCN4 and TNNI3) should be justified. What about for example absence of Cx43/Gja1? What is the range of SIZE of ROIs in wT and mutant mice? How were differences between SAN HCN4^{high/low} determined consistently across samples? The second image in Fig 3a has only part of RA annotated as RA. Why is this?
5. Fig 3cP: Plotting logFC rather than counts normalised to wildtype SAN would be better to show the differences.
6. Fig 5c: How was the reduction in ISL1 in PC cluster quantified?
7. Fig5f: Please clarify what is plotted here. How was the change in proportion of cells quantified? Was this statistically significant? Unless the method used depends on manifold embedding (e.g. Milo) plotting on a UMAP is confusing.
8. Line 318: cells with high expression of MALAT1 expression were excluded. How was the correlation between MALAT1 expression and quality ascertained? What other quality metrics were used to have a full picture of the quality of the cells?
9. How were the scRNAseq data integrated and clustered?
10. The discussion should include limitations of the technologies used.

Version 1:

Reviewer comments:

Reviewer #1

(Remarks to the Author)

The additional data and clarifications that have been incorporated during the revision greatly enhance the manuscript.

I only have 1 outstanding question related to my original comment 1. I appreciate the detailed response to this point. The authors state that they included data for the expression of Pitx2c in wildtype vs delB/delB mice for the LA and RA. While Fig. 3c shows that there is no change in Pitx2c expression in the RA I am struggling to find the data showing Pitx2c expression in the delB/delB mice in the LA. Fig 4a and 4c only include expression levels in the wildtype LA but not in delB/delB LA. To convincingly show that Pitx2c expression is unchanged in both the right and left atria and therefore not a cause of the observed atrial fibrillation it would be important to include this data.

Reviewer #2

(Remarks to the Author)

I appreciate the detailed responses to the specific points raised in my original review, including some additional experimental evidence in favor of Pitx2-mediated reprogramming. The revised manuscript is improved. I don't have any additional comments.

Reviewer #3

(Remarks to the Author)

The authors have carefully addressed all of my queries, and those of the other reviewers.

Reviewer #4

(Remarks to the Author)

The authors have improved the introduction which reads well and added a discussion on the limitations.

The abstract reads better except for a couple of sentences

1. Line 22 'across a broad range of levels': please clarify
2. Lines 23-24, the sentence is long and could be improved

While authors include a lay explanation of AF the text of the introduction section does not convey what the unmet clinical needs in the realm of AF are and how this work aims to address them. This undermines the importance of findings in this paper.

Fig3c and d: These panels don't add new information compared to plots in Supplementary Fig 4 d and e. In fact, they are confusing as no statistical significance is shown.

The methods used to analyse spatial transcriptomic data are explained better, but it remains unclear how HCN4^{high/low} domains were annotated. Although it is indicated that this is based on a signal threshold, how is this threshold defined?

Fig 5c From the data presented here it remains unclear if there were any statistically significant cell type abundance differences.

The authors respond to the queries about single cell RNA seq analysis pipelines and about quality control metrics used for cell filtering. However, this is not reflected in the methods section of the manuscript.

Statistical analysis and the n numbers are missing in several panels including data showing tissue staining (E.g. figure 1 and 2). Some experiments were done with n=2, quantification and statistical analysis are needed across all experiments.

Version 2:

Reviewer comments:

Reviewer #1

(Remarks to the Author)

The authors have now sufficiently addressed all my comments.

Revisions NCOMMS-25-26166

Response to reviewers' comments:

We thank the Editor and the Reviewers for their constructive comments and for the opportunity to submit a revised manuscript. We have addressed the editorial and reviewer points as detailed below.

Reviewer #1 (Remarks to the Author):

This manuscript builds on a previous study including some of the same authors published in Nature Communication in 2024 describing the effects of a deletion in the gene desert 1 Mbp upstream of PITX2 identified in patients and the development of a *delB* mouse model that recapitulates the patient phenotype of sinus node bradycardia and atrial fibrillation. Using the *delB* mouse model, the previous study revealed an ectopic upregulation of PITX2 in the right atria due to the deletion.

Here the authors further investigated the effect of ectopic expression of PITX2 in the right atria and identify that the upregulation is confined to the pacemaker cells of the SAN. Using both the *delB* mouse model and induced human pluripotent stem cell derived SAN pacemaker cells they elegantly show how increased expression of PITX2 results in changes to the pacemaker phenotype of the cells in the SAN. Using *delB/+* and *delB/delB* mice the authors were able to make conclusions on the correlation of PITX2 gene dosage with phenotype severity both on the expression level and functional level. This study contributes valuable new insights into the underlying mechanisms of sinus node dysfunction in patients with mutations in non-coding regions of the genome. However, the following questions should be addressed to further strengthen the manuscript before publication.

Response:

We thank the reviewer for their appreciation of our study and for the constructive comments, which have been addressed below.

1) In the introduction the authors mention that reduced expression of PITX2 in the left atria has been shown to cause atrial fibrillation (AF) in several studies. However, they do not mention the levels of PITX2 expression in their *delB* mouse model. What is the expression of PITX2 in the RA and LA and ventricles of the *delB* mouse compared to wildtype. Could these changes explain the AF phenotype shown by the authors? Related to this question, and assuming that there is no changes in PITX2 expression in the atria of the *delB* mouse, how do the authors explain that the bradycardia due to ectopic PITX2 expression in the SAN is resulting in AF. For the general (not cardiovascular expert) audience of Nature Communications it would be helpful to add a more clear explanation of this in the discussion.

Response:

We thank the reviewer for bringing this to our attention. While highly relevant, in our previous study we did not directly compare PITX2 expression levels in the *delB* SAN with that in the 'native' PITX2-expressing LA or ventricles. We have performed additional experiments to address this comment as detailed below. Furthermore, we have modified the introduction to clarify that normally, PITX2 expression is absent in the RA and SAN. While we have previously shown a reduction in ventricular *Pitx2* expression in *delB/delB* mice at P21 (PMID: 38643172), the absence of a ventricular arrhythmogenic phenotype (Supplementary Tables 1 and 2) indicates that any functional consequences thereof are minimal.

To address the point of the reviewer, we have generated new data on the expression levels of *Pitx2* in the LA and SAN of *delB* mice. We first quantified PITX2 immunofluorescence signal in the *delB* SAN, LA body and LA auricle and found that PITX2 expression levels in the *delB* SAN are intermediate to those found in the LA body and auricle (revised Fig.

4a). Including the LA body and auricle in our spatial transcriptomics analysis showed that *Pitx2* expression is unchanged in the LA of *delB* mice, and further revealed that there is minimal overlap in the transcriptomes of LA-CMs and *delB* PCs (revised Fig. 4b and c), indicating that PITX2 expression in PCs does not induce a LACM state. The PCA plots comparing PC transcriptomes and RA or LA transcriptomes (revised Fig. 3b and 4b) and cluster analysis (revised Fig. 3f) also indicate that while *delB* PCs acquire a state more similar to atrial CMs, they also acquire unique and functionally relevant features (ex. increased *SCN5A*/Nav1.5-driven sodium current). We have also included a comparison of the transcriptomic changes in the *delB/delB* HCN4^{low} subdomain and the transcriptional differences between atrial CMs and PCs using the Goodyer et al 2019 scRNA-seq dataset (PMID: 31284824) to show that PITX2+ PCs do acquire some aspects of atrial CM state (Supplementary Fig. 4f). However, comparing these transcriptomic changes with those found between *Pitx2* haploinsufficient LACMs and *wild-type* LACMs (Steimle et al., 2022; PMID: 35471998) suggests that there is little overlap in the response to changes in PITX2 dosage in the SAN and LA (Supplementary Fig. 4g).

As our spatial transcriptomic datasets show that the deregulation of *Pitx2* expression in *delB* atria is restricted to the SAN, the AF phenotype is probably not the result of altered PITX2 expression in the LA/RA (revised Fig. 3 and Fig.4). To gain more insight into this issue, we have performed spatial transcriptomics of the adult SAN and RA of *delB* mice and controls (revised Supplementary Fig. 10 and Supplementary Data Tables 17,18). While the fetal *delB* RA is transcriptionally unaffected (revised Fig. 3b-d), we do see that several genes associated with cardiac stress are upregulated in the adult *delB/delB* RA but not in the *delB/+* RA (Supplementary Fig. 10e). This correlates with the arrhythmogenesis being much stronger in the homozygous *delB/delB* mice compared to the heterozygous *delB/+* mice. As shown in Supplementary Fig. 8, Supplementary Fig. 9 and Supplementary Table 1, *delB/delB* mice show irregular and inconsistent activation of the atria. We hypothesize that prolonged, severe SND and irregular atrial activation drives a pro-arrhythmogenic stress response in *delB/delB* RA. We have integrated this explanation in the Discussion. Clinically, the correlation between AF and SND has been well established, although a mechanism remains to be demonstrated (PMID: 27166347, PMID: 28155995, PMID: 39747593). We believe our model and study provides some mechanistic insight into this relationship, and suggests that SND can predispose to AF.

2) The model presented in Fig. 7c suggests upregulation of PITX2 in pacemaker cells and downregulation of PITX2 in the left atrium. Have both of these expression changes ever been shown in the same patient /animal model? It would be important to more clearly explain this in the discussion. Otherwise, a model focused on the *delB* mutation studied here might be more appropriate.

Response:

Demonstration of the contribution of PITX2 to AF has to date been limited to altered expression in the LA (or pulmonary vein myocardium). Generally, it is assumed, and in a few cases experimentally validated, that PITX2 expression is reduced in AF risk SNPs carriers with AF compared to non-risk allele carriers with AF (see for example, PMID: 21282332, PMID: 20457925, PMID: 29892015). Beyond the *PITX2* locus deletions reported by Baudic et al., altered PITX2 expression in the SAN has not yet been described in experimental models or in patients carrying deletions or AF/SND-associated SNPs in this locus. However, the recent finding that previously identified AF-associated SNPs are also associated with SND (PMID: 39747593) strongly suggests a common mechanism. The discussion and figure caption have been changed to make it clear that this is our running model.

3) The manuscript text in lines 292-297 and Fig. 4g, h does not align. The text states that PC nuclei positive for ISL1, PITX2 or both were analyzed but the figures only show analysis of PITX2 positive nuclei. In addition, the authors make the statement that “the *delB/delB* HCN4^{high} subdomain contains a smaller fraction of total PC nuclei and the HCN4^{low} subdomain contains a larger fraction of total PC nuclei compared to their respective subdomains in *delB/+*

SANs". To support this claim statistical analysis between the HCN4^{high} and HCN4^{low} groups in the *delB/delB* vs *delB/+* samples needs to be included.

Response:

We thank the reviewer for pointing out this discrepancy. In this analysis, we selected regions and quantified nuclei in sections stained for TNNI3, HCN4, PITX2 and DAPI and only used adjacent sections stained for ISL1 to assist in defining the SAN domain. We have added the statistical analysis in revised Fig. 5f.

4) The use of *delB/+* and *delB/delB* vs wildtype mice and dissection of HCN4^{high} vs HCN4^{low} regions allowed the authors to make conclusions on gene expression changes based on PITX2 dosage. However, they never show a direct correlation of PITX2 expression levels to pacemaker gene expression changes. A graph detailing the actual expression levels of PITX2 and the corresponding gene expression changes would be a valuable addition to the manuscript. Such analysis might further help to define the expression threshold of PITX2 that needs to be reached to delineate pacemaker cell differentiation.

Response:

To more clearly show how *Pitx2* expression level in the *wild-type*, *delB/+*, and *delB/delB* SAN and *Pitx2*-native LA body and LA auricle align with stepwise changes in pacemaker and atrial myocardium-associated gene expression, we have added spatial transcriptomic data of the LA body and LA auricle (Fig. 4b,c). In addition to these figures, the spatial transcriptomic datasets in the Supplementary Data Tables provide further data on expression levels and fold-changes between the groups.

Furthermore, in revised Fig. 3c and d, we show *Pitx2* expression levels in *wild-type* and *delB* SAN (HCN4^{high} and HCN4^{low}, respectively) and RA alongside that of several relevant pacemaker and atrial genes to visualize the correlation between *Pitx2* expression and PC transcriptional deregulation. These data indicate that the smaller increase in *Pitx2* expression in the HCN4^{high} domain is associated with less profound changes in gene expression compared to the larger increase in the HCN4^{low} domains. We quantified levels of both ISL1 and PITX2 in individual nuclei of the *wild-type* and *delB* SAN, which revealed that the decrease of ISL1 is proportional to the increase in of PITX2 (revised Fig.5b). However, the changes in HCN4 seem to follow a threshold response (Fig. 8a).

5) The MALAT1-based exclusion of clusters 4+8 is an unconventional practise not widely used in the field (manuscript text lines 311-312 and Supplementary Figure 4c). In addition, the authors exclude cells based on high levels of MALAT1 expression while the cited publication recommends to exclude cells with 0/low MALAT1 expression. Could the authors please explain this discrepancy and show further gene expression of these clusters to confirm poor cell quality. For example, what is the mitochondrial gene count for these clusters? This is particularly important because based on Supplementary Fig. 4i the AAV6-PITX2 sample is skewed towards contribution to the MALAT1+ clusters. By excluding these clusters, the authors ignore a large fraction of AAV6-PITX2-derived cells. Why do AAV6-PITX2 cells contribute stronger to the MALAT1+ cell clusters? This should be discussed in the manuscript.

Response:

We acknowledge the comment of the reviewer that we did not clearly describe the exclusion criteria and the role of MALAT1 herein. We imposed the following quality control cut-offs for determining which cells were included in our analysis: the number of reads detected in an individual cell should be >500, the number of different genes detected >350 and the percentage of reads mapped to mitochondrial genes should be <20%. The default threshold that is used in the Seurat vignette for mitochondrial read percentage is 5%, but for cardiomyocytes, which naturally exhibit elevated mitochondrial gene expression due to their high energy demand, more permissive thresholds can be applied

to avoid exclusion of metabolically active, viable cardiomyocytes (<https://www.nature.com/articles/s41586-020-2797-4>).

The long non-coding RNA MALAT1 is highly expressed in nuclear-retained fractions and has been used as a marker for nucleus-containing cells in single-cell RNA-seq workflows. As the reviewer correctly points out, low or absent MALAT1 expression has previously been associated with ambient RNA or cytosolic debris (PMID: 39574015; Clarke and Bader, 2024; <https://www.biorxiv.org/content/10.1101/2024.07.14.603469v1.full.pdf>; and references therein). However, very high MALAT1, such as observed in cluster C8 (revised Fig. 6b), has been associated with damaged cells (loss of cytoplasmic RNA: increased nuclear/cytoplasmic RNA ratio)(PMID: 34857027; PMID: 39574015) and with cell stress. C8 is disproportionately composed of cells with AAV6-*PITX2* (revised Fig. 6c), raising the possibility that these cells are stressed by high levels of *PITX2*.

Importantly, as explained in the response to comment 6 below, only clusters C0 and C7 express PC marker genes (revised Fig. 6b) which is why we excluded all other clusters from our analysis of the response of PCs to *PITX2* overexpression, irrespective of their MALAT1 content. We have revised Fig. 6 and the Results section to clarify this issue.

6) It is not clear how the authors go from the original UMAP with 9 cell clusters to the reduced UMAP with 5 cell clusters (Fig. 5b, Supplementary Fig. 4c, d). Removal of clusters 4 and 8 due to high MALAT1 expression would have resulted in 7 clusters. Where did the other 2 cell clusters go? Along the same lines it is not clear whether cluster 7 in Supplementary Fig. 4c and Supplementary Fig. 4d are the same cells. Was the data re-clustered after removal of clusters 4 and 8? Please provide a more clear explanation of this analysis step.

Response:

We thank the reviewer for bringing this to our attention. The first cluster reduction step from 9 to 5 clusters was indeed superfluous and somewhat unclear. To simplify and clarify our analyses, we have kept the original UMAP with 9 clusters. Based on the expression of PC marker genes and cardiomyocyte-specific genes we have identified clusters C0 and C7 as representing PC-like cells. The other clusters represent by-products of the PC differentiation process, including atrial and ventricular cardiomyocyte-like cells, epicardial-like cells, non-cardiomyocytes (about 60-70% in this case). Further analysis focused on the 2 clusters with PC-like cells. First, clusters C0 and C7 were taken together (representing the total PC population) and the AAV6-*PITX2* PCs were compared to the AAV6-*mCherry* PCs. Subsequently, because we noted that AAV6-*PITX2* PCs were enriched in C7 (and not in C0) and expressed more *PITX2* (*EGFP*) compared to C0, we tested the hypothesis that C7 represented PCs that were more affected by *PITX2*, with reduced expression of PC gene expression, and increased expression of some atrial genes that were also increased in mouse *deIB* HCN4^{low} vs HCN4^{high} subdomains (revised Fig. 6f, g). Fig. 6 and Supplementary Fig. 6 and accompanying Results text have been modified accordingly.

7) The claims that the authors make with regards to *PITX2* expression and proportion of AAV6-*PITX2* cells in the different clusters is not well supported with the current display figures. It would be helpful to add an expression UMAP of *PITX2* similar to what is shown for *ISL1* in Figure 5c. It would further be helpful to show a UMAP that displays the origin of each cell from either the *PITX2* or *mCherry* experimental group in the main Figure 5 following figure 5b. In addition, Figure 5f is elegant but could be better explained / labelled in the graph to make it clear that this is the change in cell distribution between AAV6-*mCherry* and AAV6-*PITX2* samples and does not display gene expression.

Response:

The analysis of the scRNA-seq data, Fig. 6 and Supplementary Fig. 6 have been modified and simplified to clarify how we arrived at the interpretation that clusters C0 and C7 comprise the PCs. We have moved information regarding

cluster composition and gene expression to the forefront in Fig. 6, including the proportion of AAV6-*PITX2* cells in each cluster in Fig. 6c and a UMAP showing the distribution of *EGFP* expression in AAV6-*mCherry* and AAV6-*PITX2c* cells in Fig. 6d. Because the iPSC-PC differentiations contain >60% of non-PC-like cells (i.e. both *Cherry* and *PITX2*-transduced populations), we initially removed some but not all non-PC cell clusters as was shown in the original Figure. We now skip this intermediate step and directly focus on C0 and C7, based on the expression of PC marker genes, cardiac genes, AAV6-*Cherry* or AAV6-*PITX2*, and their close proximity (intermingling) in the UMAP. We acknowledge that this may result in dismissing original PC cells that acquired a strongly altered state due to *PITX2* expression, and therefore present in one of the cell clusters with undefined identity. However, we estimate that these cells form a minority and that their interpretation involves great uncertainty. We have replaced Fig. 5f with a bar graph (Fig. 6c) and added a volcano plot showing differential gene expression between AAV6-*PITX2c* C0 and C7 (Fig. 6f) to more clearly show the differences between these clusters.

8) In the abstract the authors state: "Ectopic expression of *PITX2c* in human induced pluripotent stem cell-derived PCs resulted in dysfunction and *PITX2* dosage dependent transcriptional changes reminiscent of those observed in the sinus node subdomains of *delB* mice." However, the presented data does not support the claim of *PITX2* dose dependent transcriptional changes well. A more detailed analysis of how *PITX2* expression levels relate to expression changes e.g. via a graph displaying *PITX2* expression levels and corresponding gene expression changes, similar to the comment in point #4, would be better supporting this claim.

Along the same lines it would be interesting to directly compare changes in gene expression between clusters 0 and 7 in the AAV6-*PITX2* samples to identify which expression changes define whether cells lose their PC phenotype and "fall" into cluster 7 or maintain the PC phenotype and "remain" in cluster 0. How does *PITX2* expression differ between cells in these clusters – is it significantly different?

Response:

In order to further strengthen our claim regarding *PITX2* dosage-dependent transcriptional changes, we have performed an analysis of the correlation between gene expression and *EGFP* expression (= *mCherry* or *PITX2*) in AAV6-*mCherry* and in AAV6-*PITX2* PCs in clusters C0 and C7 (revised Supplementary Fig. 6g,h). PC genes like *PLN*, *HCN1*, *HCN4* show negative correlation with *PITX2* levels, not with *mCherry* control levels, and atrial genes like *CLU* show positive correlation with *PITX2* and not *mCherry*. As positive control, revised Supplementary Fig. 6h shows positive and negative correlations with *PITX2* and *EGFP* follow the same trend, which is to be expected as both *EGFP* and *PITX2* transcripts report for AAV6-driven *PITX2-p2a-EGFP*.

We have additionally modified Fig. 6 to more clearly show differential gene expression between clusters C0 and C7 in AAV6-*PITX2* cells and which of these changes were statistically significant. As shown in Fig. 6f, *PITX2* (*EGFP*) expression in AAV6-*PITX2* PCs is indeed significantly higher in C7 compared to C0, like it is in *delB/delB* *HCN4*^{low} PCs compared to *delB/delB* *HCN4*^{high} PCs. When comparing transcriptome differences between C0 and C7 with the differences found between the *delB/delB* *HCN4*^{high} vs *HCN4*^{low} subdomains, we observed that conserved markers were comparably deregulated (e.g. PC genes down regulated, *SCN5A* and *CLU* upregulated in C7-*PITX2* vs C0-*PITX2*) (revised Fig. 6). To more systematically assess whether C0-*mCherry*, C0-*PITX2* and C7-*PITX2* correspond to PCs in the *wild-type* SAN, *delB/delB* *HCN4*^{high} and *HCN4*^{low} subdomains, respectively, differential gene expression between C0-*PITX2* and C0-*mCherry* was plotted against differential gene expression between C7-*PITX2* and C0-*mCherry* (revised Fig. 6g). Likewise, differential gene expression between the *delB* *HCN4*^{high} subdomain and *wild-type* SAN was plotted against differential gene expression between the *delB* *HCN4*^{low} subdomain and *wild-type* SAN. In both models, the direction of change is comparable (i.e. the same genes either up- or downregulated in both C0-*PITX2* and C7-*PITX2* and in both *delB/delB* *HCN4*^{low} and *HCN4*^{high} PCs compared to their controls), but the degree of change is larger in C7-*PITX2* compared to C0-*PITX2* (slope 1.3) and in *delB/delB* *HCN4*^{low} compared to *HCN4*^{high} (slope 1.4). Taken together, gene expression changes (including *PITX2*) in the C0-*PITX2* and C7-*PITX2* clusters show behavior similar to that in *delB/delB*

HCN4^{low} and HCN4^{high} subdomains, which suggests that in both the mouse model and the iPSC-PC model, PITX2 dose-dependent threshold responses are observed. Nevertheless, more extensive quantitative experiments would be necessary to define which genes show such a threshold response to specific PITX2 expression levels.

9) The presented data suggests that ectopic PITX2 expression in pacemaker cells results in upregulation of the atrial gene program based on a few genes (SCN5A, GJA1, GJA5, NKX2-5). How do the PITX2+HCN4^{low} cells compare to atrial and ventricular cardiomyocytes on the transcriptome level? Do they convert to atrial cardiomyocytes? Or do they take on a non-physiological intermediate phenotype? This question could be addressed using either the mouse or human induced pluripotent stem cell expression data and comparison to existing transcriptome data of atrial and ventricular cardiomyocytes.

Response:

As also outlined in response to Reviewer comment 1, the transcriptomes of PITX2+ PCs overlap minimally with those of RA/LA-CMs. However, these cells do express some functionally relevant RA/LA-CM associated genes, as illustrated in Supplementary Fig. 4f, where we compared the transcriptomic changes in the *delB/delB* HCN4^{low} subdomain with the differences found between atrial CMs and PCs by Goodyer et al 2019 (PMID: 31284824).

10) Finally, the use of human induced pluripotent stem cells is an asset to this study allowing the validation of the observed mechanism in a human model system. To increase the impact and potential future clinical translation of the findings of this study it would be important to know whether inhibition of the ectopic PITX2 expression can rescue the phenotype and re-establish the pacemaker phenotype in the AAV6-PITX2 SAN pacemaker cells.

Response:

We fully agree with the suggestion that inhibition of ectopic PITX2 expression would provide important information regarding the reversibility of PITX2-mediated pacemaker cardiomyocyte (PC) state changes. We suspect the changes will be irreversible because the transcription factor genes known to be required for establishing and maintaining the PC state are completely (*Isl1*, *Shox2*, *Tbx3*) or partly (e.g. *Gata6*) downregulated, and other factors that antagonize PC differentiation are upregulated (e.g. *Nkx2-5*). As disruption of single PC transcription factors of this network is already sufficient to cause PC state loss, and these factors establish and maintain the regulatory network by positive feedback, it is likely that disruption of the entire transcriptional network will cause its irreversible loss. While we are currently trying to establish rather laborious and costly experiments to address this question (CRISPRi *in vivo* and Xon-mediated inducible expression of PITX2 in iPSC-PCs), which we believe are outside the scope of this manuscript.

Nevertheless, we have performed an experiment addressing an important issue related to this question, which we have added to the manuscript (Supplementary Fig. 3a). The PITX2-induced PC state changes in *delB* mice take place during embryonic development, when the PCs are differentiating from precursors and the sinoatrial node (SAN) is being formed. To assess whether PITX2 also causes such state changes when PCs are fully differentiated, we transduced postnatal mice with AAV9-PITX2 (under control of the cardiomyocyte-specific TNNT2 promoter). After 2 weeks, the hearts were harvested, the expression of PITX2 (or control EGFP), ISL1 and HCN4 in the SAN was assessed by immunofluorescence (Supplementary Fig. 3b-d and Results). The results show that ISL1 and HCN4 are downregulated in PITX2+ PC in the SAN, suggesting that PITX2 causes a state change in differentiated PCs that is similar to the change in embryonic PCs.

Minor:

1) In line 167-186 the authors state: “Immunostaining further revealed co-expression of ISL1 and PITX2 in the differentiating PCs of the E10.5 *delB/delB* SAN (Fig. 1b)”. Fig 1b does not clearly show the co-expression. Please show separate fluorescent channels or a more clear overlay image with insets to better support this claim.

Response:

We thank the reviewer for their minor comments.

We have added insets to revised Fig. 1b to more clearly show the overlapping DAPI/ISL1/PITX2 signal in the *wild-type* and *delB/delB* E10.5 SAN nuclei.

2) In Supplementary Fig. 4b the labels of the flow plots appear to be switched. It is also misleading to label the upper right quadrant in the 2nd plot with 93% while it has almost no cells. I recommend adding the enlarged label in the appropriate quadrant.

Response:

The flow plot labels have been corrected and we moved the # label to the appropriate quadrant in Supplementary Fig. 6b.

3) In Fig. 5e, it is not clear what the purple color code indicates – what is the legend? In addition, genes seem randomly highlighted in bold. It would be more clear if the authors highlight all the genes mentioned in the text in bold or explain the reasoning as to why some genes are bolded and others are not.

Response:

A color legend and the text have been altered to clarify that purple dots indicate genes that were significantly differentially expressed in both comparisons ($p_{adj} < 0.05$) and the labels have been restricted to those relevant for the discussion.

4) Similarly, in Fig. 5h it is not clear what the purple color code indicates.

Response:

A color legend has been added and the text has been altered to clarify that the purple dots indicate significantly differentially expressed genes in both comparisons ($p < 0.005$).

5) The authors are referring to the PITX2 transfected cells as Pitx2+ PCs which gets confusing when talking about the single cell RNA-seq data because it is not clear whether the authors refer to all the cells expressing PITX2 or to the experimental group of cells that was transfected with PITX2 AAV6. To make this more clear to the reader I suggest referring to the PITX2 overexpressing cells as AAV6-PITX2 cells consistently.

Response:

We have modified the text to consistently refer to AAV6-PITX2 cells as such.

6) In Supplementary Fig. 4g the GO terms seem to be sorted from lowest fold change to highest fold change based on the color code. This is counterintuitive and similar figures in the manuscript seems to be sorted from highest to lowest fold change as one would expect.

Response:

We have adjusted Supplementary Fig. 6e to show GO terms from highest to lowest fold change.

7) In line 325 the figure reference is probably supposed to state “Supplementary Fig. 4i, j” instead of “Fig 4i, j”

Response:

The text has been corrected accordingly.

8) In the legend of Fig. 7 the labels for panel c and b are switched.

Response:

Fig. 8 has been modified accordingly.

Reviewer #2 (Remarks to the Author)

Overview

This data-rich manuscript builds on a prior report of a novel Mendelian arrhythmia syndrome causing sinus node dysfunction (SND) and atrial fibrillation (AF). In these families, deletion of a noncoding region containing a CTCF binding site disrupts a TAD boundary at the *Pitx2* locus, resulting in ectopic expression of *Pitx2* in the SAN, presumably causing premature sinus node dysfunction and atrial fibrillation. In the present work, the mechanistic basis for these phenotypes is investigated further through an examination of SAN development, spatial organization of SAN gene expression, and heart rhythm in a mouse model of the human deletion syndrome ('delB' mice). In addition, the consequences of *Pitx2* misexpression are explored in an SAN iPSC model system. Although the mouse model was generated and characterized in the prior study, the present work provides substantial additional mechanistic data on the phenotypes of these mice and introduces spatial transcriptomics to characterize distinct regions within the SAN.

The main findings are (1) a few *Pitx2*-expressing cells are present in or near the WT SAN, (2) misexpression of *Pitx2* in the SAN primordium of delB mice is present at the onset of SAN morphogenesis and results in downregulation of key SAN transcription factors; (3) delB/delB embryonic SAN exhibits patchy expression of *Hcn4* with *Hcn4*^{low} and *Hcn4*^{high} areas that can be distinguished using spatial transcriptomics; (4) *Pitx2* dose-dependently represses *Isl1* and SAN gene expression in mouse and hiPSC SAN cultures with corresponding changes in SAN electrophysiology in vivo and in vitro. Based on these findings, the authors conclude that some rare *Pitx2* expressing cells in the SAN and perinodal area may be necessary for normal heart rhythm, that pacemaker cells are sensitive to low levels of *Pitx2* expression, but that once *Pitx2* expression exceeds a threshold in a given PC, pacemaker gene expression essentially collapses with reprogramming to a different non-PC-like cell state. These findings may explain the clinical phenotype and variable expressivity observed in patients with the deletion syndrome and could have implications for the impact of common genetic variation at this locus on heart rhythm.

Overall, the quality of the raw data and data analysis is uniformly high. New and rich datasets are offered to the field. Mouse physiology studies are, if anything, overpowered and these data are very convincing. In general, the data support the conclusions although, as detailed below, there may be alternative explanations for some key findings that are not thoroughly considered but could have a major effect on the interpretation and explanatory model. In terms of conceptual novelty, it is already well established that *Pitx2* represses SAN gene expression so in that sense these findings fit well with existing literature and can be seen as incremental. However, if a threshold effect of *Pitx2* could be more convincingly demonstrated, this could offer novel insights on transcription factor dose-dependence in the SAN.

Response:

We thank the reviewer for their appreciation of the data and analysis presented in this study and for their insightful comments.

As detailed below, we have revised our analyses and figures and have added a new spatial transcriptome analysis including left atrial tissue with the aim to more convincingly show the dosage-dependent effect of *PITX2* on pacemaker cardiomyocyte (PC)-associated gene expression and state. Moreover, we have performed additional experiments and analyses to address the reprogramming of PCs versus blocking of specification or replacement of PCs.

Regarding the comment that *Pitx2* has been shown to repress SAN gene expression: The role of *Pitx2* in heart development has been investigated in *Pitx2* loss-of-function mouse models, in which *Pitx2* deficiency causes right isomerism of the left sinus venosus and atrium, resulting in left-sided SAN gene expression and left-sided SAN development. The role of *Pitx2* in SAN development has not been studied as *Pitx2* is not expressed in the SAN or right

atrium and these regions are therefore not directly affected in loss-of-function models. The *delB* mouse model therefore represented an opportunity to gain mechanistic insight into the role of PITX2 in the SAN. Furthermore, as outlined in the Introduction, several studies have associated SNPs upstream of *PITX2* with AF, presumably acting through reduced *PITX2* dosage in the left atrium (possibly pulmonary vein). It is only recently, however, that an association between these AF-associated SNPs and SAN dysfunction (SND) has been identified (Weng, 2025 Nature Genetics; PMID: 39747593). In light of the reported absence of PITX2 from the SAN and right atrium, these associations are not understood. Our study provides insight into this association.

Specific

Comments:

1. Line 173. The authors define a “dosal SAN head domain” that is Pitx2+, Hcn4-low, Isl1-negative. This profile resembles the adjacent SVC/sinus venosus myocardium. From the images in Supplementary Figure 1C that are intended to demonstrate this anatomical structure, it’s not clear what would justify ascribing a new transcriptional domain to the SAN head versus just a region where there is some interdigitation of SAN PCs with SVC myocardium.

Response:

We agree with the reviewer that this domain received more attention than was justified. A PITX2⁺, HCN4^{low} and ISL1⁻ profile does indeed resemble that of the adjacent SVC/sinus venosus myocardium, and as far as we know, this structure may not be transcriptionally distinct. We found this observation noteworthy, however, as it is the only region in the otherwise isolated SAN head domain that forms a continuum with the SCV. Additionally, as this region expresses CX40, which is normally absent from the SAN head domain, and only expresses low levels of HCN4, we would expect that these cells have an electrophysiological phenotype distinct of that of PCs typically found in the SAN head. The observation we deem important in the paragraph describing this region, however, is the presence of a few sporadic PITX2+ cardiomyocytes (presumably PCs) in the otherwise PITX2-negative SAN of *wild-type* mice. We have revised this paragraph to emphasize the sporadic PITX2+ cells in the SAN, and put the PITX2+ SAN domain associated with the SCV into perspective.

2. Figure 2e proposes that Pitx2 gradually “reprograms” PCs by repressing Isl1/Tbx3/Shox2, resulting in de-repression of Nkx2-5 and eventually upregulation of Gja1, downregulation of Hcn4, and further reduction in Isl1/Shox2/Tbx3. This model is certainly plausible and could explain the observations. However, can the authors exclude another scenario wherein Pitx2 misexpression either blocks specification of SAN head cells, or leads to SAN hypoplasia, allowing the expansion of other nearby cell types into the SAN (SVC myocardium, SAN transitional cells, Osr1+ DMP-derived tissue), accounting for some additional cell type heterogeneity in SAN sections? Because PCs are not irreversibly labeled prior to the onset of Pitx2 misexpression, it’s hard to rule out this other scenario from the data presented. Because such an experiment is not currently feasible, do the authors have any data on proliferation of the SAN head and/or nearby tissue compartments in *delB* versus WT mice after the onset of Pitx2 misexpression?

Response:

We thank the reviewer for their insightful comment. Indeed, we cannot exclude the possibility that the HCN4^{low} regions are the product of the expansion of other embryonic cell types beyond their normal domains. It is important to note, however, that at E12.5, 3 days after the formation of the definitive SAN, we see that while the expression of TBX3, ISL1 and SHOX2 is reduced in the *delB/delB* SAN, the expression of HCN4 and of Cx40 is not (yet) altered (Fig. 2a and Supplementary Fig. 1b). This indicates that the initial specification of pacemaker cardiomyocytes (PCs) in *delB* mice is not disrupted.

Gene Ontology analysis of the E17.5 *wild-type* vs. *delB/delB* SAN (spatial transcriptomics, Supplementary Data Table 4) and of the AAV9-*mCherry* cluster 0 (PCM) vs AAV9-*PITX2c* cluster 0 (scRNA-seq, Supplementary Data Table 13)

shows that relevant cell cycle or growth-associated GO terms are not present among the top enriched terms (neither by significance nor fold-change). These data indicate that at this stage, there is no difference between cell cycle-associated processes in PITX2+ HCN4^{low} PCs and *wild-type* PCs. In addition, the iPSC-derived PCs used in our experiments are differentiated and show very little, if any, proliferative activity (Reviewer Figure 1 and PMID: 36217819) at the time they are transduced with AAV6-*Cherry* or AAV6-*PITX2* (day 19). Therefore, it is more likely that the subdomains in the *deIB* SAN arise from PITX2+ PCs that change state rather than replacement of PCs by a different cell population.

Reviewer Figure 1. a. UMAP visualization of single cell RNA-sequencing data illustrating the cell composition generated by our hiPSC-PC differentiation protocol. We assigned a proliferation score to the cells based on the relative expression level of a panel of genes involved in proliferation/cell cycle activity (*MKI67*, *PCNA*, *TOP2A*, *CDK1*, *CCNB1*, *CCNB2*, *CCND1*, *CCND2*, *MCM2*, *MCM3*, *MCM4*, *MCM5*, *MCM6*, *MCM7*, *CDC20*, *CDCA8*, *BUB1*, *BUB1B*, *AURKA*, *AURKB*, *PLK1*, *CCNE1*, *CCNE2*, *E2F1*, *E2F2*, *RRM2*). The proliferation score was computed in Seurat using AddModuleScore, which calculates the average expression of a predefined set of genes, subtracted by the aggregated expression of control gene sets. The proliferation score shows that the expression of these proliferation markers is higher (red) during the early stages of differentiation and starkly reduces (blue) as cells proceed to terminally differentiate in either ventricular or pacemaker cardiomyocyte (VC, PC, respectively). **b.** UMAP showing that the proliferation score corresponds with higher *TNNT2* expression levels (red) in late-stage PCs. Plots were generated using data from PMID: 36217819. D0, 0 days of differentiation; D03, 3 days of differentiation etc. D19-PCs represent the terminally differentiated PCs used in the current study.

To validate whether PITX2 is capable of reprogramming fully differentiated, non-dividing PCs, we performed an additional experiment in which *PITX2c* was overexpressed in cardiomyocytes, including SAN PCs, of postnatal P2 *wild-type* mice by intraperitoneal injection of AAV-*PITX2c-ires-EGFP* or AAV-*Cherry-ires-EGFP* (control) expression vectors. 14 days later, resting ECGs indicate that SAN function was disrupted in the mice that had received the *PITX2c* virus (see Reviewer Figure 2). Immunohistochemical staining for EGFP, PCM-1 (cardiomyocyte marker) and PITX2 show high transduction efficiency and ectopic PITX2 expression in CMs, including SAN PCs (revised Supplementary Fig. 3b). We found that ISL1 and HCN4 expression was strongly reduced in PITX2+ cells in the SAN (revised Supplementary Fig. 3d). Taken together, these data indicate that ectopic PITX2 expression is capable of reprogramming differentiated PCs and suggest that the HCN4^{low} regions do indeed consist of “reprogrammed” previously-specified PCs rather than of nearby cell types that have expanded into the SAN.

Reviewer Figure 2 Reduced heart rate and elevated heart rate variation in *wild-type* P16 mice 14 days post-transduction with AAV9-h*PITX2c* expression vectors indicate that ectopic *PITX2* expression in fully-differentiated PCs disrupts SAN function (one-way ANOVA with Tukey’s multiple comparisons test).

3. Related to the previous comment, could the alternative scenario also partly explain the apparently patchy nature of different expression domains (clustering of HCN4-LOW cells into domains) despite the stochastic nature of *Pitx2* mis-expression level within the SAN which does not seem to show the same degree of spatial clustering (e.g. Figure 4a)? This would be an alternative to postulating a “threshold response” to explain the data in Figure 4D, where *delB/+* and *delB/delB* cluster by cell state (HCN4-low vs HCN4-high) rather than genotype. Because other embryonic cell types are not represented on the PCA (e.g., DMP derivatives, non-SAN head PCs), it’s hard to know whether the HCN4-LOW regions might represent expansion of these cell types beyond their usual borders when the SAN head is smaller. In figure 4c, for example, the HCN4LOW region appears non-uniform with a predominantly *Hcn4*-negative region interdigitating with a few *Hcn4*-high cells. Put differently, could the focus on identification and characterization of gene expression domains be obscuring a broader morphogenetic defect in the dorsal atrium and can the authors comment on whether SAN, RA, SVC/sinus venosus, and venous valve/DMP are different in the *delB* genotype?

Response:

As the initial formation of the SAN is not disrupted, the loss of downstream PC gene expression predominantly takes place during the fetal period, and the absence of increased cell cycle/growth associated gene expression in the *delB/delB* SAN subdomains, we doubt whether the patchy nature of the transcriptional subdomains is the product of expansion of other cell types into this region. Despite the apparent stochastic nature of *Pitx2* mis-expression in the SAN, we do see that clusters of HCN4^{high} PCs in the *delB/delB* SAN, as a whole, express less *PITX2* than adjacent HCN4^{low} PCs. Additionally, in our immunohistochemical analysis, we did not detect unusual (clusters of) non-CMs in or adjacent to the SAN.

We did not detect any structural alterations or changes in HCN4, CX40 or troponin T expression in the SCV or VV in fetal *delB* mice. Immunostainings did not show any changes in *PITX2* expression pattern or level in the fetal *delB* SCV. As the quality of NanoString GeoMx-DSP data is strongly influenced by the number of cells (nuclei) in a ROI (a minimum of 50 nuclei/ROI), we were unable to include the fetal SCV or VV in our initial spatial transcriptomics experiments. The fetal RA, however, is clearly transcriptionally unaffected and we have added additional data that shows that the fetal *delB* LA body (adjacent to the inter-atrial septum) and the LA auricle are also transcriptionally unaffected (revised Fig. 4b-c). We performed spatial transcriptomics analysis on of SAN and RA on sections of adult hearts, shown in revised Supplementary Fig. 10 and Data Table 14. We were able to include a few SCV samples of

wild-type and *delB* mice, in which we did not detect differences in expression of *Pitx2* or of genes that do respond to *Pitx2* in SAN (Reviewer Figure 3). Taken together, it is unlikely that the structural/transcriptional defects in the *delB* SAN domain are the product of gene deregulation in other nearby cell populations.

Reviewer Figure 3. Expression levels of pacemaker- and working myocardium-associated genes in the adult *wild-type*, *delB/+* and *delB/delB* SCV myocardium.

4. Related to the above comments, since the hiPSC-PC *Pitx2* overexpression system revealed a wide range of *Pitx2c* overexpression levels, was a threshold effect observed in terms of downstream gene expression or cell state that paralleled the dichotomization observed in the mice into *Hcn4*-HIGH and *Hcn4*-LOW regions? If not, would these human data then militate against the threshold model?

Response:

In order to address this question, we have performed an analysis of the correlation between gene expression and *EGFP* expression (= *mCherry* or *PITX2*) in AAV6-*mCherry* and in AAV6-*PITX2* PCs in clusters C0 and C7 (revised Supplementary Fig. 6g,h). Of note, the scRNA-seq data is intrinsically noisy, and the expression of many genes was sparsely detected in the cell population. For this reason, we could not assess the correlation with *PITX2/EGFP* for the majority of genes. Nevertheless, PC genes like *PLN*, *HCN1*, and *HCN4* show negative correlation with *PITX2* levels, not with *mCherry* control levels, and atrial myocardium genes like *CLU* show positive correlation with *PITX2* and not *mCherry*. As positive control, revised Supplementary Fig. 6h shows positive and negative correlations with *PITX2* and *EGFP* follow the same trend, which is to be expected as both *EGFP* and *PITX2* transcripts report for AAV6-driven *PITX2-p2a-EGFP*.

We have additionally modified Fig. 6 to more clearly show differential gene expression between clusters C0 and C7 in AAV6-*PITX2* cells and whether these changes were statistically significant. As shown in Fig. 6f, *PITX2* (*EGFP*) expression in AAV6-*PITX2* PCs is indeed significantly higher in C7 compared to C0, like it is in *delB/delB* *HCN4*^{low} PCs compared to *delB/delB* *HCN4*^{high} PCs. Upon comparing the transcriptome differences between C0 and C7 with the differences found in the *delB/delB* *HCN4*^{high} vs *HCN4*^{low} regions, we observed that conserved markers were comparably deregulated (e.g. PC genes down regulated, *SCN5A* and *CLU* upregulated in C7-*PITX2* vs C0-*PITX2*) (revised Fig. 6g). To more systematically assess whether C0-*mCherry*, C0-*PITX2* and C7-*PITX2* correspond to *wild-type* SAN, *delB/delB* *HCN4*^{high} and *HCN4*^{low}, respectively, differential gene expression between C0-*PITX2* and C0-*mCherry* was plotted against differential gene expression between C7-*PITX2* and C0-*mCherry* (revised Fig. 6g). Likewise, differential gene expression between *delB* *HCN4*^{high} and *wild-type* SAN was plotted against differential gene expression between *delB* *HCN4*^{low} and *wild-type* SAN. In both models, the direction of change is comparable, i.e. the same genes either up or

downregulated in both C0-*PITX2* and C7-*PITX2* and in both *delB/delB* HCN4^{low} and HCN4^{high} PCs compared to their controls (C0-*mCherry* and wild-type SAN PCs), but the degree of change is larger in C7-*PITX2* compared to C0-*PITX2* (slope 1.3) and in *delB/delB* HCN4^{low} compared to HCN4^{high} (slope 1.4). Taken together, gene expression changes (including *PITX2*) in the C0-*PITX2* and C7-*PITX2* clusters show behavior similar to that in *delB/delB* HCN4^{low} and HCN4^{high} PCs suggesting that in both the mouse model and the iPSC-PC model, *PITX2* dose-dependent threshold responses are observed. Nevertheless, more extensive quantitative experiments would be needed to define which genes show such threshold response to which level of *PITX2*.

5. How does the level of *pitx2* seen in the *delB* SAN compared to WT LA?

Response:

We quantified the *PITX2* signal intensity in *wild-type* and *delB/delB* fetal SAN, left atrial body (LAB) and left atrial auricle (LAA), and observed that the ectopic *PITX2* expression in the *delB/delB* SAN is intermediate to that found in the LAA and LAB (Revised Fig.4a). As *PITX2* expression levels in the LAB differs substantially from that in the LAA, we separately compared the transcriptomes of *delB* SAN with that of the LAB and LAA. In addition to capturing the distinct transcriptional profiles of the *wild-type* SAN, *delB/delB* HCN4^{high} and HCN4^{low} regions, principal component analysis (PCA) of our fetal spatial transcriptomics data shows that the transcriptomes of the LAA and LAB are distinct from one another (Fig. 4b). *Pitx2* transcript levels in the LAB are greater than those in the LAA and those in the *delB/+* and *delB/delB* SAN do not exceed those found in the LAA (Fig. 4c). Taken together, these data indicate that *Pitx2* transcript and protein levels in the *delB* SAN remain within physiologically relevant levels.

6. Since some C7 cells were observed in the AAV6-mCherry hiPSC cells, can the authors assign a cell type to this cluster? Could C7 represent SVC/sinus venosus cells?

Response:

While it is possible that C7 cells resemble SCV cells, the transcriptional profile of C7 aligns well with that of PCs (PMID: 36217819).

Minor:

8. Figure 5m – dV/dt max not Vmax

Response:

Revised Figure 6l (previously Figure 5) and the associated figure legend have been adjusted accordingly.

Reviewer #3 (Remarks to the Author):

In this manuscript van der Mareel et al., investigate the phenotype of the 'delB' mouse, with a 15kb deletion of an intergenic region upstream of Pitx2. Previous work from the lab (Baudic et al.) had found that delB mice show an increased risk of atrial arrhythmias, mimicking the phenotype observed in 7 human families with deletions within the paralogous region. Furthermore, they found increased expression of Pitx2 in delB mice, particularly within the SAN, which was proposed to be downstream of changes in chromatin conformation.

In this work, the authors further characterize the delB phenotype, finding that heterozygous and homozygous delB embryonic hearts show stepwise increases in the number of Pitx2 expressing cells within the SAN. Downstream of this misexpression of Pitx2, SAN cells show transcriptional changes, downregulating the 'pacemaker program' (including TFs such as ISL1 and Shox2, and key channel HCN4), while upregulating an atrial cardiomyocyte-like program (increasing NKX2.5, and gap junction proteins). The authors also generated human iPSC-derived pacemaker cells with PITX2C overexpression, and found that they undergo transcriptional changes recapitulating the observations in the delB mouse, as well as showing electrophysiological changes indicative of pacemaker dysfunction. The manuscript combines a number of models to provide convincing evidence that increased Pitx2 expression (due to either altered transcriptional regulation or transduction), alters the cell state of pacemaker cells causing a shift towards a more CM-like state with accompanying functional changes.

Response:

We thank the reviewer for supporting our study and for their insightful comments, which have been addressed as detailed below.

Please find specific comments below:

It would be interesting to understand more about how Pitx2 is interconnected with Gata6 expression in regulating SAN development and function. The authors discuss how Gata6 +/- mice also show similar mixed domains of HCN4 positive and negative cells and how both delB and AAV6-PITX2c-hiPSC-PCs show decreased Gata6 expression. Gharibeh, L. et al. also found that Pitx2 is upregulated in Gata6 +/- mice. Would increased Gata6 expression rescue the phenotype seen in the delB or AAV6-PITX2c model? On a more minor note, in Figure 2E Gata6 (and TBX5) is shown without yet any discussion or staining, perhaps this could be incorporated into the final model, or an additional staining could be performed- this would also validate the transcriptional changes seen at the protein level to observe the reduction in Gata6.

Response:

This is an interesting question. We have overexpressed *GATA6* in iPSC-PCs with and without *PITX2* using lentiviral delivery. The effects of *PITX2*-mediated downregulation, i.e downregulation of key pacemaker genes *SHOX2*, *TBX3* and *ISL1*, was not rescued by *GATA6* overexpression (Reviewer Figure 4). While this was a pilot experiment that needs further optimization, it suggests that *GATA6* downregulation in *PITX2* ectopic expression context is not sufficient to cause the phenotype.

Reviewer Figure 4. To assess whether GATA6 rescues *PITX2*-induced transcriptional deregulation in hiPSC-PCs, PCs were transfected with lentivirus (LV) expression vector driving *PITX2* expression. After 4 days, they were transfected with a GATA6 expression vector and PC-associated gene expression was determined by RT-qPCR.

For the immunofluorescence or in situ panels show in figures 1 and 2 how many hearts were examined?

Response:

Panel	Stage	Target	Genotype	n
Figure 1a	E10.5	Pitx2	wild-type	3
			delB/delB	4
Figure 1b	E10.5	PITX2/ISL1/TNNI3/DAPI	wild-type	3
			delB/delB	3
Figure 1c	E12.5	Pitx2	wild-type	2
			delB/+	3
			delB/delB	2
Figure 1d	E17.5	PITX2/ISL1/TNNI3/DAPI	wild-type	3
			delB/+	3
			delB/delB	7
Figure 2a	E12.5	ISL1/TNNI3/DAPI	wild-type	5
			delB/delB	5
Figure 2a	E12.5	HCN4/CX40/TNNI3/DAPI	wild-type	5
			delB/delB	4
Figure 2b	E14.5	HCN4/NKX2-5/TNNI3/DAPI	wild-type	2
			delB/delB	3
Figure 2b	E14.5	ISL1/CX40/TNNI3/DAPI	wild-type	2
			delB/delB	2
Figure 2c	E17.5	HCN4/SHOX2/TNNI3/DAPI	wild-type	3
			delB/+	3

			delB/delB	3
Figure 2d	E17.5	TBX3/ISL1/TNNI3/DAPI	wild-type	2
			delB/delB	2
Figure 2d	E17.5	HCN4/CX40/TNNI3/DAPI	wild-type	3
			delB/delB	4

In figure 2C the HCN4 low and high cells seem to form distinct domains with the HCN4 low domain somewhat consistently towards the outer aspect of the SAN - is this representative and could the authors comments on why this pattern might be occurring if so? In contrast to the sporadic distribution of Pitx2+ cells described for the WT SAN, is the spatial distribution of the HCN4^{low} cells (which presumably are also Pitx2 high) in the *delB* (heterozygotes and homozygotes) SAN regulated?

Response:

Our 3D volumetric reconstructions of the *delB/delB* E17.5 SAN, based on *in situ* hybridization for *Hcn4* and *Gja5* in adjacent sections, show that the formation of the HCN4^{low} domain is not consistently restricted to the other aspect of the SAN (Reviewer Figure 5). However, we cannot exclude the contribution of the PITX2+ myocardium in the dorsal SAN head domain we detected in the *wild-type* SAN to the HCN4^{low} subdomains identified in the *delB* SAN. While variable *Pitx2* expression levels in the mouse SAN coincide with dichotomous PC gene expression profiles, whether and how the spatial distribution of PC transcriptional profiles is regulated remains unclear.

Reviewer Figure 5 *In situ* hybridizations for *Hcn4* and *Gja5* showing the distribution of HCN4^{high} and HCN4^{low} SAN subdomains in different *delB/delB* E17.5 fetuses.

The panels showing changes in CX40 expression are perhaps not the clearest- the arrows help, but alongside the punctae indicated by the arrows there appears to be cytoplasmic/nuclear staining (in figure 2D too). Why might this differential localization be occurring?

Response:

In Supplementary Fig. 2 we have now provided high resolution images of CX40 immunohistochemical staining at E14.5 and E17.5 in the *wild-type* and *delB/delB* SAN and RA as well as the CX40 signal by itself to clearly demonstrate the ectopic activation of CX40 expression in the *delB/delB* fetal SAN and show to that the additional cytoplasmic/nuclear staining is background noise that appears independent of genotype.

In supplemental figure 1B for the ISL1 staining at E12.5 it is slightly difficult to see a clear decrease (also in comparison to the later stages examined, at E17.5), perhaps this could be quantified (number of positive cells or intensity) to make this clearer? This would also clarify the timecourse of this progressive loss of the 'PC program' during SAN development.

Response:

We have modified Supplementary Fig. 1b to more clearly show the graded absence of ISL1, TBX3 and SHOX2 signal in the E12.5 *delB/delB* SAN. We have additionally clarified in the figure legend that the leftmost panel shows the HCN4 expression domain in an adjacent section and that these panels thus illustrate the loss of expression of these transcription factors within the established SAN domain.

In figure 3D are these step wise changes in gene expression also significant (beyond Pitx2)?

Response:

We have plotted the $\text{Log}_2(\text{Fold-change})$ of these genes in Supplementary Fig. 4d and e where the symbols indicate whether the changes in gene expression relative to the *wild-type* SAN or RA are statistically significant. These data are additionally provided in Supplementary Data Tables 1-3.

In figure 3H there are grey dots and purple dots, is the grey representing genes below a cut off (and what would this cut-off be)?

Response:

The purple dots indicate genes that were significantly deregulated ($p < 0.001$) in both the *delB/delB* HCN4^{low} domain and HCN4^{high} domain compared to the *wild-type* SAN. The remaining genes (that are not significantly altered in either the HCN4^{low} and/or HCN4^{high} subdomain) are depicted in grey. To clarify, we have modified the panel legend and figure description.

In figure 3G what statistical test is being used and do the bars represent average values? Is only NKX2.5 significant? Other TFs and ion handling genes appear to have larger fold changes and also given from the normalized count Pitx2 at least did seem to show a significant difference? Is there some underlying variability? There is also a switch from saying WT SAN to WT HCN4+, but does this represent the same population or a subpopulation of the WT SAN cells, i.e. was just the HCN4+ population considered?

Response:

In Figure 3g we plotted the $\text{Log}_2(\text{Fold-change})$ for transcription factors that have previously been implicated in PC differentiation and SAN development as well as functionally relevant atrial CM-associated genes that were significantly differentially expressed upon comparing the *delB/delB* HCN4^{low} regions with the *wild-type* HCN4⁺ SAN ($p_{\text{adj}} < 0.05$). *Nkx2-5* did not reach the significance threshold used in this figure ($p_{\text{adj}} < 0.05$), but its p value reached a less stringent significance threshold ($p < 0.05$.) As it is an important driver of atrial CMs and transition PCs, we included it in the panel and used the asterisk to indicate that in this comparison its p value was < 0.05 .

In the fetal spatial transcriptomic experiments we segmented ROIs based on TNNI3 and HCN4 expression. As such, when we refer to the *wild-type* SAN spatial transcriptomic data sets we do indeed only consider the TNNI3+HCN4+ population.

In supplementary figure 2C the panels on the left are not described (and seems to be possibly be a duplicate of figure 2D)?

Response:

The purpose of these IHC panels is to compliment the ranked relative gene expression plot. We have modified the figure legend to describe them and we have replaced them by a section of a different *delB/delB* fetus to visualize the HCN4+ SAN and the HCN4^{high} and HCN4^{low} regions in the SAN.

In supplementary figure 3C the staining for HCN4 is quite difficult to see, could a different LUT be used, or the LUT inverted? Arrows or a box might also help in distinguishing the positive and negative signal.

Response:

We have made it easier to distinguish between the HCN4^{high} and HCN4^{low} regions.

In order to examine whether increased Pitx2 expression also impacts human pacemaker cells, AAV6-PITX2c-hiPSC-PCs is used, is the level of Pitx2 expression comparable to the hiPSC 15kb +/- model? Is this a physiologically/pathologically relevant level of altered expression?

Response:

The RT-qPCR data from Baudic et al. (PMID: 38643172) indicates 4-fold (range: 1-150 fold) higher *PITX2* expression in CTCF-/- iPSC-PCs compared to unmodified iPSC-PCs. The large range suggests variation in differentiation efficiency, variation in mRNA measurements and possibly also biological variation. Nevertheless, the difference is comparable with the difference between mouse *wild-type* PCs and *delB* PCs. We cannot directly compare these levels with our AAV-mediated *PITX2* expression levels. To gain some insight into whether the level of *PITX2* in AAV6-*PITX2* hiPSC-PCs is (patho)physiologically relevant, we have performed an analysis of the correlation between gene expression and *EGFP* expression (=mCherry or *PITX2*) in AAV6-mCherry and in AAV6-*PITX2* PCs in clusters C0 and C7 (revised Supplementary Fig. 6g). PC genes like *PLN*, *HCN1*, and *HCN4* show negative correlation with *PITX2* levels, not with mCherry control levels, and atrial genes like *CLU* show positive correlation with *PITX2* and not mCherry. As positive control, revised Supplementary Fig. 6h shows positive and negative correlations with *PITX2* and *EGFP* follow the same trend, which is to be expected as both *EGFP* and *PITX2* transcripts report for AAV6-driven *PITX2-p2a-EGFP*.

Furthermore, in Fig. 6e we show that similar relevant transcription factors and ion handling-associated genes deregulated in the *delB* mouse are also deregulated in AAV6-*PITX2c* hiPSC-PCs. Additionally, the functional

perturbation of AAV6-*PITX2c* hiPSC-PCs (including reduced spontaneous activity and increased action potential cycle length) is comparable to that in *delB/delB* mice. Together, these data suggest that the overexpression levels are indeed within a biologically meaningful range.

In supplementary figure 4B are the labels flipped over? It seems that the PITX2 cells are mCherry positive and the mCherry cells are negative?

Response:

Well spotted, thank you. The labels have been adjusted accordingly.

In figure 5E a comparison is made between cluster 0 in the human data and the mouse *delB* HCN4 low population, with many genes showing no correlation, how comparable are these systems given the differences between them (15kb deletion vs *pitx2* OE, in vitro vs in vivo)? Later in 5H when comparisons are made within each system and then presented side by side, the broader conservation of transcriptional changes seems more apparent. Also, in supplementary figure 4I there are fewer PITX2c cells and it is also mentioned that there are more many more PITX2 cells in cluster 8, do these cells show proliferation defects or increased cell death?

Response:

In the *delB* mouse model, *Pitx2* is the only coding gene in the genomic neighborhood of the deletion that is significantly deregulated (Fig. 3c). We therefore assumed that PITX2 is the main mediator of the PC state changes and the SAN dysfunction in deletion carriers. When we modeled this in hiPSC-PCs, we observed that important changes observed in the *delB* mouse model were recapitulated, including the downregulation of all PC-associated genes, and upregulation of several key atrial-enriched genes such as *SCN5A* and *CLU*. Moreover, the electrophysiological measurements (patch clamp) of the hiPSC-PCs further indicated that *PITX2* mediates specific functional changes consistent with the *delB* mouse model and the clinical data. We conclude from this that ectopic expression of *PITX2* in PCs is sufficient to explain most of the observations in the *delB* mouse model and in deletion carriers.

Nevertheless, the hiPSC-PC model is noisy compared to the *in vivo* model, with the formation of different non-PC-like cells, variation among PCs, and variable *PITX2* expression in AAV6-transduced cells, etc. This likely explains the inconsistencies in gene expression response between the 2 models shown in Fig. 6e. We reduced noise by adding a comparison of the hiPSC-PC clusters (cluster C0 and C7) and the mouse *wild-type* and *delB/delB* HCN4^{high} and HCN4^{low} PCs, thus increasing the power of the model. In doing so, we observed comparable behavior of transcriptional responses between C0-*PITX2* PCs vs C7-*PITX2* PCs and HCN4^{high} PCs vs HCN4^{low} PCs, respectively (Fig. 6g).

Regarding the composition of cluster C8, these cells express very high levels of MALAT1 (revised Fig 6b). High levels of *MALAT1* expression has been associated with damaged cells (loss of cytoplasmic RNA: increased nuclear/cytoplasmic RNA ratio) (PMID: 34857027; PMID: 39574015) and with cell stress. C8 is disproportionately composed of cells with AAV6-*PITX2* (revised Fig 6c), raising the possibility that these cells are stressed by high (supra-physiological) levels of *PITX2*.

In the model shown in 7B are the RNA/ATAC seq tracks from mouse or human? And from which model or the WT?

Response:

In Fig. 8b we show the structure of the human (hg19) *PITX2* locus. The RNA-seq track was generated using human iPSC-derived PCs (the same line as used for the *PITX2* overexpression model described here) and the ATAC-seq tracks

were generated from hiPSC-PCs (PMID: 33040635) as well as from human adult LA cardiomyocytes (PMID: 31628324). Figure 8 and the legend have been modified to clearly show that the tracks are human.

Finally, in the manuscript there is discussion about a shift in the metabolism in both of these models of increased *PITX2* expression, but would it be possible to examine this further? Perhaps by staining for mitochondria or looking at the oxygen consumption of the hiPSCs?

Response:

The shift in PC metabolism upon *PITX2* expression is a relevant topic. The impact of *PITX2*-deficiency on energy metabolism and mitochondrial activity in atrial (and ventricular) cardiomyocytes has been described in mouse, fish and iPSC models in PMID: 40099370, 39989351, 39129206, 31704768 and 30143541, whereas aspects of the impact of *PITX2* gain of function was described in PMID: 31704768, 38028792 and 27251288. While we agree that it is relevant to gain insight into the impact of ectopic *PITX2* expression on these important properties in PCs, we consider our transcriptomic data currently sufficiently consistent with the reported data on this topic that we do not prioritize these experiments at this time. We have added this consideration to the Discussion section.

Reviewer #4 (Remarks to the Author):

Here the authors present a study evaluating the effect of dosage-dependent changes in PITX2 expression in the sinus node and atrial arrhythmias. The overall topic is important and the use of complementary models is a strength. Please see below specific comments.

Response:

We thank the reviewer for recognizing the importance of the overall topic and their constructive comments, which we have addressed below.

1. The abstract should be revised to provide a more concise and clearer message with an easier structure to follow, including the broad relevance of the topic.

Response:

The abstract has been revised to make the message/broad relevance more clear and easier to follow.

2. The introduction would also benefit from including a definition of AF for the non-expert and explain why it is important to study it – what are the gaps in that area.

Response:

The introduction has been revised to be more accessible to the non-expert by defining terms like AF and highlighting their importance.

3. Line 208: cell numbers per ROI should be shown in the supplementary data section

Response:

The size of the ROIs and the number of cells per ROI are shown in Supplementary Data Table 14.

4. How was annotation of spatial transcriptomic slides undertaken? The choice for markers used to define the ROI (HCN4 and TNNI3) should be justified. What about for example absence of Cx43/Gja1? What is the range of SIZE of ROIs in wT and mutant mice? How were differences between SAN HCN4^{high}/low determined consistently across samples? The second image in Fig 3a has only part of RA annotated as RA. Why is this?

Response:

The annotation of the slides used for spatial transcriptomics was achieved by staining each section for TNNI3, HCN4 and SYTOX-G. TNNI3 was used to enrich for cardiomyocytes and HCN4 (membrane marker) clearly demarcated PCs from non-PCs as well as the boundaries between HCN4^{high} and HCN4^{low} PCs in *delB* mice. We were able to consistently determine the differences between HCN4⁺, HCN4^{high} and HCN4^{low} regions per slide by refining our ROI selection based on HCN4 signal intensity thresholds. As described in the GeoMx DSP Instrument manual (MAN-10152-06 Software v3.1.2), the maximum size of each ROI is restricted to 660 x 785 μm . As we had already determined that ectopic expression of PITX2 in the atria is restricted to the SAN (PMID: 38643172), we consistently selected anatomically comparable RA regions for reference purposes. The size of the ROIs and the number of nuclei per ROI are shown in Supplementary Data Table 14.

5. Fig 3cP: Plotting logFC rather than counts normalised to wildtype SAN would be better to show the differences.

Response:

We have added the L2FC alongside their statistical significance in revised Supplementary Fig. 4d, e. L2FC of relevant transcription factors and ion channel genes in the *delB/delB* HCN4^{low} domain compared to the *wild-type* SAN are shown in Fig. 3g and that of other genes can be found in Supplementary Data Table 2. We decided to keep the normalized counts in Fig. 3c and d as it is our intention to clearly show the relative expression levels of genes of interest per tissue and genotype. This serves to demonstrate the stepwise changes in gene expression in the *wild-type* SAN and the *delB/delB* SAN subdomains and these levels compare to those in the RA.

6. Fig 5c: How was the reduction in ISL1 in PC cluster quantified?

Response:

The two clusters exhibiting pacemaker markers (C0 and C7) were merged for this analysis. Using the FindMarkers function in Seurat, we compared gene expression between AAV6-*mCherry* and AAV6-*PITX2c* conditions within the merged PC cluster population. The FindMarkers function performs a Wilcoxon rank-sum test to identify genes that are significantly differentially expressed between groups. We used it to quantify the reduction of *ISL1* expression in *PITX2*-overexpressing PCs compared to control. We have modified Fig. 6 to more clearly show the transcriptional differences in AAV6-*mCherry* and AAV6-*PITX2c* cells and clusters 0 and 7. This has made this panel redundant, which was therefore removed.

7. Fig5f: Please clarify what is plotted here. How was the change in proportion of cells quantified? Was this statistically significant? Unless the method used depends on manifold embedding (e.g. Milo) plotting on a UMAP is confusing.

Response:

For each given cluster in the UMAP, we counted the number of cells from the AAV6-*mCherry* and AAV6-*PITX2* samples, and to determine the relative contribution of each sample to a cluster the ratio between these cell numbers normalized to total cell number of each sample was calculated (see Supplementary Fig. 6c). A Z-test (<https://pubmed.ncbi.nlm.nih.gov/12407185/>) was used to determine whether the difference in cell contributions per sample were significant. It is correct that the UMAP clustering itself is not based on these ratios in any way. To clarify this issue, we have replaced the panel showing ratio plotted on UMAP by a simple bar graph showing the relative contribution of each sample to each cluster (revised Fig. 6c).

8. Line 318: cells with high expression of MALAT1 expression were excluded. How was the correlation between MALAT1 expression and quality ascertained? What other quality metrics were used to have a full picture of the quality of the cells?

Response:

We imposed the following quality control cut-offs for determining which cells were included in our analysis: the number of reads detected in an individual cell should be >500, the number of different genes detected >350 and the percentage of reads mapped to mitochondrial genes should be <20%. The default threshold that is used in the Seurat vignette for mitochondrial read percentage is 5%, but for cardiomyocytes, which naturally exhibit elevated mitochondrial gene expression due to their high energy demand, more permissive thresholds can be applied to avoid exclusion of metabolically active, viable cardiomyocytes (<https://www.nature.com/articles/s41586-020-2797-4>).

The long non-coding RNA MALAT1 is highly expressed in nuclear-retained fractions and has been used as a marker for nucleus-containing cells in single-cell RNA-seq workflows. As the reviewer correctly points out, low or absent MALAT1 expression has previously been associated with ambient RNA or cytosolic debris (PMID: 39574015; Clarke and Bader,

2024; <https://www.biorxiv.org/content/10.1101/2024.07.14.603469v1.full.pdf>; and references therein). However, very high MALAT1, such as observed in cluster C8 (revised Fig 6b), has been associated with damaged cells (loss of cytoplasmic RNA: increased nuclear/cytoplasmic RNA ratio)(PMID: 34857027; PMID: 39574015) and with cell stress. C8 is disproportionately composed of cells with AAV6-*PITX2* (revised Fig 6c), raising the possibility that these cells are stressed by high levels of *PITX2*.

More importantly, only clusters C0 and C7 express PC marker genes (revised Fig 6b), which is why we focused on these two clusters, and excluded the others from our analysis delving into the PC response to *PITX2* overexpression, irrespective of their MALAT1 content. We have revised Fig. 6 and the Results section to clarify this issue.

9. How were the scRNAseq data integrated and clustered?

Response:

To integrate scRNAseq data across samples, we used the Seurat v4 integration workflow with SCTransform (SCT) normalization. Raw counts were first normalized using `NormalizeData`, and data were split by sample. Each sample was independently normalized with SCTransform. To ensure consistent gene representation, we restricted the dataset to the common gene set shared across all samples. We then selected 3,000 highly variable features using `SelectIntegrationFeatures`, followed by `PrepSCTIntegration` and `FindIntegrationAnchors` to identify shared biological states across samples. These anchors were used to perform integration via `IntegrateData` (normalization.method = "SCT"), yielding a batch-corrected expression matrix. Dimensionality reduction was performed using PCA, and the first 10 PCs were used to compute a nearest-neighbor graph (`FindNeighbors`). Clustering was performed at multiple resolutions (`FindClusters` with resolution = 0.4, 0.6, 0.8, 1.0, 1.4), and the resolution 0.4 clustering was used for downstream analyses.

10. The discussion should include limitations of the technologies used.

Response:

We have incorporated limitations of the technologies used, including the spatial transcriptomics approach and cell models, in our discussion.

Revisions NCOMMS-25-26166B

Response to reviewers' comments:

We thank the Editor and the Reviewers for their continued efforts to further improve our manuscript and for the opportunity to submit a revised manuscript. We have addressed any additional reviewer points as detailed below.

Reviewer #1 (Remarks to the Author):

The additional data and clarifications that have been incorporated during the revision greatly enhance the manuscript.

I only have 1 outstanding question related to my original comment 1. I appreciate the detailed response to this point. The authors state that they included data for the expression of *Pitx2c* in wildtype vs *delB/delB* mice for the LA and RA. While Fig. 3c shows that there is no change in *Pitx2c* expression in the RA I am struggling to find the data showing *Pitx2c* expression in the *delB/delB* mice in the LA. Fig 4a and 4c only include expression levels in the wildtype LA but not in *delB/delB* LA. To convincingly show that *Pitx2c* expression is unchanged in both the right and left atria and therefore not a cause of the observed atrial fibrillation it would be important to include this data.

Response: We thank the reviewer for supporting our study and for their constructive comments. We apologize for this oversight and putting the *Pitx2* levels only in the supplementary files. The levels of *Pitx2* in the left atrial compartments (LAB, LAA) of *wild-type*, heterozygous and homozygous *delB* mice is shown in revised Fig. 4a-c. *Pitx2* expression levels in left atrial tissues are not different between genotypes. In addition, the PCA plot in Fig. 4b indicates that genotype explains variance between SAN samples, but not between LAB or LAA samples.

Reviewer #2 (Remarks to the Author):

I appreciate the detailed responses to the specific points raised in my original review, including some additional experimental evidence in favor of *Pitx2*-mediated reprogramming. The revised manuscript is improved. I don't have any additional comments.

Response: We thank the reviewer for their appreciation of the additional evidence in favor of *PITX2*-mediated reprogramming and the revised manuscript.

Reviewer #3 (Remarks to the Author):

The authors have carefully addressed all of my queries, and those of the other reviewers.

Response: We thank the reviewer for their efforts and support.

Reviewer #4 (Remarks to the Author):

The authors have improved the introduction which reads well and added a discussion on the limitations.

Response: We thank the reviewer for their continued efforts to further improve our manuscript.

The abstract reads better except for a couple of sentences

1. Line 22 'across a broad range of levels': please clarify

2. Lines 23-24, the sentence is long and could be improved

Response: As we have previously revised our abstract to avoid vague statements like ‘a broad range of levels’ and for clarity, we hope that the current abstract is sufficiently clear.

While authors include a lay explanation of AF the text of the introduction section does not convey what the unmet clinical needs in the realm of AF are and how this work aims to address them. This undermines the importance of findings in this paper.

Response: We have modified the introduction to emphasize the clinical need for interventions that target the pathophysiology of AF and thus, a deeper understanding of AF etiology.

Fig3c and d: These panels don’t add new information compared to plots in Supplementary Fig 4 d and e. In fact, they are confusing as no statistical significance is shown.

Response: Fig. 3c and d have been replaced by Supplementary Fig. 4 d and e to more clearly show differential gene expression in the *delB* SAN. These panels include the statistical significance, which can also be found in Supplementary Data Tables 1-3.

The methods used to analyse spatial transcriptomic data are explained better, but it remains unclear how HCN4^{high}/low domains were annotated. Although it is indicated that this is based on a signal threshold, how is this threshold defined?

Response: Prior to our spatial transcriptomics experiments, as illustrated in Fig. 2c, d and Supplementary Fig. 2, we determined that HCN4/TNNI3 immunohistochemical staining of the anatomically-defined E17.5 *delB* SANs consistently show clearly demarcated HCN4^{high} and HCN4^{low} subdomains. As the difference in HCN4 signal intensity between the HCN4^{high} and in the HCN4^{low} subdomains is greater than the variation in HCN4 signal intensity within the subdomains, we conclude that there is no continuous HCN4 expression gradient between the two subdomains. Per experiment, all tissue samples (sections) were stained together on a single slide and the same HCN4 signal threshold was used to select the subdomain ROIs across all sections. This threshold was set to a signal intensity level that clearly demarcated the distinct HCN4 signals in the HCN4^{high} and the HCN4^{low} subdomains, respectively. This has been clarified in the Methods section.

Fig 5c From the data presented here it remains unclear if there were any statistically significant cell type abundance differences.

Response: The statistics supporting the cell type abundances in each cluster in Fig. 6c are listed in Supplementary Fig. 6c. We have added a reference to these statistics in the Results section and Methods.

The authors respond to the queries about single cell RNA seq analysis pipelines and about quality control metrics used for cell filtering. However, this is not reflected in the methods section of the manuscript.

Response: We have now provided additional information based on Reviewer 4’s previous questions regarding the scRNA-seq analysis pipeline and quality control in the Extended Methods section.

Statistical analysis and the n numbers are missing in several panels including data showing tissue staining (E.g. figure 1 and 2). Some experiments were done with n=2, quantification and statistical analysis are needed across all experiments.

Response: Thank you for pointing this out, we have added further information on sample size, quantification and statistical analyses as needed.